# Estimating fractional snow cover from passive microwave brightness temperature data using MODIS snow cover product over North America

Xiongxin Xiao[1], Shunlin Liang[2], Tao He[1], Daiqiang Wu[1], Congyuan Pei[1], Jianya Gong[1]

[1] School of Remote Sensing and Information Engineering, Wuhan University, Wuhan 430079, China

[2] Department of Geographical Sciences, University of Maryland, College Park, MD 20742, USA

*Correspondence to*: Tao He (taohers@whu.edu.cn)

**Abstract**: The dynamic characteristics of seasonal snow cover are critical for the hydrology management, the climate system, and the ecosystem functions. Optical satellite remote sensing has proven to be an effective tool for monitoring global and regional variations of snow cover. However, accurately capturing the characteristics of snow dynamics at a finer spatiotemporal resolution continues to be problematic as observations from optical satellite sensors are greatly impacted by clouds and solar illumination. Traditional methods of mapping snow cover from passive microwave data only provide binary information at a spatial resolution of 25 km. This innovative study applies the random forest regression technique to enhanced-resolution passive microwave brightness temperature data (6.25 km) to estimate fractional snow cover over North America in winter months (January and February). Many influent factors, including land cover, topography, and location information, were incorporated into the retrieval models. Moderate Resolution Imaging Spectroradiometer (MODIS) snow cover products between 2008 and 2017 were used to create the reference fractional snow cover data as the "true" observations in this study. Although over- and under-estimation around two extreme values of fractional snow cover, the proposed retrieval algorithm out-performed the other three approaches (linear regression, artificial neural networks, and multivariate adaptive regression splines), using independent test data for all land cover classes, with higher accuracy and no out-of-range estimated values. The method enabled the evaluation of the estimated fractional snow cover using independent datasets, where the root-mean-square error of evaluation results ranged from 0.189 to 0.221. The snow cover detection capability of the proposed algorithm was validated using meteorological station observations with greater than 310 000 records. We found that binary snow cover obtained from the estimated fractional snow cover was in good agreement with ground measurements (kappa: 0.67). There was significant improvement in the accuracy of snow cover identification using our algorithm; the overall accuracy had increased by 18% (from 0.71 to 0.84), and the omission error had reduced by 71% (from 0.48 to 0.14), when the threshold of fractional snow cover was 0.3. The experimental results show that passive microwave brightness temperature data may potentially be used to estimate fractional snow cover directly, where this retrieval strategy offers a competitive advantage in snow cover detection.

## 1. Introduction

Snow cover is a critical indicator of climate change, playing a vital role in the global energy budget (Flanner et al., 2011), water cycle (Gao et al., 2019), and atmospheric circulation (Henderson et al., 2018). Snow cover directly modulates the release of carbon and methane from underlying soil (Zhang, 2005; Zona et al., 2016), and influences permafrost conditions and active layer dynamics (Zona et al., 2016). Snowpack also stores a huge amount of water providing for both domestic and industrial water needs (Sturm, 2015; Cheng et al., 2019). Accurate and timely monitoring of the spatiotemporal variation of snow cover is beneficial for hydrologic forecasting, climate predictions and water resources management (Barnett et al., 2005; Bormann et al., 2018).

Snow cover data is typically obtained from meteorological stations or in-situ manual measurements, which is spatially discontinuous and labor intensive. Remote sensing has become an attractive alternative tool to ground-based measurements as it is able to cover a wide area and is capable of high frequency observations. Numerous studies have focused on snow cover detection and snow cover products used optical and microwave satellite data (Tsai et al., 2019; Liu et al., 2018; Hori et al., 2017). Most of these snow cover products provide binary information at the pixel-level; snow-covered or snow-free. However, snow cover often varies within a limited scale area, characterized by high spatial heterogeneity, especially in alpine terrain areas. Dobreva and Klein (2011) demonstrated that the use of binary snow cover classification in snow cover area estimation may produce considerable uncertainties. Binary snow cover lacking fractional features hinders accurate characterization of the spatial distribution of snow cover and cannot accurately capture variations in seasonal snow cover dynamics. In terms of the energy budget perspective, binary snow cover introduces significant uncertainties into the global energy budget estimation because of the large surface albedo differences between snow-covered and snow-free surfaces (He et al., 2014). Thus, there is an urgent need to acquire snow cover area within a sub-pixel to provide accurate snow cover information. Fractional snow cover allow for the derivation of snow cover area at the sub-pixel level; this is a better option compared to binary snow cover (Salomonson and Appel, 2004).

Fractional snow cover maps derived from optical imagery have been produced for over 40 years. Optical satellite observations have been recognized for their suitability in estimating fractional snow cover because of their high spatial resolution. Moderate- to high- resolution optical observations have been popular in previous snow cover studies, for example Fengyun (FY) series sensors (0.5 – 4 km) (Wang et al., 2017), Moderate Resolution Imaging Spectroradiometer (MODIS) (500 m) (Kuter et al., 2018), and Landsat (30 m) (Berman et al., 2018). There are also many predictive methods for fractional snow cover, such as linear regression (Salomonson and Appel, 2004; Salomonson and Appel, 2006), spectral mixture analysis (Wang et al., 2017; Rosenthal and Dozier, 1996), machine learning (e.g., artificial neural network, ANN) (Liang et al., 2017; Moosavi et al., 2014), and multivariate adaptive regression splines (MARS) (Kuter et al., 2018). A simple linear regression cannot fully describe the complex relationship between satellite observations and fractional snow cover. As such, non-linear approaches

have recently been developed to replace this traditional method (Berman et al., 2018). Kuter et al. (2018) estimated fractional snow cover from MODIS data using the MARS technique, where the Landsat 8 binary snow cover data served as the reference fractional snow cover data. They found that the estimated fractional snow cover using MARS method was in good agreement with the reference fractional snow cover, with the average correlation coefficient being R = 0.93 (Kuter et al., 2018). However, polar regions contend with clouds and limited solar illumination, which are the greatest challenges for snow cover detection using optical satellite data. This has resulted in snow cover maps with incomplete spatial coverage, at times with gaps of up to 70% (Parajka and Blöschl, 2008). Although there have been constant efforts to fill the gaps mainly caused by cloud contamination by fusing multi-source data (Chen et al., 2018), such as passive microwave snow cover products (Hao et al., 2018; Huang et al., 2016), and different spatiotemporal information on snow cover (Dong and Menzel, 2016; Gafurov and Bárdossy, 2009), most studies have focused on binary snow cover.

When there are consecutive cloudy days, the use of data fusion technology introduces significant uncertainties in detecting snow cover from optical imagery. Passive microwave sensors are largely advantageous because they have capacity to measure microwave radiation emitted from the ground under the clouds and in darkness. Compared with active microwave sensors, passive sensors, have a large swath width and generate a large amount of daily observation that extends for several decades (Cohen et al., 2015).To date, passive microwave brightness temperature data has been widely applied in monitoring soil moisture (Qu et al., 2019), sea/lake ice (Peng et al., 2013), frozen soil (Han et al., 2015), and snow cover. Previous studies on snow cover usually have typically focused on snow depth (Xiao et al., 2018; Che et al., 2008), snow water equivalent (SWE) (Takala et al., 2011; Lemmetyinen et al., 2018) and snow cover area (Liu et al., 2018; Xu et al., 2016). All studies on snow cover area were limited to binary information. Specifically, they involved the application of common passive microwave snow cover mapping algorithms, such as Grody's algorithm (Grody and Basist, 1996), National Aeronautics and Space Administration (NASA) Advanced Microwave Scanning Radiometer – Earth Observing System (AMSR-E) SWE algorithm (Kelly, 2009), Singh's algorithm (Singh and Gan, 2000), Neal's algorithm (Neale et al., 1990), the FY3 algorithm (Li et al., 2007), and the South China algorithm (Pan et al., 2012). All these algorithms utilize different thresholds for brightness temperature to identify binary snow cover. Recently, Xu et al. (2016) applied the brightness temperatures of different channels and their linear combinations into the Presence and Background Learning (PBL) algorithm to identify global binary snow cover.

As the effect of environmental factors (e.g., vegetation, topography, and wind) on snow cover distribution produces great heterogeneity, snow cover monitoring still bears larger uncertainties when only using passive microwave data. These large uncertainties may result from "patchy" (shallow/discontinuous) snow cover and the use of coarse resolution (25 km) (Xiao et al., 2018). Despite the coarse resolution of passive microwave sensors, its ability to detect snow cover in the presence of clouds demonstrates its effectiveness as a snow cover monitoring tool. There is an urgent need for daily time-series and full space-

covered sub-pixel snow cover area data for climate and reanalysis studies. Thus, it is necessary to derive high resolution fractional snow cover that can describe snow cover distribution patterns and capture its rapid evolution processes. Brodzik et al. (2018b) recently published the Calibrated Enhanced-Resolution Passive Microwave Daily Equal-Area Scalable Earth Grid (EASE-Grid) 2.0 Brightness Temperature data (see Section 2.1 below), which has high spatial resolution (3.125 and 6.25 km) depending on frequency (Brodzik et al., 2018a; Long and Brodzik, 2016). This passive microwave data with enhanced resolution provides an opportunity for fractional snow cover estimation.

The main objective of this study is to develop a feasible method utilizing the enhanced-resolution passive microwave brightness temperature data to retrieve daily fractional snow cover at a 6.25 km resolution. The datasets used in this study are described in Section 2, including the enhanced-resolution passive microwave data, ground-based measurements, MODIS snow cover and land cover products, and topographic data. Section 3 details the proposed retrieval algorithm using the random forest method as a retrieval function. Section 4 presents the results from the methods comparison, evaluation, and validation experiments. Finally, Section 5 discusses the possible factors that impact on the accuracy of the fractional snow cover estimates derived from passive microwave data.

## 2. Datasets

### 2.1 The enhanced-resolution passive microwave data

The NASA Making Earth System Data Records for Use in Research Environments (MEaSUREs) program provides a new version of passive microwave brightness temperature data known as the Calibrated Enhanced-Resolution Passive Microwave Daily EASE-Grid 2.0 Brightness Temperature. This passive microwave gridded data spans from 1978 to mid-2017, using the Level-2 satellite records from multiple passive microwave sensors (Brodzik et al., 2018b; Brodzik et al., 2016, Updated 2018.). This enhanced-resolution data may be downloaded from the National Snow and Ice Data Center (NSIDC, https://nsidc.org/data/NSIDC-0630/versions/1). To explore the feasibility of estimating fractional snow cover using passive microwave data, this study mainly selected January and February of 2008 – 2017 as the study period. The Special Sensor Microwave/Imager (SSMIS) sensor (F-16) used in this present study offers three channels (19, 37 and, 91 GHz) in both horizontal (H) and vertical (V) polarization, and 22 GHz with vertical polarization. These datasets were gridded into EASE-Grid 2.0 projections at two spatial resolutions (19 and 22 GHz with 6.25 km, 37 and 91 GHz with 3.125 km). Only observations from descending orbit (morning, 03:52) were used to avoid the effects of wet snow as much as possible (Derksen et al., 2000). To achieve a common resolution, we aggregated the 3.125 km spatial resolution data to 6.25 km by averaging the surrounding 4 pixels.

## 2.2 Ground measurements

Although ground measurements of snow cover have limited spatial representation in passive microwave coarse spatial resolution, in-situ measurements continue to be the most authentic and reliable data source for snow depth estimation or snow cover detection (Chen et al., 2018; Sturm et al., 2010). Ground measurements from the Global Historical Climatology Network-Daily (GHCN-Daily) data were used to assess the snow cover detection capability (Menne et al., 2012a). The GHCN-Daily dataset was provided by the National Climatic Data Center (available in http://doi.org/10.7289/V5D21VHZ), and integrates daily observations from approximately 30 different data sources. The new version data was updated on June 13, 2018, and contained measurements from over 100 000 stations worldwide. These stations record various aspects of meteorological observations, including snow depth and snowfall (Menne et al., 2012b). More than 50 000 measurement sites across Canada and United States were collected, and all available records applied in validation stage are from approximate 18000 sites.

## 2.3 MODIS land surface products

### 2.3.1 Snow cover product

MODIS snow cover products were considered the most suitable reference data because of their wide application, high accuracy (Hall and Riggs, 2007; Zhang et al., 2019; Coll and Li, 2018), and high spatiotemporal resolution (1 day; 500 m). The accuracy of the version 6 MODIS snow cover products has improved compared to that of version 5 (Dong et al., 2014; Huang et al., 2018). The most noticeable change for version 6 is that the Normalized Difference Snow Index (NDSI) snow cover has replaced fractional snow cover, while binary snow covered area (SCA) datasets are no longer available (Riggs and Hall, 2016). A snow cover detection method using NDSI was applied in version 6 to alleviate commission errors (Riggs et al., 2017). The NDSI index helps to distinguish snow from other surface features and to describe the presence of snow (Hall et al., 1998; Hall et al., 2001). These products were available from NSIDC website (MOD10A1: https://nsidc.org/data/MOD10A1; MYD10A1: https://nsidc.org/data/MYD10A1)(Hall and Riggs, 2016a, b). The local equatorial crossing times of MODIS onboard the Terra and Aqua satellites are approximately 10:30 a.m. and 01:30 p.m., respectively. This study used both MOD10A1 and MYD10A1 NDSI snow cover products to generate reference fractional snow cover for North America. The NDSI snow cover data was initially converted to binary snow cover for aggregation into fractional snow cover data at 6.25 km spatial resolution (see Section 3.2).

### 2.3.2 Land cover product

Generally, the retrieval accuracy of snow cover parameters is strongly dependent on the land cover type (Xiao et al., 2018; Kuter et al., 2018; Dobreva and Klein, 2011; Huang et al., 2018). We indirectly considered the land cover effect when estimating fractional snow cover by establishing retrieval models on different land cover classes derived from MODIS land cover data (2008 – 2017). MODIS Land Cover Type Yearly Product (MCD12Q1, version 6) incorporates five different

classification schemes and is globally available at a 500 m spatial resolution spanning 2001 to the present (https://search.earthdata.nasa.gov/). The International Geosphere–Biosphere Program (IGBP) classification scheme categorizes land cover into 17 classes (Sulla-Menashe and Friedl, 2018). In this study, MCD12Q1 data was resampled into the 6.25 km grid using a simple majority method, then integrated into five classes; forest, shrub, prairie, bare land, and water (refer to Xiao et al. (2018)). Fractional snow cover retrieval models were established for four of these land cover types, excluding water.

**2.4 Topographic data**

Previous studies have demonstrated that topography plays an important role in snowpack distribution (Dai et al., 2017) and snow evolution (Savoie et al., 2009). The ETOPO1 data was used as the topographic auxiliary data; this data has a 1 arc-min spatial resolution and was developed by the National Geophysical Data Center of the National Oceanic and Atmospheric Administration (NOAA) (Amante and Eakins, 2009). This study also considered elevation, slope, and aspect factors. Elevation was directly acquired from ETOPO1, which was re-projected and resampled into the grid at the 6.25 km spatial resolution. The slope and aspect data were obtained from ETOPO1 data by ArcGIS 10.5 (Buckley, 2019). Fig. 1 shows the elevation pattern for North America, limited to Canada and United States in this study.

**3. Methodology**

Microwave radiation constantly emitted from the substratum can be measured by passive microwave sensors. However, the overlying snow pack attenuates the upward microwave radiation (Chang et al., 1987). This microwave radiation attenuation was mainly dominated by volume scatter relying on properties of the snow cover. Previous studies have demonstrated that there is great heterogeneity in snow properties and the distribution of snow cover, both of which may be influenced by many factors (Xiao et al., 2019), including, the most prevalent land-cover (Che et al., 2016; Kim et al., 2019), topography (e.g., elevation, topographic relief) (Smith and Bookhagen, 2016; Revuelto et al., 2014), time (Sturm et al., 2010; Dai et al., 2012), and climatic conditions (e.g., wind speed, near-surface soil temperature and air temperature) (Dong et al., 2014; Grippa et al., 2004; Josberger and Mognard, 2002). Satellite sensors receive reduced upwelling microwave radiation in proportion to a greater snow cover area or a larger mass of snowpack (Chang et al., 1987; Dietz et al., 2011; Saberi, 2019). A number of published works have demonstrated the potential to derive snow depth and SWE using passive microwave radiation data (Kim et al., 2019; Wang et al., 2019). Despite the high uncertainties associated with snow depth and SWE estimations, using passive microwave data can provide useful snow cover extent information (Brown et al., 2010; Foster et al., 2011).

**3.1 Overview**

To develop a fractional snow cover prototype retrieval method combined with optical and passive microwave data, we

only used the January and February datasets, as snow cover areas are at a maximum and snowpack properties are relatively stable during this period (Xiao et al., 2018). The influential factors on snow cover, including topography factor, land cover, location and time, were indirectly or directly considered during retrieval of the fractional snow cover. To date, many researchers have applied machine learning techniques for the retrieval of snow cover parameters to explore the relationship between passive microwave signals and snow properties (Xiao et al., 2018; Tedesco et al., 2004). Random forest is an ensemble learning method, gaining the attention of many researchers because it is more efficient and robust than the single method (Breiman, 2001). As a classifier, random forest has been successfully employed to detect snow cover (Tsai et al., 2019), land cover (Rodriguez-Galiano et al., 2012), and woody invasive species (Kattenborn et al., 2019). The random forest regression method can also successfully estimate land surface temperature (Zhao et al., 2019), biomass (Mutanga et al., 2012), and soil moisture (Qu et al., 2019). In this study, random forest regression (described in Section 3.4.4) was selected as the retrieval method to mine the relationship between passive microwave brightness temperature and fractional snow cover. We also compared random forest with other three methods (linear regression, MARS and ANN) widely used in fractional snow cover retrieval from optical remote sensing data in model performance. Fig. 2 provides an overview of the workflow that consists of four parts:

First, a ground "truth" observation was necessary to produce snow cover areas in sub-pixel. Under clear-sky conditions, the reference fraction of snow cover was generated within a 6.25 km pixel cell by applying the aggregation method to the MODIS binary snow map (see Section 3.2). To make the experiment to be fully independently, the reference fractional snow cover data was divided into three parts: the data from 2011 to 2016, used in the training stage; the data from 2010, used in the testing stage; and the independent datasets in January and February (2008, 2009, 2017) and in December (2007, 2008, 2016), used in the evaluation stage.

Second, to the best of our knowledge, there are few attempts to directly develop fractional snow cover from passive microwave brightness temperature data. This meant a series of sensitivity experiments of input variables selection were required. Input parameters were selected based on a series of tests described in Section 3.3.1. Moreover, we conducted several sensitivity experiments to determine the optimal training sample size for the retrieval method used in this study (Section 3.3.2).

Third, many studies found that the separate estimation of fractional snow cover (Dobreva and Klein, 2011) and snow depth (Xiao et al., 2018) on different land cover types produced better results than those obtained from the combined retrieval model. As such, the random forest models were developed separately for the four land cover types.

Fourth, the last stage consisted of evaluation and validation of the established model. Data from 2010 was used to assess the performance of four different approaches (linear regression, MARS, ANN, and random forest) for estimating fractional snow cover. Additionally, the independent datasets were used to evaluate the performance of the random forest-based retrieval algorithm for the four land cover types. Independent validations of snow cover detection capability were conducted using the fractional snow cover retrieval results and station snow depth measurements across North America. There were compared with

the results of Grody's snow cover mapping algorithm.

## 3.2 Preprocessing of MODIS snow cover products

The base data for this study was the reference fractional snow cover data obtained from the interpretation of MODIS snow cover products. The highest priority was to produce daily binary snow cover area from NDSI snow cover. Previous snow cover detection studies recommend a 0.4 NDSI threshold on global and regional scale snow cover investigations (Parajka et al., 2012; Hall et al., 1995); However, for the new version of MODIS snow cover products, several studies employed a threshold of NDSI > 0 to identify snow cover (Dong et al., 2014; Riggs et al., 2017; Huang et al., 2018). The NDSI of other features (e.g., cloud-contaminated pixels at the edges of cloud, salt pans, and pixels with very low visible reflectance) may also be greater than 0 (Riggs et al., 2017). For this reason, Zhang et al. (2019) demonstrated that a 0.1 NDSI threshold was more reasonable than 0.4 for snow cover identification in no-forest regions, whereas, forest-covered regions insufficient station measurements for a reliable and complete evaluation. MODIS snow cover performance is better for non-forest landscapes than forest-covered counterparts, where it is less accurate for snow cover identification (Hall and Riggs, 2007; Parajka et al., 2012).

This study selected conservative NDSI thresholds of 0.1 and 0.4 for no-forest-covered areas and for forest-covered areas, respectively (Riggs and Hall, 2016) to determine "snow-covered" or "snow-free" areas. The original NDSI snow cover layer classes were reclassified into five types; snow-covered, snow-free, water, cloud, and fill value (refer to Table S1 in Appendix). In addition, MCD12Q1 datasets (500 m) were used as auxiliary data to mask water bodies (Fig. 3) in order to alleviate the uncertainty caused by frozen water bodies when using passive microwave data to detect snow cover (Tedesco and Jeyaratnam, 2016). The MODIS binary snow cover data was generated based on the NDSI snow cover basic quality assessment (QA), with values of 0 (best), 1 (good) and 2 (OK) (Liang et al., 2017).

Despite the high spatiotemporal resolution and overall accuracy of snow cover detection (85% – 99%) using MODIS snow cover products (Parajka et al., 2012; Tran et al., 2019; Zhang et al., 2019), the cloud effect hinders its widespread applicability. Previous studies have reported that clouds may cover more than 40% of MODIS snow cover data, in some cases exceeding 60% (Dong and Menzel, 2016; Yu et al., 2016; Parajka and Blöschl, 2006). As such, cloud removal processing is essential to mitigate the cloud obstruction of MODIS products. This study adopted the cloud removal method combining the MOD10A1 and MYD10A1 snow cover products, as proposed by Gafurov and Bárdossy (2009). This method consists of two main filters as shown in Fig. 3:

1) Combining snow cover images from two sensors on a given day: the first simple filter was applied under the assumption that snowmelt and snowfall did not occur within the two sensor observations. Whether a pixel in Terra ($S_t^{Terra}$) or Aqua ($S_t^{Aqua}$) snow cover image in a given day (t) was observed as snow cover or snow-free, the pixel in the output image (MCD10A1) was assigned the same ground status (shown in Eq. 1). The results showed about 3% of cloud cover was

removed compared to MOD10A1 (Gafurov and Bárdossy, 2009).

2) Short-term temporal filter: if the status of a pixel in the input image (MCD10A1) in a given day (t) was cloud and both the preceding (t - 1) and succeeding (t + 1) days were snow-covered (or snow-free), the pixel in the output image (MCTD10A1) in the given day (t) was assigned as snow-covered (or snow-free) (summarized by Eq. 2). Compared to the first filter, this short-term temporal filter may markedly reduce the number of days (10% ~ 40%) for cloud coverage and increase the overall accuracy of snow cover detection (Gafurov and Bárdossy, 2009; Tran et al., 2019).

$$S_{(output,t)} = max\left(S_t^{Aqua}, S_t^{Terra}\right) \tag{1}$$

$$S_{(output,t)} = 1 \ if\left(S_{(t-1)} = 1 \ and \ S_{(t+1)} = 1\right) \tag{2}$$

where, t is the time and S represents the ground status observed in the image (0 or 1); 0 denotes cloud presence and 1 indicates snow-covered or snow-free.

Theoretically, the MODIS fractional snow cover map should calculate the percentage of snow cover in a strictly delimited area of the passive microwave pixel. Calculated areas should have a larger footprint area than the pixel resolution to avoid MODIS geolocation uncertainties (Wolfe et al., 2002; Dobreva and Klein, 2011). In this study, a window of 15*15 pixels of MODIS binary snow cover data (MCTD10A1; 500 m) was used to calculate the fraction of snow cover in a 6.25 km pixel. We adopted the most rigorous pixel filtering rule, by which one clouded pixel cannot be allowed within a 15*15 pixels window. This is slightly different from a previous study that allowed 10% of clouds (Dai et al., 2017).

**3.3 Sensitivity study**

**3.3.1 Selecting input variables**

After determining the retrieval function, selecting the fewest number of variables to establish an efficient estimation model is a major challenge (Mutanga et al., 2012). Many factors influence snowpack distribution, and the consideration of all factors in snow cover properties estimation is unrealistic. Therefore, we conducted six scenarios to evaluate and finally screen input variables. The topographic factors (digital elevation model (DEM), slope, aspect) (Revuelto et al., 2014) and location information (longitude and latitude) (Xiao et al., 2018; Sturm et al., 2010) were directly take as the basic input variables. Additionally, consideration was also given to the passive microwave brightness temperature (19 GHz, 37 GHz, and 91 GHz; both H and V polarization) (Xiao et al., 2018; Xu et al., 2016) and the difference of brightness temperature between different channels (Xu et al., 2016; Liu et al., 2018) (listed in Table 1). The 22 GHz channel was excluded because it is sensitive to water vapor.

A decision tree was established using all variables shown in Scenario -1 (Table 1), and was utilized to compare with five scenarios in terms of prediction performance and efficiency. Note that these 19 input variables were determined by using the Correlation Attribute Evaluation method in the Waikato Environment for Knowledge Analysis 3.8.3 (WEKA) data mining software. This method evaluates the importance of the attribute by measuring the correlation between the attribute and the

target (Frank et al., 2004; Eibe Frank, 2016). The brightness temperature and its linear combination can also directly be used to detect snow cover based on Xu et al. (2016) study; thereby, Scenario -2 only contained brightness temperature and its linear combination without consideration to the effects of location and topographic factors. Wiesmann and Mätzler (1999) reported that V and H polarizations were dominated by scattering and snow stratigraphy, respectively. Thus, Kim et al. (2019) only assimilated V polarization with an ensemble snowpack model to estimate snow depth. Therefore, in Scenario -3, we attempted to evaluate the performance of the established retrieval model by only using the brightness temperature in 19, 37 and 91 GHz (V polarization) based on Wiesmann and Mätzler (1999) and Kim et al. (2019). In Scenario -4, we used similar input variables to those used for snow depth estimation in Xiao et al. (2018), and examined whether these same parameters can or cannot estimate the fractional snow cover. In Scenario -5, unlike the variables used in Scenario -4, we attempted to use the basic input variables coupled with the brightness temperature linear combination for fractional snow cover retrieval.

There are other variable selection strategies based on the importance rank when using random forest method. For example, Mutanga et al. (2012) implemented a backward feature elimination method to progressively eliminate less important variables, whilst Nguyen et al. (2018) summarized the grade of the variable and selected the top eight important variables as the input variables in the training model. Similarly, this study assessed the importance of input variables on four land cover types using the same size of the training sample (15 000) (Xiao et al., 2018). We then counted the number of times of each variable that was ranked in the top nine important variables (summarized in Table S2, Appendix), which were then used as the input variables for Scenario -6 (listed in Table 1). By assessing the performance of models established by these six scenarios, an optimal combination of input variables for the fractional snow cover retrieval model may be selected (see Section 4.1.1). All input variables were normalized to [0, 1].

### 3.3.2 Determining sample size

Although the random forest method can avoid overfitting (Breiman, 2001), it is important to evaluate its sensitivity to sample selection types and the size of the training sample (Belgiu and Drăguţ, 2016; Millard and Richardson, 2015; Nguyen et al., 2018; Colditz, 2015). The performance of predicted models trained by machine learning methods is strongly dependent on the quality of the training sample (Dobreva and Klein, 2011). Good quality training samples indicate that the sample data is not biased towards a certain value. The distribution of the fractional snow cover value from our dataset shows that more than 70% of values were near 0 and 1. As such, the use of the random selection or equal proportional selection method (Millard and Richardson, 2015; Lyons et al., 2018; Nguyen et al., 2018) would hinder the interpretation of the final fractional snow cover estimation model by reducing the accuracy of the estimation. To address this, we adopted stratified random sampling as a sample selection strategy (Xiao et al., 2018; Dobreva and Klein, 2011), where stratification was performed on the fractional snow cover at 0.01 increments.

From previous studies, the sample size, approximately 0.25% of the total study area, was adopted by Colditz (2015) when

using the random forest method. This value has also been evaluated in optical and active remote sensing studies (Nguyen et al., 2018; Du et al., 2015). In this study, we generated the training sample datasets separately from 0.15% to 0.35% of the total cover area for each land cover class (in 0.05% increments). Then, sensitivity tests were carried out for the four land cover types. This means the training dataset would represent the values of fractional snow cover categories for each land cover type (see Section 4.1.2). All selection operations were completely random.

### 3.4 Description of different estimation methods

In this study, we compared the random forest method with the other three methods for retrieving fractional snow cover, including linear regression, ANN, and MARS. Note that the four methods input the same variables selected by the sensitivity test, including 12 characteristic variables and one target variable (see Section 3.3.1 and 4.1.1).

### 3.4.1 Linear regression

For optical remote sensing studies, there is a classical and general linear regression method used to estimate the sub-pixel snow cover area in medium- to high-spatial-resolution image. This only involve the relationship between NDSI and fractional snow cover derived from high-resolution snow cover maps (Salomonson and Appel, 2004; Salomonson and Appel, 2006). This type of regression method has been applied in generating the standard MODIS fractional snow cover product Collection 5. Similarly, the multiple linear regression method was used as a reference method in this study to estimate fractional snow cover based on passive microwave data. The inputs were the same as the other three methods in this study. This method was undertaken in WEKA 3.8.3 and did not use any attribute selection method. In the Appendix, we present the linear regression formulas of fractional snow cover estimation for the four land cover types (Eq. S-1 and Table S6).

### 3.4.2 ANN

ANN is a popular machine learning technique widely applied in remote sensing studies. Tedesco et al (2004) developed an SWE and snow depth retrieval algorithm based on an ANN technique using passive microwave brightness temperature. Xiao et al. (2018) also sued ANN to derive snow depth, and Kuter et al. (2018) and Czyzowska-Wisniewski et al. (2015) used ANN to retrieve fractional snow cover from MODIS data.

ANN consists of multiple layers; an input layer, one or more hidden layers, and an output layer (Hecht-Nielsen, 1992). The network with multilayer perceptron can easily handle the nonlinear relationship between the input and output without any prior knowledge (Haykin, 2009). The inputs of each neuron were multiplied and summed by the connection weight. The output results were subsequently computed using a nonlinear logistic sigmoid transfer function. For numerical data, the transfer function in WEKA substitutes the pure linear unit function for the logistic sigmoid.

Aside from data preprocessing, a crucial step in this process is to design and optimize the ANN structure for improved

estimation performance and good generalization capability (Kuter et al., 2018). Kuter et al. (2018) demonstrated that multidimensional function modeling can successfully achieved with one hidden layer network. All parameters were set to the default with the exception of the learning rate, which was optimized through a simple trial-and-error method. Based on the accuracy index and the modeling speed, Table S3 (Appendix) shows that a learning rate of 0.2 generated the best performance for the ANN-retrieval model.

### 3.4.3 MARS

The MARS technique has been applied in a number of studies and in many fields, such as classification and mapping (Quirós et al., 2009), atmosphere correction (Kuter et al., 2015), pile drivability prediction (Zhang and Goh, 2016), and fractional snow cover estimation (Kuter et al., 2018). Unlike ANN, the modeling process of MARS is flexible and straightforward. Friedman (1991) first proposed a MARS technique that organizes a simple model for the complex and high-dimensional relationship between input variables and the target by smoothly connecting piecewise linear polynomials (known as basis functions (BFs)). The ranges of the input variables were cut into a series of sub-ranges by knots; these were the connection points for two pieces of BFs. A simple BFs format of MARS is expressed in Eq. 3, where $max(\cdot)$ indicates that only positive parts were take; otherwise, it was assigned a zero; and $\tau$ is a univariate knot.

$$max(0, x - \tau) = \begin{cases} x - \tau, & if\ x > \tau \\ 0, & otherwise \end{cases} \tag{3}$$

The MARS method involves two phrases to establish a regression model; forward phase and backward phase. In the forward phase, BFs were generated using a stepwise search of all univariate candidate knots and all variables interactions. These adopted knots and their corresponding pair of BFs should produce the greatest decrease in residual error. The BFs were successively added to the model until it reached the maximum number of BFs, resulting in an over-fitted and complicated model. In the backward phase, the redundant BFs that least contribute to model prediction are completely excluded from the regression model. These two phases are an iterative process (Kuter et al., 2018; Zhang and Goh, 2016).

Two important parameters of MARS determine the model "growing" and "pruning" processes; the maximum number of basis functions (max_BFs), and the maximum degree of interactions among input variables (max_INT) (Kuter et al., 2018). Kuter et al. (2018) reported that the increase in the structural complexity of the model does not significantly contribute to improving the performance of the MARS model. We conducted several tests to optimize the structure of MARS and found that more complex structures had a longer modeling time, however, did not significantly improve model performance. Specifically, the modeling time of the complex structure (max_BFs = 100, max_INT = 2) was four times greater than the simple structure (max_BFs = 40, max_INT = 2) based on our analysis experiments. As such, the simple structure was chosen, as per Kuter et al. (2018). We implemented an open MARS MATLAB source code available from Jēkabsons (2016) for fractional snow cover estimation. These codes were compiled on a 2.40 GHz Intel Xeon Central Processing Unit (CPU) server.

### 3.4.4 Random forest

Random forest builds a large series of decision trees by applying the bootstrap sampling method. During the training stage, each tree grows by randomly selecting several variables and samples from input datasets (Mutanga et al., 2012). Input data was repeatedly split into training and test data using the bootstrapping method. Each randomly selected bootstrap sample in each iteration contained approximately 2/3 of the input elements. The remaining data, referred as out-of-bag (OOB) data, was used for validation. The predicted value of OOB data was produced from all the tree results that were generated and the OOB error was subsequently calculated. For classification, the output was determined by voting the results from all decision trees, whereas for regression, the output results were determined by averaging. The random forest was conducted in WEKA 3.8.3. As several attempts to optimize the parameters of random forest structure had failed, all parameters used were the default values.

### 3.5 Snow cover identification

The microwave radiation characteristics of precipitation, cold deserts and, frozen ground are similar to those of snow cover (Grody and Basist, 1996), and as such, snow cover area is likely to be overestimated. Grody and Basist (1996) proposed a snow cover identification algorithm, distinguishing snow cover from precipitation, cold desert, and frozen ground. Many researchers have since used Grody's algorithm and its derivative algorithm to detect snow cover (Che et al., 2008; Xiao et al., 2018; Wang et al., 2019). Liu et al. (2018) reported on the assessment of different passive microwave snow cover detection algorithms and demonstrated that Grody's algorithm had a higher precision (positive predictive value) than those of other algorithms. We adopted the revised snow cover decision tree of Grody's algorithm (Table 2) as the highest frequency in this study was 91 GHz instead of 81 GHz in Special Sensor Microwave Images/Sounder (SSMIS) sensors (Che et al., 2008).

There were two main objectives for using the revised Grody's algorithm (hereafter referred to Grody's algorithm) in this study. The first was to compare the snow cover identification capability of the proposed fractional snow cover estimation algorithm with respect to ground snow depth measurements (see Section 4.4). The second purpose was to assess the effect of non-snow scatterer in estimating fractional snow cover, due to this algorithm's special ability to distinguish the non-snow scatterer (i.e., precipitation, cold desert, and frozen ground). In both optical and microwave remote sensing research, capability assessment of snow-free detection has been regularly neglected in most snow cover detection studies.

### 3.6 Validation of snow cover identification

When using the in-situ snow depth (or SWE) measurements to quantitatively validate the accuracy of snow cover area data, converting snow depth into binary snow cover using an appropriate threshold is the first challenge. Many different depth thresholds have been suggested in previous studies, for instance 2 cm for 20 m spatial resolution (Gascoin et al., 2019); 0 cm (Parajka et al., 2012), 1 cm (Zhang et al., 2019), 3 cm (Hao et al., 2018), 4 cm (Huang et al., 2018; Wang et al., 2008) and, 15

cm (Gascoin et al., 2015) for 500 m spatial resolution; 2.5 cm for 5 km spatial resolution (Hori et al., 2017); 3 cm (Xu et al., 2016) and 5 cm for 25 km spatial resolution (Liu et al., 2018); and 2 cm for 0.75° grid resolution (Brown and Derksen, 2013). Due to these significant disagreements in the depth thresholds, Gascoin et al. (2019) conducted a sensitivity experiment that tested the agreement between in-situ measurements and optical snow cover area products. The sensitivity of passive microwave snow cover identification results to snow depth at 6.25 km spatial resolution was also tested by computing the accuracy metrics with snow depth increasing from 0 to 10 cm.

Then, we needed to determine the threshold for converting fractional snow cover to binary snow cover. To date, few studies exist on fractional snow cover from the passive microwave pixel-level. Dai et al. (2017) considered snow cover on the grid if fractional snow cover (25 km) was larger than 10%. If the fraction of snow cover was less than 0.25, the SWE was set to 0 mm to correct the snow cover area in the daily SWE product based on Luojus et al. (2018). However, optical remote sensing studies often adopted 0.5 as the threshold of fractional snow cover (Hall and Riggs, 2007). Sensitivity experiments of fractional snow cover similar to snow depth were conducted to obtain the optimum conversion threshold. Both sensitivity experiments were carried out using 2017 bare land type datasets in Section 4.4.

**3.7 Performance accuracy assessment**

When evaluating the estimation performance of fractional snow cover in Section 4.1-4.3, we used conventional accuracy metrics; correlation coefficient (R; Eq. 4), mean absolute error (MAE; Eq. 5) and root mean squire error (RMSE; Eq. 6).Where $\overline{x}$ is the mean value of all predicted values $x_i$; $\overline{y}$ is the mean value of all target values, $y_i$; and n denotes the number of used data.

$$R = \frac{\sum_{i=1}^{n}(x_i - \overline{x})\,(y_i - \overline{y})}{\sqrt{\sum_{i=1}^{n}(x_i - \overline{x})^2 \sum_{i=1}^{n}(y_i - \overline{y})^2}} \tag{4}$$

$$MAE = \frac{1}{n}\sum_{i=1}^{n}|x_i - y_i| \tag{5}$$

$$RMSE = \sqrt{\frac{1}{n}\sum_{i=1}^{n}(x_i - y_i)^2} \tag{6}$$

We evaluated the predicted accuracy of fractional snow cover and assessed snow cover identification performance (see Section 4.4). Six accuracy assessment indices were used for the analysis of snow cover detection capability (Liu et al., 2018; Gascoin et al., 2019); overall accuracy (OA), precision (that is, a positive prediction value), recall, specificity (that is, the true negative rate), F1 score (Zhong et al., 2019), and Cohen's kappa coefficient (Foody, 2020). OA refers to the proportion of correctly classified pixels as snow-covered and snow-free. The F1 score is a weighted average measurement of precision and recall ranging from 0 to 1 to measure the accuracy of binary classification. Cohen's kappa coefficient measures the agreement between snow cover products and ground measurements. All these indices were calculated from the confusion matrix (Table 3). OE is the omission error; CE is the commission error.

# 4. Results analysis

## 4.1 Sensitivity in the training sample

### 4.1.1 Influence of input variables on model performance

We evaluated the results from 24 random forest fractional snow cover retrieval models (four types * six scenarios) to better understand which input variables have a good relationship with fractional snow cover and the combination of the variables that can improve retrieval model performance. The data used for variable sensitivity tests in this part spanned only two years (2014 – 2015) as the 91 GHz H polarization data was missing over the area south of 50° N for 2016 – 2017. The OOB error and 10-fold cross-validations error were used to measure the performance of fractional snow cover retrieval models in each scenario (Mutanga et al., 2012). Table 4 shows the results of the six scenarios for the bare land type datasets.

The variable selection tests were used to identify a better combination of different variables (Table 4). Scenario -3, which only involves V polarization data, yielded the smallest R (0.590) and the largest MAE (0.197) and RMSE (0.248) of OOB error, and also for 10-fold cross validation error (R: 0.596; MAE: 0.197; RMSE: 0.246). Scenario 3 performed the poorest of the six scenarios, which may be due to the lack of further available information from input variables that could be fully exploited (Xiao et al., 2018). Scenario -2, only containing passive microwave brightness temperature data similar to the variables used in Xu et al.(2016), had the second poorest performance. This shows that location information and topographic factors play a crucial role in snowpack distribution (Revuelto et al., 2014; Czyzowska-Wisniewski et al., 2015; Sturm et al., 2010). In this study, the retrieval method required these five basic input variables as auxiliary information in order to learn the characteristics of snow cover under different surface conditions to assist in accurately estimating snow cover properties. In contrast, in the absence of these basic input variables, the established model has no advantage in accurately predicting the characteristics of fractional snow cover under complex surface conditions. The major difference between Scenarios -1, 4, 5, 6 and Scenario -2 and 3 (Table 1) was whether or not considering the basic input variables (location information and topographic factors). The comparison results (i.e., Scenarios -1, 4, 5, 6 vs. Scenarios -2 and 3) indicate that the effect of location information and topography need to be considered to estimate snow parameters. Moreover, when compared to Scenarios-1, 4, 5, the setting in Scenario-6, where input variables were selected by importance, had the third poorest performance, with a low R, and a high MAE and RMSE. Scenarios -1, 4 and 5, generated better results; there were no significant differences in R, MAE and RMSE for the tests on the four land cover types (Table 4; Tables S4 and S5 in Appendix). The comparison among Scenarios -1, 4, 5 indirectly indicates that the variables used in Scenario -1 may have some information redundancy and slightly weaken the efficiency of the random forest retrieval model. Although the selection methods of Scenario -4 and 5 performed well (in terms of modeling time and accuracy of predicted target), only one scenario was selected; the other may be used as an alternative in future. To this end, the variable combinations in Scenario -5 were selected for further analysis.

**4.1.2 Determination of sample size**

The datasets from 2014 – 2016 were used to examine sensitivity to training sample size where the accuracy metrics used were the same as Section 4.1.1. As the values and variation trends of the accuracy metrics of the OOB error and 10-fold cross validation error were almost equivalent, only the OOB error is shown in Fig. 4 and Figs. S-1, 2, 3 (in the Appendix). We compared the performance of the random forest-based models altering the training sample size for the four land cover types.

Fig. 4 illustrates that there is a slow increase in R and a slight decrease in MAE and RMSE when the training sample size increased from 0.15% to 0.25% on the shrub type, whilst there was a significant increase in modeling time. As the sample size increases from 0.25% to 0.35%, the model consistently estimates fractional snow cover accurately (higher R and lower MAE and RMSE). This finding appears to be consistent with previous studies (Colditz, 2015; Nguyen et al., 2018). An applicable and eligible sample selection scheme, which can achieve an acceptable target prediction accuracy and an adequate execution time, is essential for the implementation of a random forest model with superior predictive capability. One noticeable distinction between the three sample sizes (0.25% ~ 0.35%) was modeling time. Interestingly, the 0.3% training sample size had the shortest modeling time of the three sample size (Fig. 4); Figs. S-1, 2, 3 also exhibit similar findings on modeling time. The reasons underpinning the difference in modeling time is beyond the scope of this study and requires further research. We used the sample dataset covering 0.30% of the study area of each class as a suitable size to randomly select training samples. We subsequently extracted the training samples for each land cover type from the 2011 – 2016 dataset to establish the retrieval models.

**4.2 Comparison of the four retrieval methods**

In this section, the independent testing datasets from 2010 were used to assess the predictive performance of the random forest-based models and the other three models (based on linear regression, ANN and MARS). A comparison of the modeling time for the four methods (Table 5) showed that linear regression had the shortest time, with less than 1 s for the four land cover types, followed by ANN with approximately 51 s (forest), 22 s (shrub), 156 s (prairie) and 35 s (bare land). Random forest modeling times were very close to ANN modeling times for each land cover type. In contrast, MARS was the most time consuming, with the longest time (approximately 6.5 h) for the prairie type and the shortest (19 min) for the shrub type. The absolute value of modeling time would vary under different computing capabilities.

Table 5 and Fig. 5 present the results of the four retrieval methods for the four land cover types. The retrieval models of the shrub type predominantly have the lowest RMSE in contrast with the other three land cover types using the four methods (Table 5; cf. Fig. 5 and Figs. S-4, 5, 6 in the Appendix). The random forest model had the highest R (0.916), lowest MAE (0.202) and RMSE (0.245), and no out-of-range records for the forest type (Table 5; Fig. 5 and 6). The distribution and variation of MAE and RMSE for the four methods were similar under different land cover types, with the exception of the shrub type (Table 5; Fig. 5). With the exception of ANN, the ranking of the three algorithms based on the accuracy of results (MAE and

RMSE) for the shrub test data was also the same as under other land cover types (i.e., Random forest > MARS > linear regression). Random forest had the greatest R value, followed by ANN, then MARS, and for most of the land cover types the smallest R value was from the linear regression. Fig. 6 illustrates that the random forest (Fig. 6D) produced a relatively small number of overestimated (~0) and underestimated (~1) values compared with the other three models (Fig. 6A – 6C). The MAE (0.315) and RMSE (0.401) of ANN were greater than MARS (MAE = 0.208, RMSE = 0.254). The number of out-of-range estimated values of ANN (36.62%; 161260) was also greater than MARS (2.65%; 11667), which may be attributed to a major underestimation of the fractional snow cover using the ANN method. The maximum and minimum of ANN and MARS on the forest type were 0.949 (-0.52) and 2.132 (-0.122), respectively. For the other three land cover types, the number of out-of-range pixels of the four methods were almost in the same order (random forest < ANN < MARS < linear regression).

The random forest-based models had the best performance with the highest R, and lowest MAE and RMSE (Table 5). Previous studies have generally neglected the analysis and evaluation of whether the estimated value is out-of-range (Liang et al., 2017; Wang et al., 2017; Hao et al., 2019; Masson et al., 2018). Table 5 show that the random forest models for the four land cover types produced reasonable fractional snow cover values ranging between 0 and 1. In comparison, the estimated fractional snow cover from the other three methods (linear regression, ANN and MARS) was beyond this range. From the number of out-of-range records, the linear regression method generated the largest number of out-of-range fractional snow cover estimates, with more than 0.85 million pixels (18.69%). Although the number of out-of-range records of ANN (12.31%) was less than MARS (16.39%), both numbers exceed 0.5 million. Kuter et al. (2018) estimated fractional snow cover using MARS and ANN techniques, also yielded similar out-of-range values. The linear regression method had the poorest performance in estimating fractional snow cover from passive microwave data, with the lowest R and the largest number of out-of-range records. These results indicate that nonlinear methods should first be used. Xiao et al. (2018) demonstrated the nonlinear relationship between passive microwave brightness and snow depth. De Lannoy et al. (2012) provided an exponential function for converting SWE to fractional snow cover. Thus, it is reasonable that a non-linear-relationship exists between fractional snow cover and passive microwave brightness temperature.

### 4.3 Evaluation of fractional snow cover

In this part, we analyzed the results from the training and evaluation stage for four land cover types (Table 6, Fig. 7 and Fig. 8). The independent data, which was randomly selected from the data in January and February (2008, 2009, and 2017) and the selecting rule is same as the training sample, was used to further evaluate the predictive capability of random forest models in all range values. Another independent data in December (2007, 2008, 2016) was selected to examine the predictive capability of the established retrieval models in fractional snow cover to other month. To avoid the influence of snow physical properties evolution in the evaluation tests, we only considered December (Xiao et al., 2018).

We first saw the evaluation results in January and February (Evaluation-1 in Table 6). For forest type, Fig. 7A and 7a

show that fractional snow cover around 1 are distinctly underestimated and few are above the 1:1 line. The model for forest type had the poorest performance with the lowest R (0.636) and the highest RMSE (0.221) for the evaluation dataset (Table 6). The retrieval model on the prairie type had the best performance for the evaluation data (R: 0.752; MAE: 0.148; RMSE: 0.189). In shrub and bare land types (Fig. 7B, 7b, 7D and 7d; Table 6), the retrieval models have similar performance in evaluation datasets (R: 0.712 and 0.719; MAE: 0.160 and 0.165; RMSE: 0.212 and 0.216, respectively). In the second evaluation experiments (Evaluation-2 in Table 6; Fig. 8), the best performance in predicted fractional snow cover is over prairie and the relatively large underestimation can be seen over forest (MAE and RMSE). Meanwhile, we do not see striking differences between these two evaluation experiments (Evaulation-1 vs. Evaluation-2) with respect to their RMSE values. The difference of the used evaluation sample can explain the slight diversity in statistics metrics (R). Most of "true" fractional snow cover values in the training and validation datasets were distributed at two polar ends (0.0~0.3 and 0.9~1.0) in these two land cover types. When comes to the results in the training stage and the evaluation stage, we found that the estimation performance of the retrieval model in evaluation datasets are highly dependent on the quality of training sample which was used to establish the retrieval models. Fig. 7 and Fig. 8 show that the established models in four land cover types can properly capture the characteristics of all range of fractional snow cover values.

Apart from the scatter plots and statistical analysis, Fig. 9 shows the distribution pattern of snow cover from a spatial perspective on February 27th, 2017, including MODIS composite binary snow cover (Fig. 9A), MODIS fractional snow cover (Fig. 9B), and the estimated fractional snow cover by the proposed algorithm (Fig. 9C). When the most rigorous pixel filtering rule at the 15*15 pixels window was applied (see Section 3.2), the large number of clouds covered pixels (yellow) in Fig. 9A resulted in most areas of the MODIS fractional snow cover image (Fig. 9B) being represented by a "fill value". Additionally, the number of intermediate values for MODIS fractional snow cover in winter would be much lower than the number of values near the two extreme values (0 and 1). In contrast, the estimated fractional snow cover from passive microwave brightness temperature data can provide almost complete coverage and continuous spatial information on snow cover (Fig. 9C; Fig. S-7 in the Appendix). Fig. 9 shows the comparison between our estimated fractional snow cover and the reference MODIS fractional snow cover, and more importantly, provides another perspective for snow cover identification in Section 4.4. Thus, Fig. 9B and 8C used the threshold 0.3 to define snow-covered and snow-free area, and this threshold was adopted through a series of experiments in Section 4.4. The threshold 0.5 was selected according to previous optical remote sensing study on fractional snow cover (Tran et al., 2019; Marchane et al., 2015). This means that the pixel was identified as snow cover when fractional snow cover value was greater than 0.3. From Fig. 9A – C, the spatial pattern of estimated fractional snow cover from the proposed method seems to accurately capture the distribution of snow cover from MODIS under clear-sky conditions, such as the snow-free area in most areas of North America, and snow-covered areas in northern Canada. Fig. 9D presents a specific example comparing these two fractional snow cover datasets and MODIS composite binary snow cover products in central

Canada on February 27th, 2017. We also provide another date comparison for MODIS and the estimated fractional snow cover on January 10th, 2017 in the Appendix. Based on this example, we find that our estimated fractional snow cover was capable of obtaining snow cover distribution when most of the area was covered by cloud, which was not the case for MODIS. This example also show that the extent of snowline observed in the MODIS binary snow cover image (500 m), which was the boundary between snow-covered and snow-free, was well described and exhibited by the estimated fractional snow cover (6.25 km).

Thus far, we have evaluated the performance of random forest-based models on independent datasets from 2008 – 2010 and 2017 on each land cover type. The results from the random forest (Table 5; Fig. 6D; Figs. S-4 – 6; Fig. 7) show that the minimum estimates were higher than 0 and approximately 0.01. As to this overestimation, one possible error source is the used inversion method, of which the final predicted outputs are obtained by averaging all results of the established sub decision trees (see Section 3.4.4) (Breiman, 2001; Belgiu and Drăguţ, 2016) and this overestimation can be found in other study applying random forest study (Wei et al., 2019). Additionally, the attenuation and scattering of passive microwave radiation are not only caused by the snowpack. The non-snow scatterer (e.g., precipitation, cold desert, frozen ground) may be the majority error source potentially contributing to the overestimation of snow cover area as these scatters were easily misclassified as snow cover in less snow cover conditions (Grody and Basist, 1996; Dai et al., 2017). A detailed analysis on the misclassification due to non-snow scatterer is provided in Section 4.4. Although MODIS snow cover products are highly accurate in snow cover identification (Tran et al., 2019), the estimated results indicate that a large number of fractional snow cover values were overestimated (~0) and underestimated (~1). Some fractional snow cover estimates, at the individual pixel level, show a large discrete distribution near the 1:1 line (Fig. 7). These misestimates are not confined to the results of the random forest model, but also appear in results of the other three methods (Fig. 6; Fig. S-4 – 6 in the Appendix). Satellite sensors may provide completely different snow cover information because of different satellite overpass time. In this study, the difference in the equator crossing time between MODIS and passive microwave sensor was close to 6.5 and 9.5 h (refer to Section 2.1 and 2.3.1). Generally, the error caused by the differences in the satellite overpass time may easily be neglected when using multi-sensor observations for data fusion.

**4.4 Validation using ground measurements**

In winter with clouds and snow cover, the MCTD10A1 data still contained a large number of clouds (Fig. 9A; yellow) despite the implementation of the cloud removal and filling process for MODIS snow cover data. When applying the rigorous pixel filters (see Section 3.2), there was very little snow cover data for further model training and results analysis in one imagery (Fig. 9B). To evaluate and validate the estimated fractional snow cover in the absence of reference MODIS fractional snow cover we conducted further analysis on snow cover detection capability. The ground snow depth measurements were utilized to investigate the accuracy of snow cover identification from two snow cover data; the snow cover detected by Grody's

algorithm and fractional snow cover derived from random forest. We collected all available meteorological station snow depth measurements of 2008 – 2009 and 2017 (January and February) over North America, obtaining more than 900 000 pairs of records. This includes the snow depth measurements, snow cover area converted from the estimated fractional snow cover (hereafter referred to random forest SCA), and snow cover area derived from Grody's snow cover mapping algorithm (hereafter referred to Grody's algorithm SCA).

The sensitivity to ground-based snow depth in the snow cover detection results were tested by computing the accuracy metrics using data in 2017. Fig. 10 shows that the accuracy metrics vary with increasing snow depth, whereby the metrics change significantly when snow depth exceeds 2 c, and reach a relative optimum when snow depth is equal to 2 cm. Che et al. (2008) stated that snow cover may be detected by passive microwave sensors when snow depth is greater than 2 cm. For this reason, we adopted 2 cm as the optimum depth threshold to transform ground snow depth measurements to snow-covered or snow-free information. We also conducted a series of sensitivity experiments to search the optimum threshold for converting fractional snow cover to binary snow cover (Fig. 11). Fig. 11 shows that recall and precision have opposing variation trends; the F1-score is up to the maximum value when FSC = 0.3. In addition, the other two indicators (OA, kappa) also reached their maximum when the FSC value were ranging between 0.3 and 0.4. As expected, 0.3 was used as the conversion threshold for fractional snow cover. Nevertheless, the conversion thresholds of snow depth and fractional snow cover need to be optimized with more data in the future.

We used a 2 cm snow depth threshold and a 0.3 fractional snow cover threshold to calculate the confusion matrix for Grody's algorithm SCA and random forest SCA against ground snow depth measurements in 2008-2009 and 2017 (Fig. 12 and Fig. S-9). Fig. 12 illustrates that the overall accuracy of snow cover identification had significantly improved by 20%, from 0.71 for Grody's algorithm SCA to 0.85 for random forest SCA, indicating that the latter's results were in good agreement with ground snow cover measurements (kappa = 0.65). For the random forest SCA, precision (0.84) was lower than the recall (0.86), which means that snow cover area was more likely to be overestimated (CE = 0.16) than underestimated (OE = 0.14), with respect to in-situ measurements. In contrast, for Grody's algorithm SCA, precision (0.87) was larger than the recall (0.51). By utilizing the proposed method, the OE of snow cover identification had reduced by 71% compared to the OE for Grody's algorithm SCA. The snow cover identification accuracy for the four land cover types are illustrated in Fig. S-9 (in the Appendix) by radar charts.

We subsequently explored the influence of non-snow scatterers in estimating fractional snow cover. The CE of Grody's algorithm (CE = 0.13) was lower than random forest SCA (CE = 0.16). Fig. 12 shows that the overall snow-free identification capability of Grody's algorithm SCA (specificity = 0.91) was significantly superior to the random forest SCA (specificity = 0.81), which was also apparent for the four land cover types (Fig. S-9). This may possibly be due to Grody's algorithm filtering out non-snow scatter signature (precipitation, cold desert, and frozen ground) (Grody and Basist, 1996). We counted the

number of records in which a pixel had been detected as snow-free by the station and Grody's algorithm, however, was considered snow-covered by random forest SCA. The records, which were misclassified as snow cover by random forest SCA, although they are non-snow scatter components (precipitation, cold desert, and frozen ground), account for 72.6% of total misclassification records (CE = 0.16), of which 64.3% comes from precipitation, 7.0% from cold desert, and 0.9% from frozen ground. For forest, shrub, prairie and, bare land types, this misclassification proportion because of the non-snow scatters were 76.0%, 92.3%, 68.6% and 65.4%, respectively. These results demonstrate that the non-snow scatterer is the major source of snow cover misclassification for random forest FSC results (Grody and Basist, 1996). Therefore, it is necessary to first distinguish the scattering signature of snow cover from other non-snow scattering signatures when using passive microwave data to identify snow cover. Similar preprocessing has been applied into snow depth estimation to minimize its uncertainties (Xiao et al., 2018; Wang et al., 2019; Tedesco and Jeyaratnam, 2016). An example is given to illustrate the inconsistency and consistency results of snow cover mapping between the random forest SCA and Grody's algorithm SCA (Fig. 13). Fig. 13 shows that the most areas of North America have consistent snow cover mapping results: Code A (cyan) and B (green) regions. Referring to the continue fractional snow cover image (Fig. S-7b in the Appendix), the inconsistent in monitoring snow cover extent always occur in mid-latitude regions with the value of fractional snow cover around 0.5. According to our experimental results analysis, the mid-value of fractional snow cover regions is one source of the omit error for Grody's algorithm SCA.

## 5. Discussions

### 5.1 Sensitivity to training sample size and quality

The size and quality of training samples may contribute to a large error at the individual pixel level (Dobreva and Klein, 2011; Kuter et al., 2018; Nguyen et al., 2018; Belgiu and Drăguţ, 2016). Previous studies have investigated the sensitivity to sample size and sample quality (Nguyen et al., 2018; Colditz, 2015; Lyons et al., 2018). While some studies indicate that a larger training sample size improves the accuracy of estimates, we found that a training sample dataset covering about 0.3% of the total study area was sufficient to achieve high accuracy in the estimation of fractional snow cover. When comparing to previous sensitivity tests on sample size (Nguyen et al., 2018), the major difference was taking modeling time as an index in this study.

The estimation results of the random forest model (for the training, testing and evaluation datasets; Sections 4.2 and 4.3) showed that generally, the prediction performance of the random forest model was closely related to the quality of training sample. In this study's datasets, a greater number of records were located near the extreme values of the fractional snow cover (0 and 1). Thus, it is reasonable to use stratified random sampling (Dobreva and Klein, 2011), however, not the proportional distribution of target values suggested by previous studies (Nguyen et al., 2018; Millard and Richardson, 2015). Even in these cases, the overestimation and underestimation often occur in the results of training datasets (Fig. 7 A – D) and evaluation

datasets (Fig. 7 a – d), respectively. This is mainly because the established fractional snow cover retrieval model when using the training sample with relatively low diversity of fractional snow cover values does not well learn the snow cover distribution characteristics of the various surface condition. Therefore, it is necessary for future studies to increase the number of samples by extending the study period to the snow accumulation and snow ablation stages (Xiao et al., 2018), where there is much more shallow snow and "patchy" snow cover. Another option is using data from multi-source sensors to generate reference snow cover data (e.g., Sentinel -1 Synthetic Aperture Radar data). By doing this, the proportion of fractional snow cover values in the training sample may be distributed as evenly as possible (Colditz, 2015; Jin et al., 2014; Lyons et al., 2018).

## 5.2 Effects of vegetation

Snow cover detection can be partially or completely obscured (or intercepted) by dense vegetation canopies. This introduces major uncertainties in accurate detection of snow cover (Che et al., 2016; Hall et al., 2001; Parajka et al., 2012). Forest cover is an influential factor that cannot be ignored in optical and microwave remote sensing studies (Metsämäki et al., 2005; Cohen et al., 2015). It is evident that fractional snow cover retrievals typically have the least accuracy under the forest type in comparison to other land cover types (Table 5; Figs. 6 and 7). There are two reasons that may be attribute to this error; one is the accuracy of the reference "true" fractional snow cover data in a forested area (Riggs and Hall, 2016), and the other is the microwave radiation attenuation caused by forests (Che et al., 2016).

Previous studies have reported that lower accuracy of MODIS snow cover products was found in forest covered areas and complex terrain (Hall and Riggs, 2007; Tran et al., 2019; Coll and Li, 2018). Several studies have validated and evaluated the accuracy of MODIS snow cover products, particularly in forested areas (Parajka et al., 2012; Zhang et al., 2019; Arsenault et al., 2014; Kostadinov and Lookingbill, 2015). In term of the NDSI threshold in forested areas (Section 3.2), we used 0.4 as a conservative threshold. According to previous studies, our operation (merely using NDSI as the criterion) in forest-covered areas would produce greater commission errors compared with using the Normalized Difference Vegetation Index (NDVI) as auxiliary information (Hall and Riggs, 2007). The retrieval results indicated that the NDSI threshold in forested areas needs optimization using numerous data (Riggs et al., 2017; Xin et al., 2012). In addition to the influence of forests on MODIS data, forests also hamper the upwelling microwave radiation emitted from the ground. Snow cover in forested areas may be divided into under and above forest canopy snow cover (Xin et al., 2012). This apparently distinguishes the interference effects of evergreen forests and deciduous forests on snow cover (Gascoin et al., 2019; Romanov and Tarpley, 2007). Additionally, there are major differences in the observation means for optical and passive microwave sensors in forested areas. The capacity for optical sensors to observe above forest canopy snow cover is mainly dependent on the vegetation canopy density (Kuter et al., 2018), while microwave sensor may obtain information on snow cover under vegetation canopy (under forest canopy snow cover) (Che et al., 2016; Cohen et al., 2015). Overall, the combination of these two effects may produce low estimation accuracy for fractional snow cover.

## 6. **Conclusions**

Many previous studies have focused on estimating fractional snow cover utilizing optical remote sensing imagery, which suffers from cloud contamination during data acquisition. In contrast, microwave sensors offer attractive advantages of working under all weather conditions. In this study, we attempted to develop an algorithm for applying the enhanced-resolution passive microwave brightness temperature data (6.25 km) to fractional snow cover estimation during January and February of 2010 to 2017. Using the reference fractional snow cover stem from MODIS snow cover products as the "true" observation, we established fractional snow cover retrieval models for four land cover types (forest, shrub, prairie and bare land) inputting 12 variables selected by 24 sensitivity experiments. The proposed algorithm accounted for a series of influential factors including topography, land cover, and location information. Compared with the other three methods (linear regression, ANN and MARS), the random forest-based algorithm had the best performance with high accuracy (highest R, and lowest MAE and RMSE) and no out-of-range retrievals. The results of the evaluation using the reference fractional snow cover data in 2017 showed that our proposed algorithm had good retrieval performance in estimating fractional snow cover, with RMSEs ranging from 0.167 to 0.198. Moreover, in-situ snow depth measurements were used to validate the accuracy of the proposed fractional snow cover estimation algorithm in snow cover mapping. Snow cover detection capability of the random forest-based method was superior (OA =0.85, kappa = 0.65) to that of Grody's algorithm, with overall accuracy increasing by 20% (from 0.71 to 0.85), and omission error reducing by 71% (from 0.48 to 0.14), when the fractional snow cover threshold was 0.3. Although the random forest-based models achieved an acceptable accuracy, the fractional snow cover was more likely to be overestimated (CE = 0.16) than underestimated (OE = 0.14). In addition, the effect of the non-snow scatterer was evaluated on fractional snow cover predictions by means of the good prediction of Grody's algorithm on snow-free class; the results indicated that more than 70% of CE was caused by misclassifying the non-snow scatterer (precipitation, cold desert, frozen ground) as snow cover. These models established using several data sources in January and February had better applicability in dry snow conditions, while estimation results could be less accurate in wet snow conditions.

Numerous studies have investigated the relationship between common snowpack physical properties (e.g., snow depth and water equivalent) and passive microwave brightness temperature at different frequencies and polarizations (Chang et al., 1987; Dietz et al., 2011; Kim et al., 2019; Xiao et al., 2018). Unlike many previous studies, this study innovatively used passive microwave data to directly estimate fractional snow cover. The results showed that it is possible to directly obtain an estimated fractional snow cover with high accuracy from high-spatial-resolution passive microwave data (6.25 km) under all weather conditions. Further detailed study on the use of high spatial resolution passive microwave data for fractional snow cover estimation presents itself as an interesting research direction for the development of the studies on fractional snow cover estimation. Furthermore, to reduce some of the limitations (e.g., forest effects) (Cohen et al., 2015) and deficiencies (overestimation and underestimation) identified in this study, future studies should pay greater attention to the prediction of

the fractional snow cover using passive microwave data. To the best of our knowledge, this study may represent the first attempt to establish a relationship between microwave brightness temperature and the reference "true" fractional snow cover using machine learning methods. However, it also contains significant limitations in understanding the physics that relates fractional snow cover to the signature of passive microwave brightness temperature (Cohen et al., 2015; Che et al., 2016).

Future studies need to use physical snowpack models and radiation transfer theory to explore the physical mechanistic relationships between microwave brightness temperature and fractional snow cover (Pan et al., 2014).

*Competing interests*: We declare that we have no competing interests.

*Acknowledgements*: This work was supported by the National Natural Science Foundation of China Grant (41771379) and

10 the National Key Research and Development Program of China Grant (2016YFA0600103).

**Appendix**

Supplementary material for this article is available in supplement.

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

List of Tables and Figures

Table 1. The input variables list. Line means this variable is not selected; asterisk indicated the variable is selected. The numbers in square brackets denote the number of variables. T19H is the brightness temperature (T) in 19-GHz channel with H polarization; T_19V_19H denotes the difference of brightness temperature between 19V and 19H channel; others are similar.

| ID | Elements | Scenario-1 [18] | Scenario-2 [13] | Scenario-3 [3] | Scenario-4 [11] | Scenario-5 [12] | Scenario-6 [9] |
|---|---|---|---|---|---|---|---|
| 1 | Latitude | * | - | - | * | * | * |
| 2 | Longitude | * | - | - | * | * | - |
| 3 | DEM | * | - | - | * | * | - |
| 4 | Slope | * | - | - | * | * | - |
| 5 | Aspect | * | - | - | * | * | - |
| 6 | T19H | * | * | - | * | - | - |
| 7 | T19V | * | * | * | * | - | - |
| 8 | T37H | * | * | - | * | - | * |
| 9 | T37V | * | * | * | * | - | * |
| 10 | T91H | * | * | - | * | - | * |
| 11 | T91V | * | * | * | * | - | * |
| 12 | T_19V_19H | * | * | - | - | * | - |
| 13 | T_19V_37V | * | * | - | - | * | * |
| 14 | T_19H_37H | * | * | - | - | * | - |
| 15 | T_22V_19V | * | * | - | - | * | * |
| 16 | T_22V_91V | * | * | - | - | * | * |
| 17 | T_37V_37H | * | * | - | - | * | - |
| 18 | T_37V_91V | * | * | - | - | * | * |
| References | | (Liu et al., 2018) | (Xu et al., 2016) | (Kim et al., 2018) | (Xiao et al., 2018) | | (Nguyen et al., 2018) |

Table 2. The description of the revised Grody's algorithm. The unit is Kelvin (K).

| Scattering Materials | Description |
|---|---|
| Scattering signature | $(Tb19V - Tb37V) > 0$ K |
| Precipitation | $(Tb22V \geq 259\ K)$ or $(254\ K \leq Tb22V \leq 258\ K\ and\ (Tb19V - Tb37V) \leq 2\ K)$ |
| Cold desert | $(Tb19V - Tb19H) \geq 18\ K$ and $(Tb19V - Tb37V) \leq 10\ K$ |
| Frozen ground | $(Tb19V - Tb19H) \geq 8\ K$ and $(Tb19V - Tb37V) \leq 2\ K$ |

Table 3. Confusion matrix defining the accuracy of the predicted snow cover map reference to the in-situ snow cover observation. The characters (TP, FP, FN, TN) represent the number of records of snow-covered or snow-free in a particular condition.

| | | Ground observation (true) | |
| --- | --- | --- | --- |
| | | snow-covered (Positive) | snow-free (Negative) |
| Prediction | snow-covered (Positive) | TP (true positive) | FP (false positive) |
| | snow-free (Negative) | FN (false negative) | TN (true negative) |

$OA = (TP + TN)/(TP + TN + FN + FP)$

$OE = FN/(TP + FN)$

$CE = FP/(TP + FP)$

$Pecision = TP/(TP + FP) = 1 - CE$

$Recall = TP/(TP + FN) = 1 - OE$

$Specificity = TN/(TN + FP)$

$F1 = (2 * Precision * Recall)/(Precision + Recall)$

Table 4. Variable selection tests in 6 scenarios on bare land type data for random forest method. The accuracy indexes of the estimation are calculated using OOB error estimates and 10-fold cross validation (CV).

| | Indexes | Scenario-1 | Scenario-2 | Scenario-3 | Scenario-4 | Scenario-5 | Scenario-6 |
| --- | --- | --- | --- | --- | --- | --- | --- |
| | R | 0.776 | 0.679 | 0.590 | 0.778 | 0.774 | 0.708 |
| OOB-error | MAE | 0.152 | 0.178 | 0.197 | 0.150 | 0.152 | 0.170 |
| | RMSE | 0.194 | 0.224 | 0.248 | 0.193 | 0.194 | 0.216 |
| | R | 0.777 | 0.682 | 0.596 | 0.778 | 0.775 | 0.710 |
| 10-fold CV | MAE | 0.152 | 0.178 | 0.197 | 0.151 | 0.153 | 0.170 |
| | RMSE | 0.193 | 0.223 | 0.246 | 0.193 | 0.194 | 0.215 |
| Time spent modeling / s | | 7.37 | 5.57 | 3.46 | 5.43 | 5.26 | 5.27 |

Table 5. Performance of linear regression, ANN, MARS and random forest model using test datasets from 2010 for four land cover types. FSC indicate fractional snow cover. The number outside parentheses indicate the number of pixels; The number inside brackets indicate their percentage.

| Method | Land cover type | Time spent modeling/ s | R | MAE | RMSE | Max. /Min. | FSC < 0 (%) | FSC > 1 (%) |
|---|---|---|---|---|---|---|---|---|
| Linear regression | Forest | 0.37 | 0.896 | 0.225 | 0.279 | 1.204 (-0.183) | 44978 (10.21) | 554 (0.13) |
| | Shrub | 0.24 | 0.956 | 0.174 | 0.198 | 1.605/-0.382 | 335 (0.06) | 125589 (24.17) |
| | Prairie | 0.49 | 0.902 | 0.179 | 0.215 | 1.524 /-0.331 | 23604 (0.87) | 632417 (23.22) |
| | Bare land | 0.29 | 0.892 | 0.177 | 0.213 | 1.647 /-0.087 | 912 (0.10) | 30208 (3.32) |
| ANN | Forest | 51.09 | 0.895 | 0.315 | 0.401 | 0.949 / -0.520 | 161260 (36.62) | 0 (0) |
| | Shrub | 21.73 | 0.966 | 0.103 | 0.146 | 1.251 / -0.327 | 15267 (2.94) | 38207 (7.35) |
| | Prairie | 156.29 | 0.916 | 0.197 | 0.23 | 1.527 / -0.166 | 743 (0.03) | 310285 (11.39) |
| | Bare land | 35.31 | 0.932 | 0.174 | 0.203 | 1.730 / 0.173 | 0 (0) | 39491 (4.34) |
| MARS | Forest | 2518.10 | 0.838 | 0.208 | 0.254 | 2.132 / -0.122 | 8844 (2.01) | 2823 (0.64) |
| | Shrub | 1127.24 | 0.926 | 0.149 | 0.185 | 2.053 / -0.239 | 2977 (0.57) | 121693 (23.42) |
| | Prairie | 23406.76 | 0.912 | 0.164 | 0.197 | 1.764 / -0.733 | 4371 (0.16) | 469416 (17.24) |
| | Bare land | 2518.10 | 0.911 | 0.156 | 0.191 | 2.253 / -0.844 | 469 (0.05) | 142155 (15.62) |
| Random Forest | Forest | 52.16 | 0.916 | 0.202 | 0.245 | 0.960 / 0.011 | 0 (0) | 0 (0) |
| | Shrub | 16.76 | 0.975 | 0.118 | 0.162 | 0.999 /0.023 | 0 (0) | 0 (0) |
| | Prairie | 214.06 | 0.955 | 0.134 | 0.173 | 1.000 / 0.011 | 0 (0) | 0 (0) |
| | Bare land | 38.73 | 0.967 | 0.103 | 0.148 | 0.998 / 0.027 | 0 (0) | 0 (0) |

5  Table 6. The performance of random forest models on training, and evaluation data over four land cover types. Evaluaiton-1 is used to evaluated the estimation performance of the established retrieval models on fractional snow cover in January and February (Fig). Evaluation-2 is used to analyze the prediction performance of the retrieval models in December.

| Land cover type | Training | | | Evaluation-1 | | | Evaluation-2 | | |
|---|---|---|---|---|---|---|---|---|---|
| | R | MAE | RMSE | R | MAE | RMSE | R | MAE | RMSE |
| Forest | 0.702 | 0.166 | 0.207 | 0.636 | 0.180 | 0.221 | 0.658 | 0.180 | 0.222 |
| Shrub | 0.772 | 0.146 | 0.191 | 0.712 | 0.160 | 0.212 | 0.643 | 0.167 | 0.223 |
| Prairie | 0.807 | 0.142 | 0.182 | 0.752 | 0.148 | 0.189 | 0.762 | 0.166 | 0.213 |
| Bare land | 0.807 | 0.144 | 0.190 | 0.719 | 0.165 | 0.216 | 0.744 | 0.162 | 0.217 |

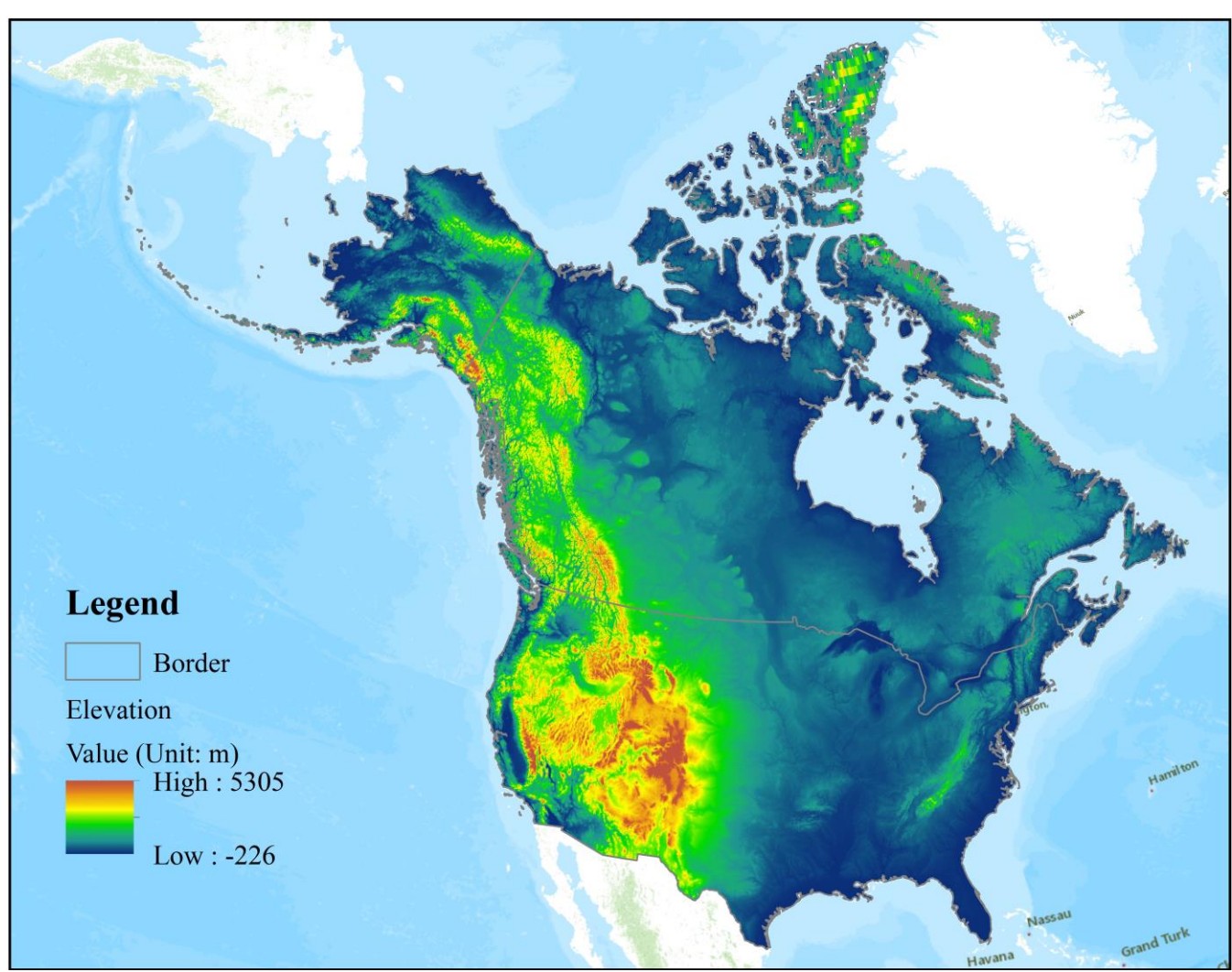

Fig. 1 Topographic map of North America.

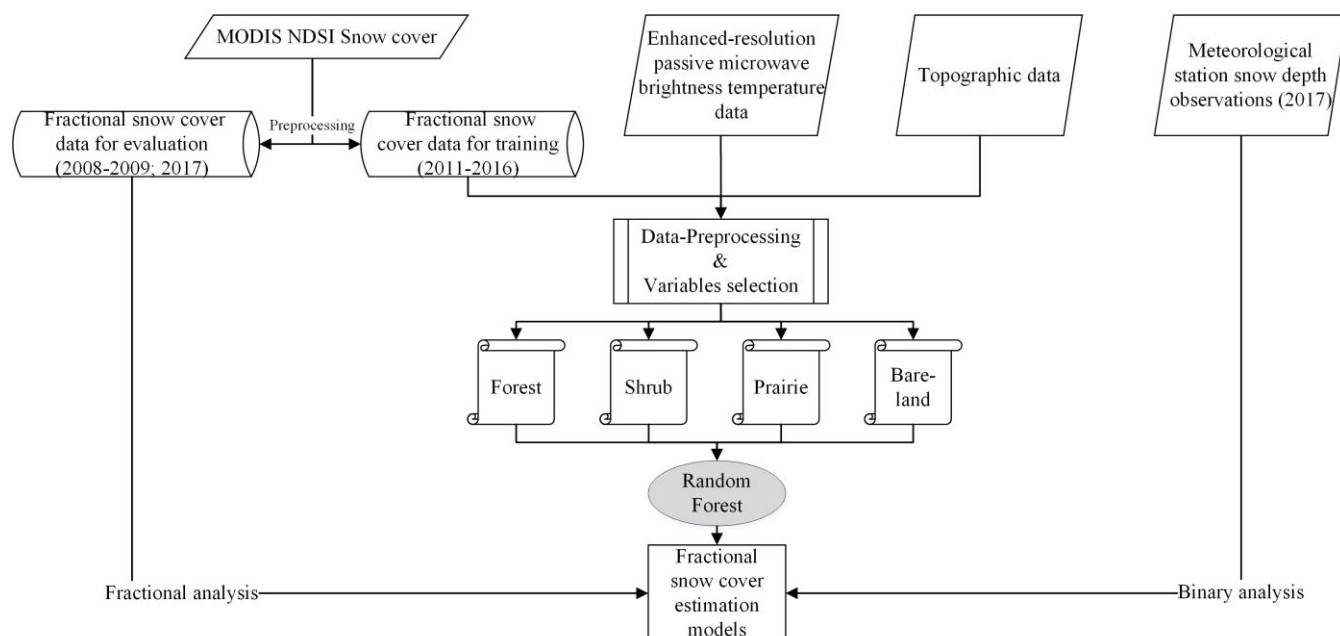

Fig. 2. Workflow diagram illustrating the processing of fractional snow cover retrieval.

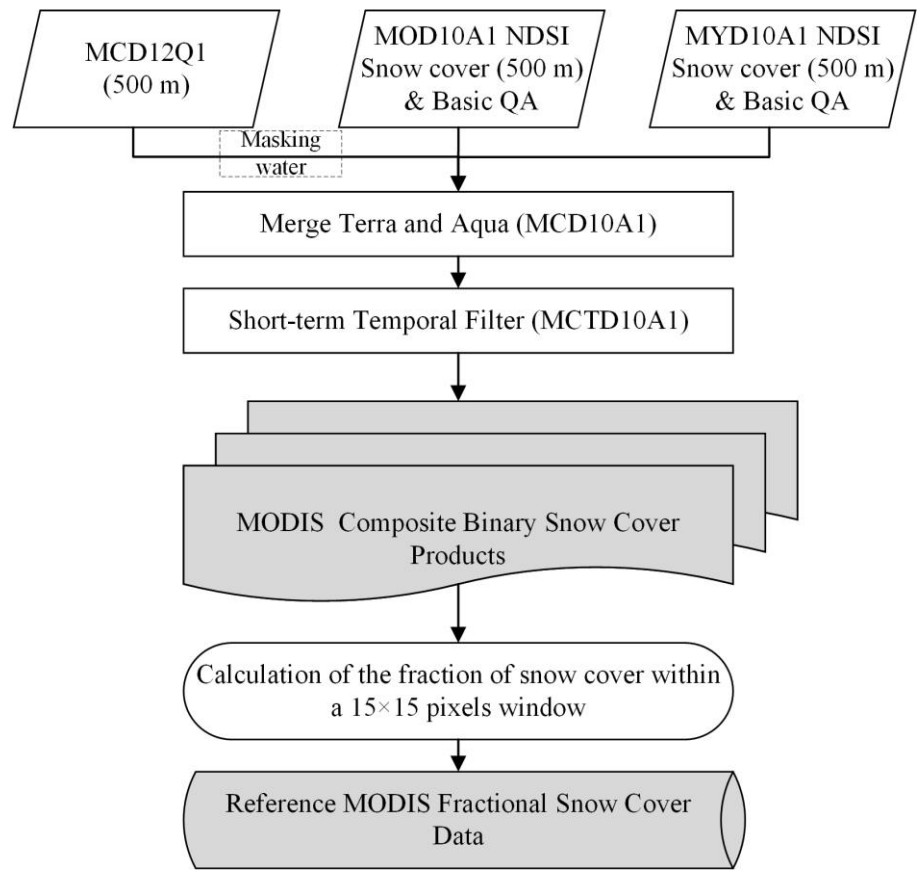

Fig. 3. The generation of MODIS fractional snow cover

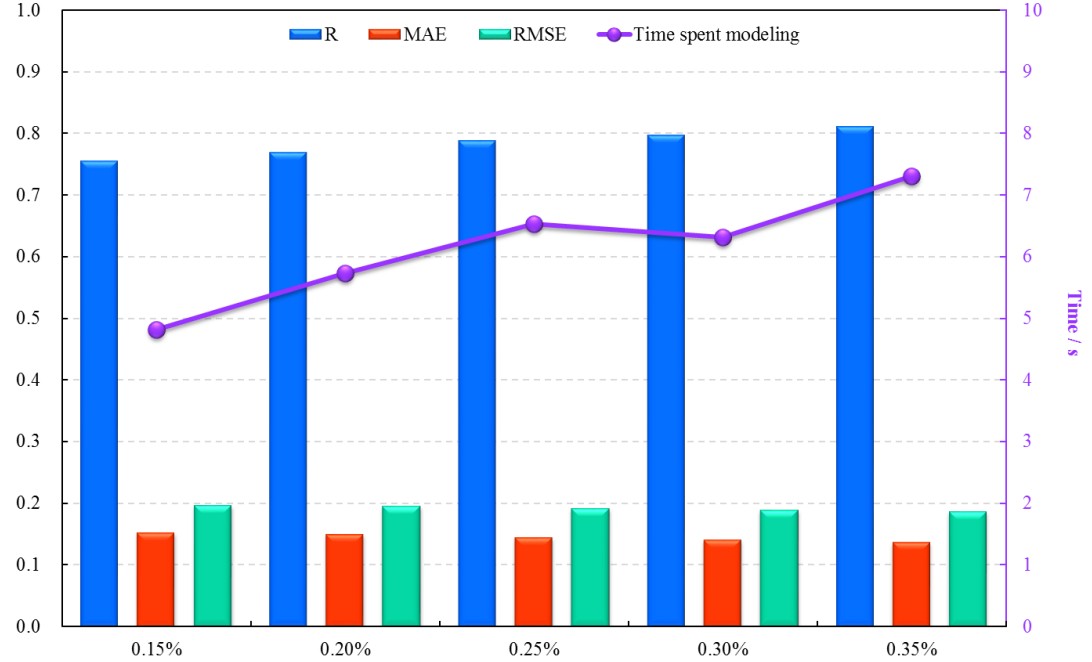

Fig. 4. The performance of random forest models with increasing the size of training sample for shrub type

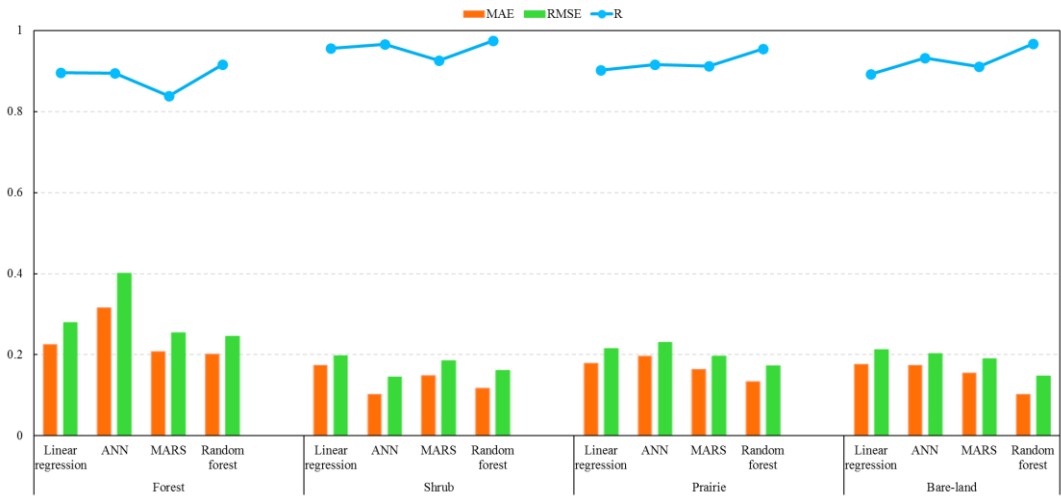

5    Fig. 5. The variation of the accuracy indexes (MAE, RMSE and R) on four algorithms (linear regression, ANN, MARS and Random forest) for four land cover.

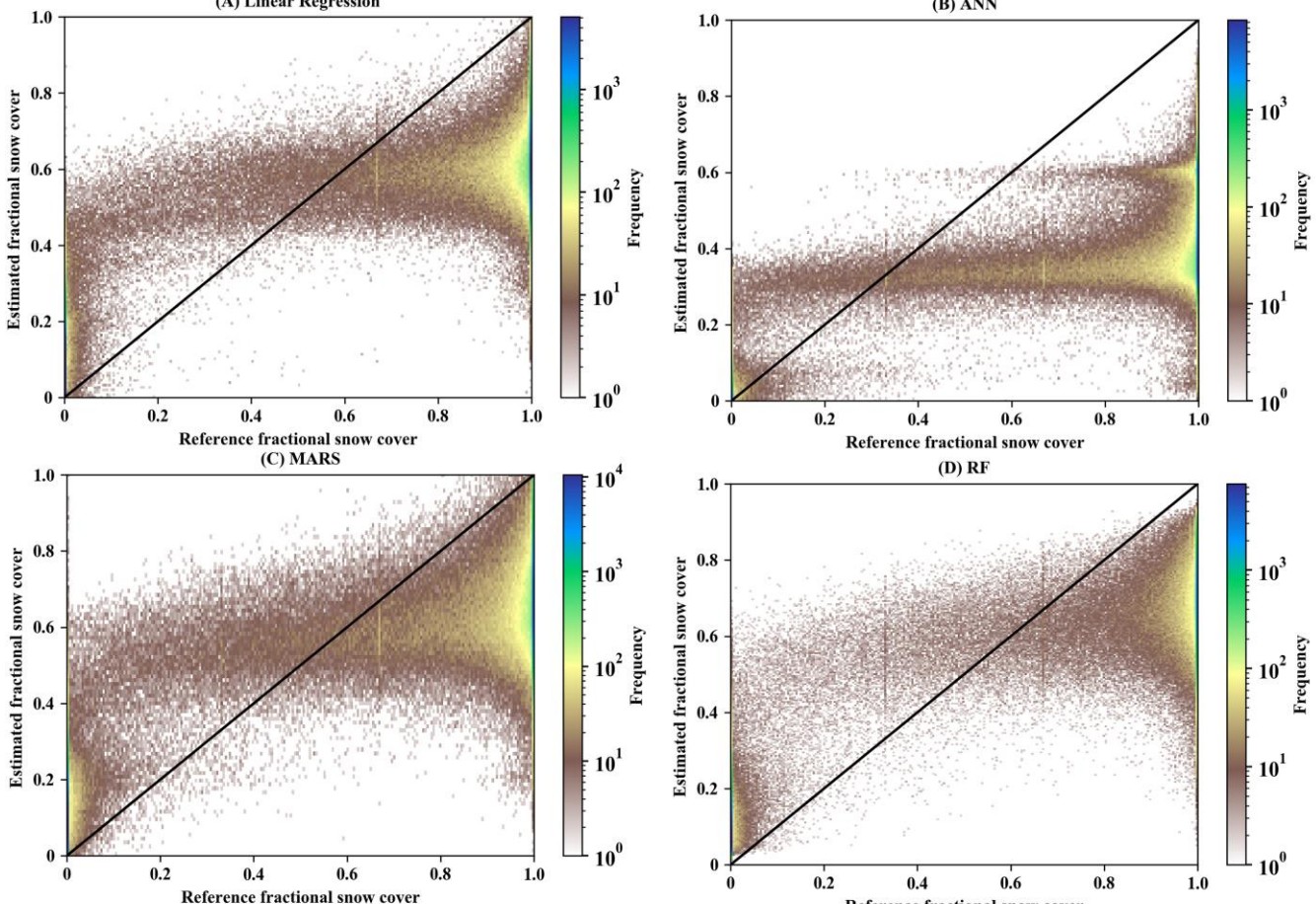

Fig. 6. The color-density scatter plots between the estimated fractional snow cover and MODIS-derived fractional snow cover for four algorithms (linear regression, ANN, MARS, and random forest) for forest type. The accuracy metric refers to Table 5. [Note: out-of-range fractional snow cover values of linear regression, ANN and MARS were truncated on 0 and 1]. Noted that: the testing sample used the entire records of 2010.

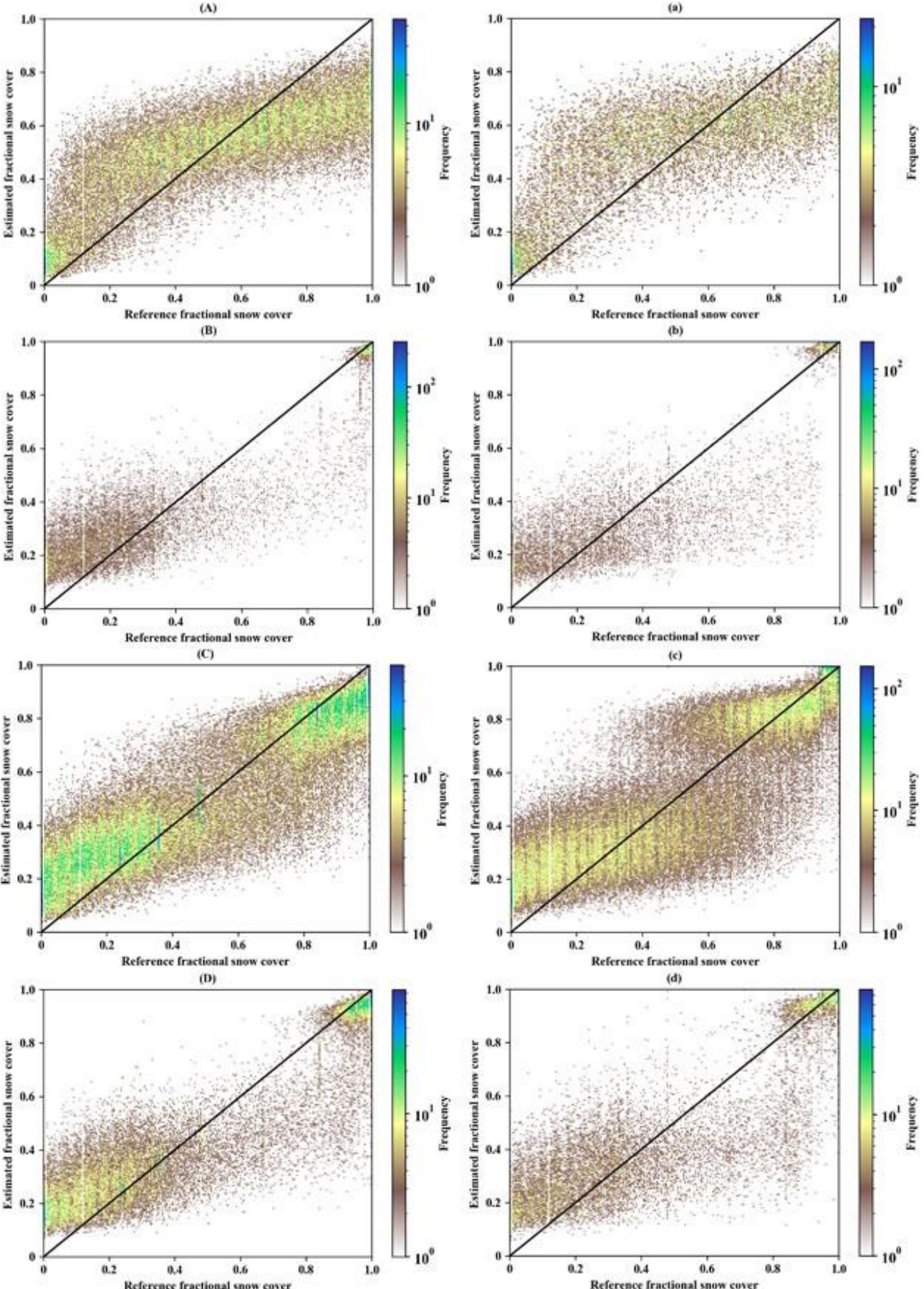

Fig. 7. The color-density scatter plots between the estimated fractional snow cover and MODIS-derived fractional snow cover in January and February for four land cover types (forest: A, a; shrub: B, b; prairie: C, c; bare land: D, d). Left column

with capital letters is the results in the training stage (A-D); right column with lowercase letters is the results in the evaluation stage (a-d). Statistics metrics refer to Table 6: Training and Evaluation-1. Noted that: the sample selection rule of evaluation data is same as the training data.

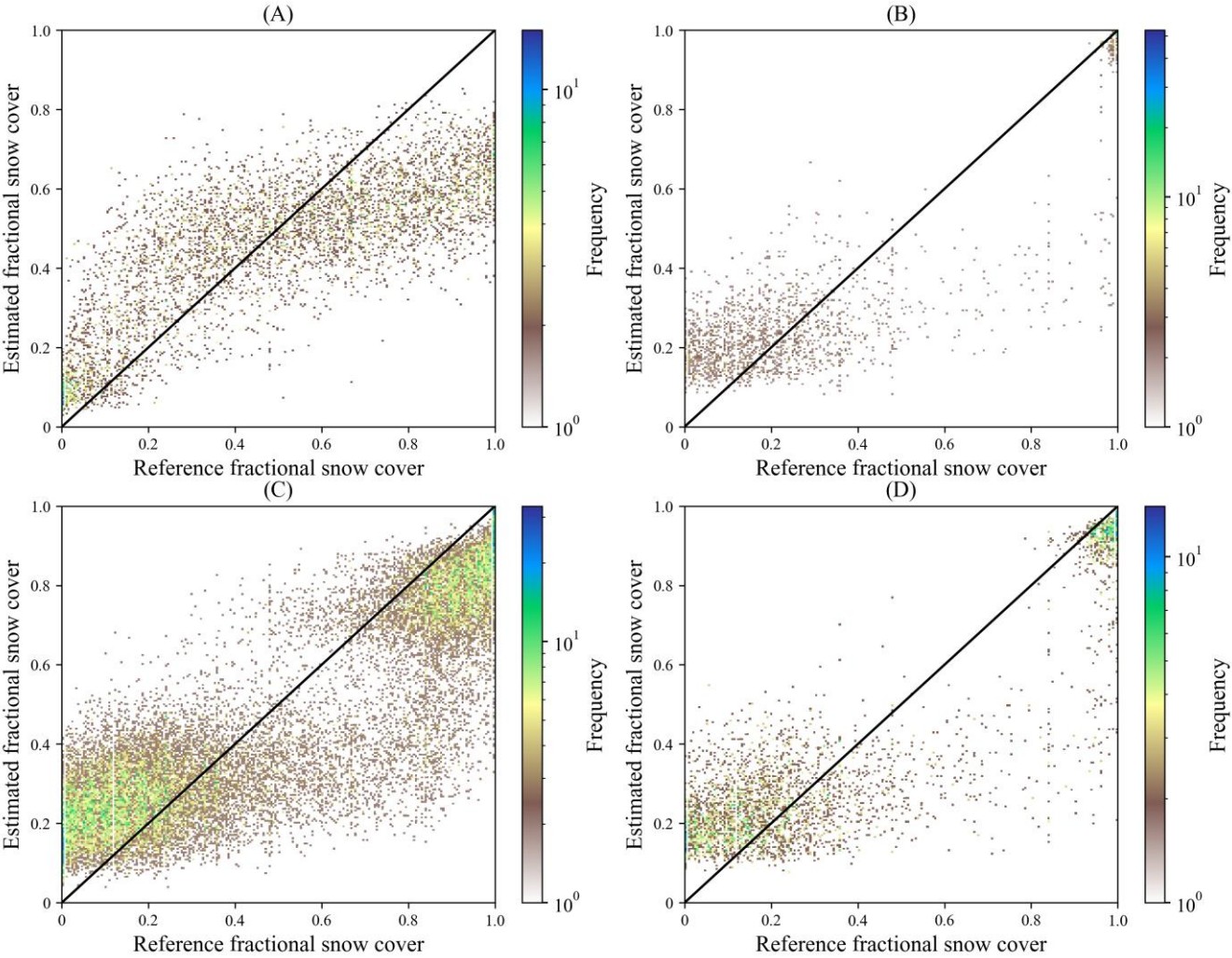

Fig. 8 The color-density scatter plots between the estimated fractional snow cover and MODIS-derived fractional snow cover in December for four land cover types: forest (A), shrub (B), prairie (C), bare land (D). Statistics metrics refer to Table 6: Evaluation-2. Noted that: the sample selection rule of evaluation data is same as the training data.

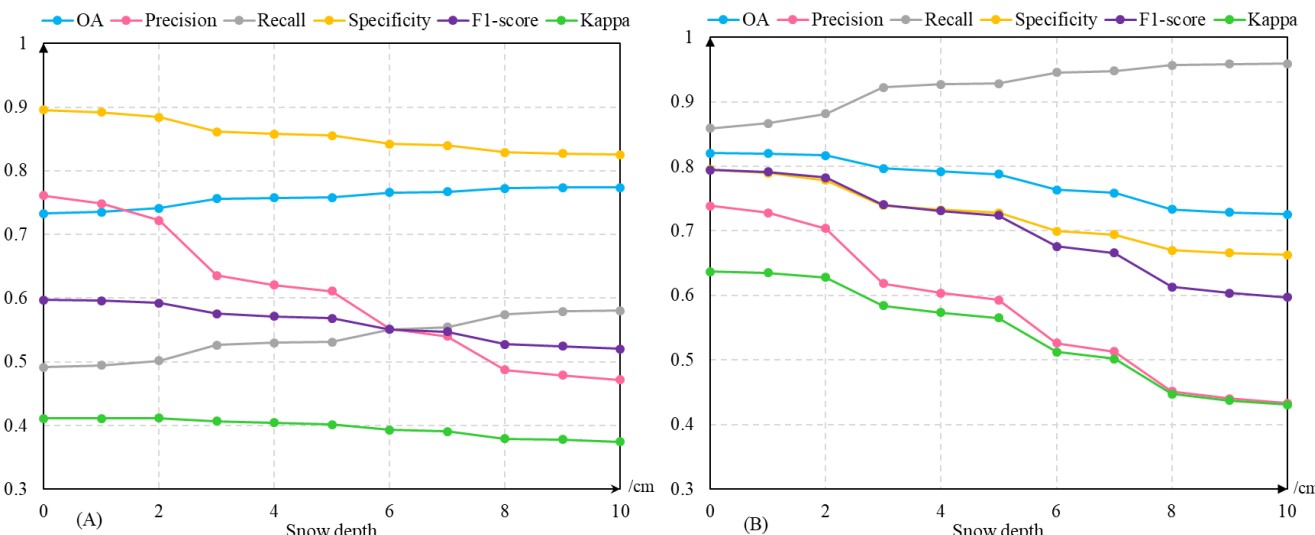

Fig. 9. Comparison of our estimated fractional snow cover (C, 6.25-km) with the reference MODIS fractional snow cover (B, 6.25-km) with respect to the MODIS composite binary snow cover products (A, 500-m); and a comparison example in the Central Canada area (D) on February 27th, 2017 (2017058). [Cf. the results in continuous value (Fig. S-7 in the Appendix)]

Fig. 10. The changes of accuracy indicators (OA, precision, recall, specificity, F1-score, kappa) for snow cover detection results of two algorithm (A: Grody' algorithm; B: Random forest) with increasing in situ snow depth value.

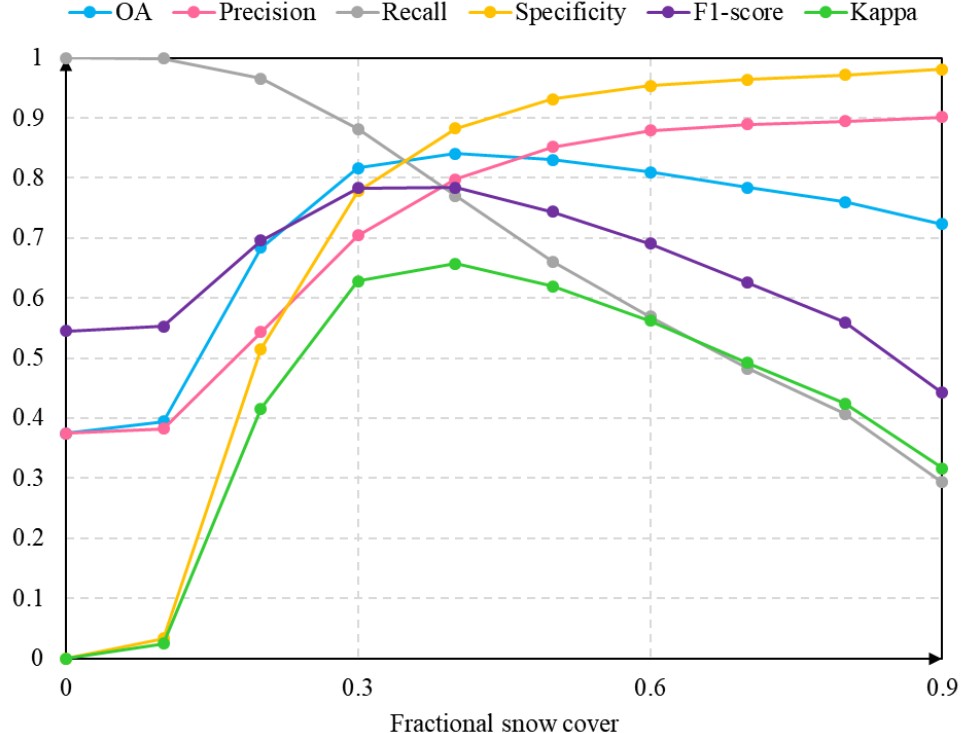

Fig. 11. The changes of accuracy indicators (OA, precision, recall, specificity, F1-score, kappa) for snow cover detection results with increasing fractional snow cover value (FSC).

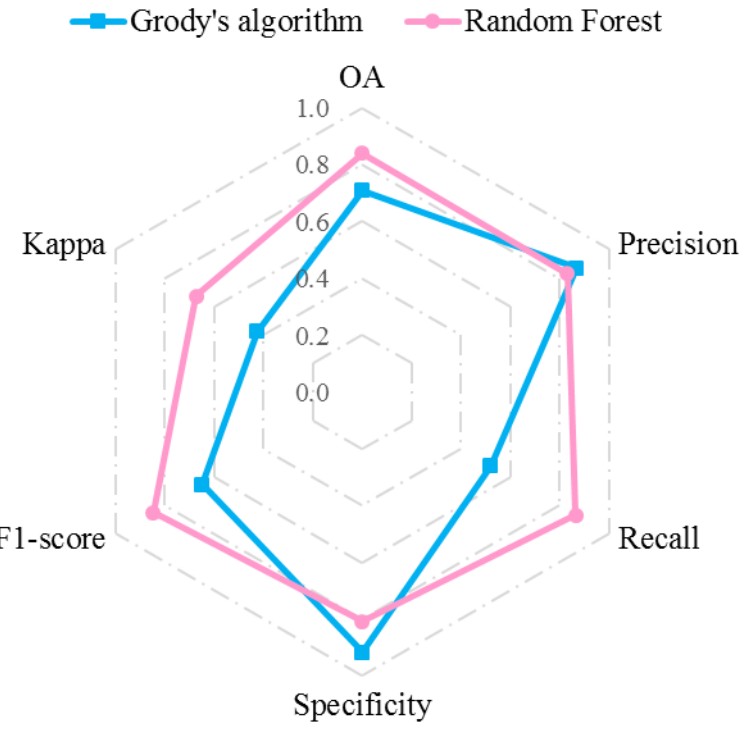

Fig. 12. The accuracy indicators (OA, precision, recall, specificity, F1-score, kappa) of snow cover detection from two algorithm (Grody' algorithm; Random forest).

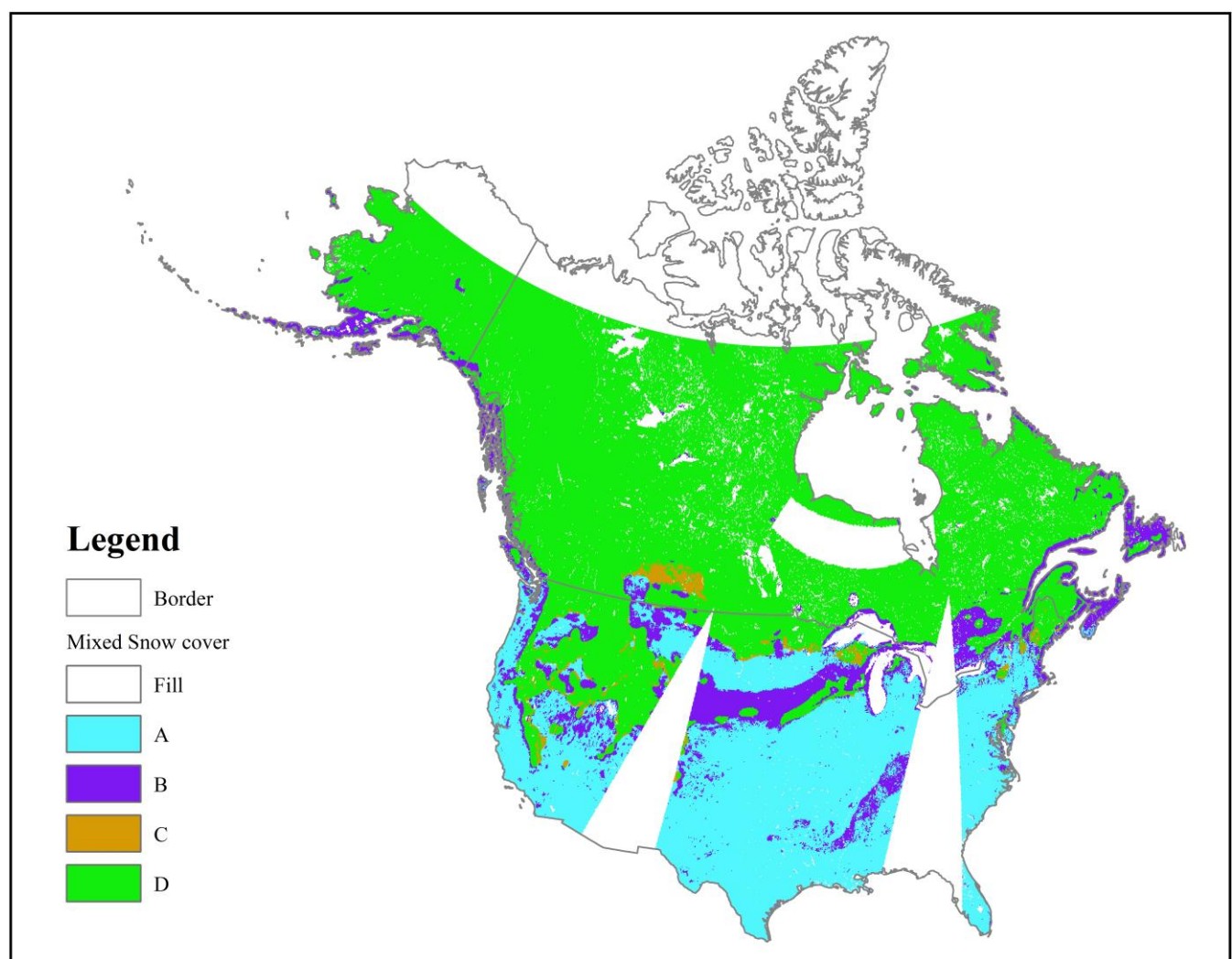

Fig. 13. The mixed snow cover detection map for different condition combinations of Random forest SCA and Grody's algorithm SCA on February 27th, 2017 (2017058). A: $SC_{Grody} = 0 \, \& \, FSC \leq 0.3$; B: $SC_{Grody} = 0 \, \& \, FSC > 0.3$; C: $SC_{Grody} = 1 \, \& \, FSC \leq 0.3$; D: $SC_{Grody} = 1 \, \& \, FSC > 0.3$; $SC_{Grody} = 0$ denotes snow-free (precipitation, cold desert and frozen ground) determined by Grody's algorithm SCA, otherwise it is snow-covered; $FSC \leq 0.3$ denotes snow-free cover detected by random forest SCA, otherwise, it is snow-covered.