# Peer review of "Estimating fractional snow cover from passive microwave brightness temperature data using MODIS snow cover product over North America"

_The Cryosphere, 2019_

## Short Comment (SC1) · 18 Jan 2020

In Sect 3.6 the authors correctly mentioned that different thresholds were used to convert snow depth into binary snow cover in the literature. In particular they referred to our studies (Gascoin et al. 2015; Gascoin et al. 2019) where we reported different optimal threshold values (15 cm and 2 cm). However, it is important to note that these thresholds are not contradictory since they were obtained from products with different spatial resolutions. In the first case, we optimized the snow detection threshold with MODIS snow products (500 m) while in the second case we used the Sentinel-2 Theia

snow collection (20 m). This threshold difference is consistent with the heterogeneous spatial distribution of the snow cover on the land surface. In other words, the larger the pixel, the deeper the snowpack needs to be, to be detected as "snow-covered" by remote sensing.

---

## Referee Comment (RC1) · Anonymous Referee #1 · 24 Jan 2020

This manuscript describes the development and validation of a technique to estimate fractional snow cover (FSC) from passive microwave brightness temperatures. Optical FSC estimates for algorithm training and validation were derived from MODIS Collection 6. Surface snow depth measurements and an independent passive microwave snow extent classifier were also used for evaluation. Overall, the study is comprehensive and detailed. I commend the authors for the thorough nature of the study – multiple combinations of passive microwave measurements are considered, sensitivity to various configurations of the retrieval are compared, and multiple datasets are used

for evaluation. Because of this comprehensive approach, description of the analysis is sometimes unclear in some places, and the logic is not always clear on the back and forth conversion between FSC information derived via the retrieval and comparison with MODIS, and binary snow extent information used for evaluation. This can get confusing in places. But overall, the technique shows good promise, and this initial overview makes for a new contribution worthy of publication The Cryosphere.

Please note that the paper requires a thorough edit for grammar, English usage, and word choice. Edits of this nature were too numerous to identify individually in my review.

General comments

Please double check all the data citations in Section 2.1 and Section 2.2. Some citations are missing from the reference list. While it's fine to provide the URL to the NSIDC webpage which hosts the data, the proper data citations (which are provided under the "Citing These Data" tab on the NSIDC webpages) must also be used.

Section 2.3.2: why is the IGBP land cover data product described here in addition to the MCD12Q1 product? This dataset does not seem to be used in the analysis...

Page 6 lines 14-23: Previous work has shown the potential for passive microwave SWE datasets, despite high uncertainty in the SWE retrievals, to provide useful snow extent information. This provides additional justification for the approach developed in this study. A brief mention of this could be added to this paragraph, including a citation to: Brown, R., C. Derksen, and L. Wang. 2010. A multi-dataset analysis of variability and change in Arctic spring snow cover extent, 1967-2008. Journal of Geophysical Research. 115: D16111, doi:10.1029/2010JD013975.

Section 3.1: I was disappointed e that the analysis period was limited to January and February. This is a real limitation because the spring period is the most important with respect to the snow-albedo feedback and the contribution of snow melt to streamflow. Additionally, the snow melt period may pose significant challenges to the use of passive

microwave data because of a loss of sensitivity to snow when it is wet. This limitation to the study is acknowledged in Section 5.1, but I suggest the conclusions and discussion clearly emphasize that these results are applicable to dry snow conditions, and that performance is likely to be weaker during snow melt.

Section 3.2: the short-term cloud filter for single days of cloud cover is clearly described (page 8 line 21) but it's not clear how longer cloudy periods are dealt with. If cloud is present for two or more consecutive days, is that pixel masked as cloud as described on page 9 line 3? Please state this clearly.

Section 4.1.1: there is virtually no difference in performance between scenarios 1, 4, and 5, as summarized in Table 4, with the main difference in performance between scenarios due to the inclusion of ancillary fields (lat/lon; topography). While I agree that "location information and topographic factors play a crucial role in snowpack distribution" can a more physically-based explanation be provided for these results?

Section 4.3/Figures 6 and 7: the scatterplots seem to illustrate that the retrieval is capable of identifying low snow fraction and high snow fraction, but with less skill across the intermediate values. This may be in large part due to issues with the reference snow fraction from MODIS, which seems to be clustered around low and high snow fraction values as shown in Figure 7 (with the exception of forested areas as shown in Figure 7a). Please consider adding some text to the first paragraph of Section 4.3, or strengthening the text on page 20 lines 10-20 to make clear how the performance of the retrieval can be influenced by the behaviour of the reference dataset.

Figure 8: the paper would be strengthened with more emphasis on the presentation of spatial results. Figure 8 is really important, but I found it unclear, especially panel D (the sub-panels within panel D are hard to read). Why is there so much white space in panel B? Zero snow fraction needs to have a separate colour than the range of 0 to 0.3, in order to clearly show where the retrieval estimates no snow versus very low fractions of snow (e.g. 0.1 to 0.3). I suggest a clear set of maps be presented, with emphasis

on a comparison between MODIS and passive microwave estimates at the continental scale (as in panels B and C) for some key events which extended the snowline.

Page 18 lines 3-6/Page 19 lines 27-28/Page 22 lines 1-3: the explanation for the potential over-identification of snow in the microwave retrievals (compared to the Grody product) is not convincing. The misclassification of snow extent due to non-snow scatterers (like cold deserts/frozen ground) is not a prevalent issue in North America. To better understand the statement that "the non-snow scatterer is the major source of snow cover misclassification for random forest FSC results" it would be clearer to show a map of locations where the RF classifier identifies snow and the Grody algorithm does not. This aspect needs to be explored in more detail in the final manuscript.

Editorial comments: Abstract line 23: change '0.31 million' to '310 000' Abstract line 26: I suggest not referring to the passive microwave dataset used for comparison as 'Grody's snow mapping algorithm' in the abstract. Page 2 line 2: change 'cycles' to 'cycle' Page 2 line 5: 'vast number of water resources' awkward wording Page 3 lines 20-25: when possible, try to use product names instead of the author names. For example, the Kelly (2009) reference refers to the NASA standard AMSR-E snow water equivalent product. The citations should be retained, just the product names changed. Page 3 line 28 and page 20 line 17: change 'patch' to 'patchy' Page 4 line 7: change 'predict' to 'retrieve' Page 5 line 7: change 'America' to 'United States' Page 8 line 4: not clear what is meant by 'fill' Page 18 lines 10-14: this text is unclear and seems very anecdotal. I think it can be removed. Figure 1: Add units to the legend. Why is there negative elevation? Figure 4: caption is not clear Figure 9: add x-axis label to indicate snow depth Figure 10: add axis labels
* * *

---

## Referee Comment (RC2) · Anonymous Referee #2 · 5 Apr 2020

Overview and General Comments

This manuscript describes a new approach of estimating fractional snow fraction from satellite-based passive microwave (PM) sensors and higher resolution MODIS snow cover estimates. The authors present different regression and machine learning type algorithms, including multi-regression, artificial neural networks (ANN), and a random forest regression technique, for estimating the PM-based snow cover fraction using the MODIS snow cover as a reference input to the algorithms along with accounting for different PM retrieval and ancillary datasets, like vegetation types. The methods

are demonstrated and validated against independent in situ measurements across the region of interest (Canada and the US).

Overall, the paper includes comprehensive descriptions of the data and methods used, and detailed background and justification for the work presented. It also is within the scope and appropriate for the journal, The Cryosphere. The supplementary material does help support the overall findings in the paper. However, some of the methods and conclusions may require some revision and may not be conclusive enough as there is a limitation on the years evaluated and the wintertime period focused on. A few major and minor comments are noted in this review that hopefully help to strengthen the paper and the organization of the methods and results presented. There are a few sections that were difficult to follow and some of the English grammar and syntax was unclear.

One downside to this study is that the authors only focused on seven years of available passive microwave and optically based snow cover observations and then just the peak snow months of January and February. Though it seems to make sense to focus only on when the snowpack is at the peak months and more spatially continuous, however, it is also worthwhile to capture the temporal and spatial heterogeneity in the accumulation and ablation seasons and more fully test the algorithms described and applied in this study. Otherwise, the algorithms are only somewhat effective for peak wintertime in US and Canada and not applicable for studies, like prescribing observational snow cover conditions in climate projection or snow-land-atmosphere climate interaction studies, which are pointed out as one primary reason to perform this present study.

Also, in relation to the timeframe of the training and validation data years, only having one year to perform the validation seems quite limiting, as a given year can be hard to note overall performance given snow cover can vary greatly from year to year (e.g., snow drought conditions). This is somewhat reflected in Figure 7 (right column panels), which show how highly variable and not as predictable in the validation year (2017).
Please explain why a longer period of record is not used, e.g., 2002-2019 (Terra+Aqua MODIS combined) and the passive microwave combined product by Brodzik et al. (2018), to perform the training and validation period. Perhaps, use Water Years (WY) 2002-2013 for training and WY 2014-2018 for validation?

Using only one year for testing and a second year for validation is very limiting for this study, and it is highly recommended for additional years to be included. Also, for the four different approaches of estimating the fractional snow cover from passive microwave should have longer evaluations performed in this context as the summary of the results would be inconclusive for one year of validation.

Some of the methods sections are hard to follow, though the authors provide many details there and in the Supplemental material. For example, Section 3.3.1 of "Selecting input variables" was at times hard to follow and why each scenario was selected. Improving the organization of the sections to flow better in terms of their logic and why different experiments were performed would be helpful for the overall background and discussions of this study.

The English grammar and syntax used require additional review and editing by editorial services to help correct these issues before resubmitting. A few suggested corrections are offered below in the technical corrections section.

Specific Comments

Abstract: The authors introduce "Grody's snow cover mapping algorithm" towards the end of the abstract without any other background. Perhaps they could provide one introductory phrase on this algorithm within the abstract to give more context.

Page 2, Lines 9-10: The authors mention that snow cover data from station measurements are "time-consuming, [and] cumbersome,". What do the authors mean by these adjectives? Please clarify here. Any dataset, including satellite, requires time and careful derivation of the final product. However, in situ snow cover data are spatially

discontinuous and require more time to maintain.

Page 6, lines 11-12: Would like to point out here that North America includes Mexico as well. The authors should specify that their study domain spans the continental U.S. and Canada only.

Page 7, lines 10-11: Authors state here that "to the best of our knowledge, there are no researchers have developed fractional snow cover . . . using passive microwave data." Please take a look at the following references and cite appropriately:

Foster, J.L., D. K. Hall, J. B. Eylander, G. A. Riggs, S. V. Nghiem, M. Tedesco, E. Kim, P.M. Montesano, R. E. J. Kelly, K. A. Casey and B. Choudhury (2011): A blended global snow product using visible, passive microwave and scatterometer satellite data, International Journal of Remote Sensing, 32:5, 1371-1395, DOI: 10.1080/01431160903548013

Page 8, lines 24-27: It would be helpful here to provide a lead in sentence to introduce your first two equations.

Page 8, last line: "Calculation areas should be in a larger feet . . ." What is meant here by "feet"? It does not seem to make sense to use this word here, but perhaps "footprint area" makes more sense? Please correct.

Page 11, lines 2-3: MODIS Collection 5 products are considered older and not "current", as they have been replaced by Collection 6. Recommend removing "current" here.

Subsection 3.4.1: The authors discuss both the linear and multi-linear regression methods here, which makes the discussion confusing to follow. They then have the reader refer to the Supplementary material for more information. It is recommended that the authors better describe in this subsection how the "linear regression" is applied. Was it based on the equations in Salomonson and Appel (2004) or new linear equations and parameters derived for the four different vegetation categories? Please try to better

organize and explain this linear method in this subsection.

Page 14, lines 10-11: Please provide citations and references where possible for the metrics, especially Cohen's kappa coefficient and the F1 score.

Page 15, Lines 11-12: Authors indicate here that their "Scenario-6" variable sensitivity case "generated the worse performance, with the low R, the great MAE and RMSE". When looking at Table 4 results, Scenario-6 appears to perform rather well overall. Perhaps it would help if the authors specify here that of the Scenarios of 1, 4-5 and 6, Scenario-6 performs the "worst". It is also recommended to change the last part of that sentence to: " this scenario's setting had the third worst performance with lower R values and higher MAE and RSME values."

Page 15, line 31 to top of Page 16: Make "Figure" plural and change the last part of this sentence to something like: "show that this finding was not coincidental." This sentence is a bit hard to understand in what is meant by "not coincidental". Please elaborate or better explain the meaning here.

Page 16, line 29: Please clarify here what is meant by "neglected to assess the rationality of estimated value . . .". Are you referring to the out-of-bounds events that occur in the other methods, other than the random forest approach and that that "rational" was not well checked?

Page 21, line 1: Authors state that only a few studies validate the accuracy of MODIS snow cover products in forested areas. Actually, there are several in addition, including:

Arsenault, K.R., P.R. Houser and G. J.M. De Lannoy, 2014: Evaluation of the MODIS snow cover fraction product, Hydro. Proc., 30, 3, pps. 980-998. https://onlinelibrary.wiley.com/doi/full/10.1002/hyp.9636

Kostadinov, T. S., and T. R. Lookingbill, 2015: Snow cover variability in a forest ecotone of the Oregon Cascades via MODIS Terra products, Rem. Sens. Env., 164, pps. 155-169. https://www.sciencedirect.com/science/article/pii/S0034425715001303

Page 22, lines 14-16: The first statement here about the "strong limitations in the understanding of physical mechanism" is a bit hard to understand. Are the authors referring to the underlying physics and characteristics that relate the fractional snow to the signature of the passive microwave bright temperature responses? Perhaps, it might be better to frame these concluding statements more in that way vs. "mechanisms".

Table 1: In the row of references, does the Xiao et al. (2018) paper cover both Scenario-4 and -5 columns in the table? If so, it might be helpful to specify this in the body of the paper.

Figure 8: In panel A, more binary MODIS snow cover present (e.g., large green pixeled areas in Canada), but that does not seem to get translated over to panel B for the fractional MODIS snow cover (mostly filled in with no fractional values). Please explain why most of the derived MODIS snow cover fraction is removed here, especially over Canada? Also, for the MODIS snow fractional product, there is no fractional snow representation between 0.3 and 0.8, the other two categories shown in panel B. What is happening here in that regard – no fractional snow within 0.3 and 0.8 at any noticeable gridcells? Please provide an explanation in the text as well.

Finally for Figure 8, it would be helpful to assign a different color and category for the non-snow pixels (at fractional value of 0.) in panels B and C to better discriminate the non-snow areas from the snow-based areas. Currently, snow-free pixels are lumped in with the low snow fraction category of 0 to 0.3.

Fig. 11: This is a nice figure that summarize and present these results well.

Technical corrections

Page 2, line 25: Please specify what "FY" stands for in "FY series sensors".

Page 3, line 25: Awkward phrasing here: "To unite resolution, . . ." Perhaps try: " To be at a common resolution, . . ."

Page 5, line 7: Recommend here to separate the two phrases here with either a semicolon (between "collected" and "all available") or place the conjunction "and" after the comma.

Page 6, line 6: Please specify what "ETOPO1" stands for.

Page 6, line 11: Add citation and reference for "ArcGIS 10.5" software.

Page 6, line 17: Replace "heterogeneous" with the noun, "heterogeneity".

Page 7, line 7: MODIS misspelled here as "MODSI".

Page 7, line 31: Remove "with" before "accurate".

Page 9, line 21: Either replace the semicolon with a period, or make the word, "Thereby", lower-case.

Page 9, line 27: Change the "not" in this line to "cannot". Also on that same line, the word use of "Correspondingly" here does not seem to make sense.

Page 10, line 5: Make "variable" plural here in "an optimal combination of input variables".

Page 13, line 6: "researches" should be changed to "researchers".

Page 18, line 5: Remove "be" before "misclassified" and change "into" to "as". Also, please remove the phrase, "As we all know", and change the start of the second sentence there to: "Permafrost is known to be widely distributed in the northern part of . . ."

Page 20, line 4: Change "researches" to "studies".

Page 21, lines 23-24: Change "were" to "was" in relation to "The accuracy of the proposed algorithm was further . . .".

Table 2 caption: "unite" should be "unit", and "clod desert" should be "cold desert".

Figure 7: The use of the capitalized and lower-case plot labels is fine but not conventional. Would it make more sense to simply use, "A, B" then "C, D", etc., for the paired columns?

[Figure]

---

## Author Comment (AC1) · 30 Apr 2020

* * *
*************************\*\*\***Reply to comments from Simon Gascoin**\*\*\*************************
* * *
In Sect 3.6 the authors correctly mentioned that different thresholds were used to convert snow depth into binary snow cover in the literature. In particular they referred to our studies (Gascoin et al. 2015; Gascoin et al. 2019) where we reported different optimal threshold values (15 cm and 2 cm). However, it is important to note that these thresholds are not contradictory since they were obtained from products with different spatial resolutions. In the first case, we optimized the snow detection threshold with MODIS snow products (500 m) while in the second case we used the Sentinel-2 Theia snow collection (20 m). This threshold difference is consistent with the heterogeneous spatial distribution of the snow cover on the land surface. In other words, the larger the pixel, the deeper the snowpack needs to be, to be detected as "snow-covered" by remote sensing.

Response: Thanks for your constructive suggestion and comment. We revised and clarified the description about the snow depth threshold for converting snow depth into binary snow cover in Section 3.6 (page 15, lines 10-18).
"*Many different depth thresholds have been suggested in previous studies, for instance 2 cm for 20 m spatial resolution (Gascoin et al., 2019); 0 cm (Parajka et al., 2012), 1 cm (Zhang et al., 2019), 3 cm (Hao et al., 2018), 4 cm (Huang et al., 2018; Wang et al., 2008) and, 15 cm (Gascoin et al., 2015) for 500 m spatial resolution; 2.5 cm for 5 km spatial resolution (Hori et al., 2017); 3 cm (Xu et al., 2016) and 5 cm for 25 km spatial resolution (Liu et al., 2018); and 2 cm for 0.75˚ grid resolution (Brown and Derksen, 2013). Due to these significant disagreements in the depth thresholds, Gascoin et al. (2019) conducted a sensitivity experiment that tested the agreement between in-situ measurements and optical snow cover area products. The sensitivity of passive microwave snow cover identification results to snow depth at 6.25 km spatial resolution was also tested by computing the accuracy metrics with snow depth increasing from 0 to 10 cm.*"

---

## Author Comment (AC2) · 30 Apr 2020

**Dear editor, reviewers**

We would like to thank you and the reviewers for the constructive and insightful comments and suggestions to improve our manuscript. We have carefully revised the manuscript according to the suggestions and comments, and provide point-by-point response following each comment and suggestion.

In the following, reviewer comments are given in black and responses are given in blue (the revised sentence was set in italics). The corresponding changes have been made in the revised paper with track changes.

We think the revised manuscript has addressed all the reviewers' comments and hopefully it is now suitable for publication in The Cryosphere

Sincerely,

Xiongxin Xiao

**REVIEWER 1**

This manuscript describes the development and validation of a technique to estimate fractional snow cover (FSC) from passive microwave brightness temperatures. Optical FSC estimates for algorithm training and validation were derived from MODIS Collection 6. Surface snow depth measurements and an independent passive microwave snow extent classifier were also used for evaluation. Overall, the study is comprehensive and detailed. I commend the authors for the thorough nature of the study – multiple combinations of passive microwave measurements are considered, sensitivity to various configurations of the retrieval are compared, and multiple datasets are used for evaluation. Because of this comprehensive approach, description of the analysis is sometimes unclear in some places, and the logic is not always clear on the back and forth conversion between FSC information derived via the retrieval and comparison with MODIS, and binary snow extent information used for evaluation. This can get confusing in places. But overall, the technique shows good promise, and this initial overview makes for a new contribution worthy of publication The Cryosphere.

Please note that the paper requires a thorough edit for grammar, English usage, and word choice. Edits of this nature were too numerous to identify individually in my review.

Response: Thanks for your valuable comments and suggestions to improve our manuscript. We have replied to each comment below. The manuscript has been edited by a native English speaker. Additionally, to make the description of the conversion from fractional snow cover to binary snow cover clear, we changed "random forest FSC" to "random forest SCA" in binary snow cover area information evaluation in the revised manuscript.

**General comments**

Please double check all the data citations in Section 2.1 and Section 2.2. Some citations are missing from the reference list. While it's fine to provide the URL to the NSIDC webpage which hosts the data, the proper data citations (which are provided under the "Citing These Data" tab on the NSIDC webpages) must also be used.

Response: Thanks for your suggestion. We updated and added the corresponding data citations for the dataset used in Section 2.

Section 2.3.2: why is the IGBP land cover data product described here in addition to the MCD12Q1 product? This dataset does not seem to be used in the analysis: :

Response: MODIS land cover data have several classification scheme, including the IGBP classification schemes. The MODIS land cover data with IGBP classification scheme was used as the basis data of fractional snow cover retrieval model

Page 6 lines 14-23: Previous work has shown the potential for passive microwave SWE datasets, despite high uncertainty in the SWE retrievals, to provide useful snow extent information. This provides additional justification for the approach developed in this study. Abrief mention of this could be added to this paragraph, including a citation to: Brown, R., C. Derksen, and L. Wang. 2010. A multi-dataset analysis of variability and change in Arctic spring snow cover extent, 1967-2008. Journal of Geophysical Research. 115: D16111, doi:10.1029/2010JD013975.

Response: Thanks for your suggestion. We cited the related literature and added the description about snow parameters (snow cover extent, snow depth and water equivalent) retrieval in page 7 lines 16-19 as follows:

"A number of published work have demonstrated the potential to derive snow depth and SWE using passive microwave radiation data (Kim et al., 2019; Wang et al., 2019). Despite the high uncertainties associated with snow depth and SWE estimations, using passive microwave data can provide useful snow cover extent information (Brown et al., 2010; Foster et al., 2011)."

Section 3.1: I was disappointed e that the analysis period was limited to January and February. This is a real limitation because the spring period is the most important with respect to the snow-albedo feedback and the contribution of snow melt to streamflow. Additionally, the snow melt period may pose significant challenges to the use of passive microwave data because of a loss of sensitivity to snow when it is wet. This limitation to the study is acknowledged in Section 5.1, but I suggest the conclusions and discussion clearly emphasize that these results are applicable to dry snow conditions, and that performance is likely to be weaker during snow melt.

Response: Thanks for your comment. We do agree that the estimation and analysis of fractional snow cover should cover the whole snow cover season (autumn, winter and spring). Noted that the fractional snow cover estimation work we're doing will cover all the year round. Additionally, we clarified the description information of applicable condition for this study in Section 6 (page 26 lines 1-3) based on your suggestion:

"These models established using several data sources in January and February had better applicability in dry snow conditions, while estimation results could be less accurate in wet snow conditions."

Section 3.2: the short-term cloud filter for single days of cloud cover is clearly described (page 8 line 21) but it's not clear how longer cloudy periods are dealt with. If cloud is present for two or more consecutive days, is that pixel masked as cloud as described on page 9 line 3? Please state this clearly.

Response: Thanks. If cloud is present for two or more consecutive days, the pixel would be masked as cloud according to short term cloud filter. Additionally, we revised and clarified the description about the short term cloud filter (page 9 lines 21-24)

"2) Short-term temporal filter: if the status of a pixel in the input image (MCD10A1) in a given day (t) was cloud and both the preceding (t - 1) and succeeding (t + 1) days were snow-covered (or snow-free), the pixel in the output image (MCTD10A1) in the given day (t) was assigned as snow-covered (or snow-free) (summarized by Eq. 2)..."

and revised the confused term "filter" in original sentence to

"We adopted the most rigorous pixel filtering rule, by which one clouded pixel cannot be allowed within a 15\*15 pixel window" in page 10 lines 4-5.

Section 4.1.1: there is virtually no difference in performance between scenarios 1, 4, and 5, as summarized in Table 4, with the main difference in performance between scenarios due to the inclusion of ancillary fields (lat/lon; topography). While I agree that "location information and topographic factors play a crucial role in snowpack distribution" can a more physically-based explanation be provided for these results?

Response: Thanks for your comment. The results of Scenarios-1, 4, and 5 show that there indeed were no significant differences among these three scenarios. Generally speaking, inputting more information could make great contribution to improving the performance of snow cover parameters estimation. However, we found that inputting more information did not provide too much contribution for the performance improvement of fractional snow cover by analyzing the results of Scenarios-1, 4, and 5. Thus, we conclude that the input variables in Scenarios-1 have redundant information and it makes model establishment more time consuming. These statements have been similarly described in our manuscript "*The comparison among Scenarios -1, 4, 5 indirectly indicates that the variables used in Scenario -1 may have some information*

redundancy and slightly weaken the efficiency of the random forest retrieval model" in page 17 lines 17-19

Additionally, we added the explanation for the "location information and topographic factors" in page 17 lines 6-9 "In this study, the retrieval method required these five basic input variables as auxiliary information in order to learn the characteristics of snow cover under different surface conditions to assist in accurately estimating snow cover properties. In contrast, in the absence of these basic input variables, the established model has no advantage in accurately predicting the characteristics of fractional snow cover under complex surface conditions"

Section 4.3/Figures 6 and 7: the scatterplots seem to illustrate that the retrieval is capable of identifying low snow fraction and high snow fraction, but with less skill across the intermediate values. This may be in large part due to issues with the reference snow fraction from MODIS, which seems to be clustered around low and high snow fraction values as shown in Figure 7 (with the exception of forested areas as shown in Figure 7a). Please consider adding some text to the first paragraph of Section 4.3, or strengthening the text on page 20 lines 10-20 to make clear how the performance of the retrieval can be influenced by the behaviour of the reference dataset.

Response: Thanks for your comment. In order to clarify the influence of reference dataset to fractional snow cover retrieval, we added the following statement in page 20 lines 28-29.

"This is mainly because a smaller number of samples with intermediate values from the reference dataset used in the training model may not properly capture the characteristics of the surface condition with intermediate fractional snow covers"

Figure 8: the paper would be strengthened with more emphasis on the presentation of spatial results. Figure 8 is really important, but I found it unclear, especially panel D (the sub-panels within panel D are hard to read). Why is there so much white space in panel B? Zero snow fraction needs to have a separate colour than the range of 0 to 0.3, in order to clearly show where the retrieval estimates no snow versus very low fractions of snow (e.g. 0.1 to 0.3). I suggest a clear set of maps be presented, with emphasis on a comparison between MODIS and passive microwave estimates at the continental scale (as in panels B and C) for some key events which extended the snowline.

**Response: Thanks you very much for your valuable suggestion.**

1) The MODIS binary snow cover image (Fig. 8A) was translated to the reference MODIS fractional snow cover (Fig. 8B) by applying the pixel filtering rule at a 15\*15 pixels window that do not allow an cloudy pixel when calculating the fractional snow cover. Then it resulted in that many pixels to be masked as "fill value" (white in Figure 8).

2) We modified and clarified why the separate color map was used in here.

"Fig. 8 shows the comparison between our estimated fractional snow cover and the reference MODIS fractional snow cover, and more importantly, provides another perspective for snow cover identification in Section 4.4. Thus, Fig. 8B and 8C used 0.3 as the threshold of fractional snow cover to define snow-covered and snow-free area, and this was adopted through the experiments in Section 4.4" in page 20 lines 11-14.

3) Moreover, according to your suggestions, we strengthened the description of spatial results in order to improve the legibility of each image (Fig. 8), and revised the statements as follows:

"Apart from the scatter plots and statistical analysis, Fig. 8 shows the distribution pattern of snow cover from a spatial perspective, including MODIS composite binary snow cover (Fig. 8A), MODIS fractional snow cover (Fig. 8B), and the estimated fractional snow cover by the proposed algorithm (Fig. 8C). When the most rigorous pixel filtering rule at the 15\*15 pixel window was applied (see Section 3.2), the large number of cloud covered pixels (yellow) in Fig. 8A resulted in most areas of the MODIS fractional snow cover image (Fig. 8B) being represented by a "fill value". Additionally, the number of intermediate values for MODIS fractional snow cover in winter would be much lower than the number of values near the two extreme values (0 and 1). In contrast, the estimated fractional snow cover from passive microwave brightness temperature data can provide almost complete coverage and continuous spatial information on snow cover (Fig. 8C; Fig. S-7 in the Appendix). Fig. 8 shows the comparison between our estimated fractional snow cover and the reference MODIS fractional snow cover, and more importantly, provides another perspective for snow cover identification in Section 4.4. Thus, Fig. 8B

and 8C used 0.3 as the threshold of fractional snow cover to define snow-covered and snow-free area, and this was adopted through the experiments in Section 4.4. This means that the pixel was identified as snow cover when fractional snow cover value was less than 0.3. From Fig. 8A – C, the spatial pattern of estimated fractional snow cover from the proposed method seems to accurately capture the distribution of snow cover from MODIS under clear-sky conditions, such as the snow-free area in most areas of North America, and snow-covered areas in northern Canada. Fig. 8D presents a specific example comparing these two fractional snow cover datasets and MODIS composite binary snow cover products in central Canada on February 27th, 2017. Based on this example, we find that our estimated fractional snow cover was capable of obtaining snow cover distribution when most of the area was covered by cloud, which was not the case for MODIS. This example also show that the extent of snowline observed in the MODIS binary snow cover image (500 m), which was the boundary between snow-covered and snow-free, was well described and exhibited by the estimated fractional snow cover (6.25 km)" in page 20 lines 3-23.

Moreover, the estimation results comparison of fractional snow cover for MODIS and our proposed algorithm in continuous value has been shown in the supplement file:

Figure S-7. Comparison of the reference MODIS fractional snow cover (A) with our estimated fractional snow cover (B) in continuous value (6.25-km) on February 27th, 2017 (2017058)

Page 18 lines 3-6/Page 19 lines 27-28/Page 22 lines 1-3: the explanation for the potential over-identification of snow in the microwave retrievals (compared to the Grody product) is not convincing. The misclassification of snow extent due to non-snow scatterers (like cold deserts/frozen ground) is not a prevalent issue in North America. To better understand the statement that "the non-snow scatterer is the major source of snow cover misclassification for random forest FSC results" it would be clearer to show a map of locations where the RF classifier identifies snow and the Grody algorithm does not. This aspect needs to be explored in more detail in the final manuscript.

Response: Thanks for your comment. Although the commission error of the proposed algorithm in snow cover identification only have 0.17, we provided additional information to explain this kinds of error. As you say, the non-snow scatters (like cold deserts, frozen ground) is not a prevalent issue in North America. According to our study, we can also conclude that the snow cover misclassification effected by cold deserts and frozen ground is not prevalent issue in North America. We then specified the different source error for commission error and revised the statement as follows:

"The records, which were misclassified as snow cover by random forest SCA, although they are non-snow scatter components (precipitation, cold desert, and frozen ground), account for 70.1% of total misclassification records (CE = 0.17), of which 63.0% comes from precipitation, 6.4% from cold desert, and 0.7% from frozen ground" in page 23 lines 5-8.

Grody's algorithm SCA, when the in-situ station observation is absent. We provided the following statistical metrics (Table A) using the data in 2017. We can see that the percentage of "True observation" for Grody's algorithm only is 24.9% when RF classifier identifies snow-covered and the Grody's algorithm does not (Condition B); inversely, it should be classified as snow-covered. If we do not use the in-situ observation as the "true" observation, we do not have high confidence to say that the detection results by our proposed algorithm in Condition B are not right. Moreover, we show an example that provides a map for different condition combinations of Random forest SCA and Grody's algorithm SCA (Fig. S-9). The inconsistencies between Random forest SCA and Grody's algorithm SCA usually occurred in the mid-latitude region, in which it has the low fractional snow cover (Figure S-7). And also we revised the statement to

"For different results for these two snow cover mapping algorithms, we have used an example to show the inconsistencies and consistencies in mapping between the random forest SCA and Grody's algorithm SCA (Fig. S-9)" in page 23 lines 10-11.

Table A. The effect of precipitation, cold desert and frozen ground in snow cover misclassification. FP is false positive that means it is the number of pixels that are misclassified as snow cover by Random forest FSC.  $SD_{obs} = 0$  denotes snow-free measured in station, otherwise, it is snow-covered;  $SC_{Grody} = 0$  denotes snow-free (precipitation, cold desert and frozen ground) determined by Grody's algorithm, otherwise it is snow-covered;  $FSC \le 0.3$  denotes snow-free cover detected by our method, otherwise, it is snow-covered.

| No. | Conditions                      | Observation    |                | Percentage of "True observation" |                      |
|-----|---------------------------------|----------------|----------------|----------------------------------|----------------------|
|     |                                 | $SD_{obs} = 1$ | $SD_{obs} = 0$ | Random
Forest                 | Grody's
algorithm |
| Α   | $SC_{Grody} = 0 \& FSC \le 0.3$ | 17435 (13%)    | 116069 (87%)   | 87%                              | 87%                  |
| В   | $SC_{Grody} = 0 \& FSC > 0.3$   | 60601 (75.1%)  | 20063 (24.9%)  | 75.1%                            | 24.9%                |
| С   | $SC_{Grody} = 1 \& FSC \le 0.3$ | 4379 (51.5%)   | 4120 (48.5%)   | 48.5%                            | 51.5%                |
| D   | $SC_{Grody} = 1 \& FSC > 0.3$   | 80167 (90.3%)  | 8575 (9.7%)    | 90.3%                            | 90.3%                |

---

## Author Comment (AC3) · 30 Apr 2020

Dear editor, reviewers

We would like to thank you and the reviewers for the constructive and insightful comments and suggestions to improve our manuscript. We have carefully revised the manuscript according to the suggestions and comments, and provide point-by-point response following each comment and suggestion.

In the following, reviewer comments are given in black and responses are given in blue (the revised sentence was set in italics). The corresponding changes have been made in the revised paper with track changes.

We think the revised manuscript has addressed all the reviewers' comments and hopefully it is now suitable for publication in The Cryosphere

Sincerely,

Xiongxin Xiao
* * *
******************** **Reply to comments from anonymous reviewer 1#** ********************
* * *
**REVIEWER 1**

This manuscript describes the development and validation of a technique to estimate fractional snow cover (FSC) from passive microwave brightness temperatures. Optical FSC estimates for algorithm training and validation were derived from MODIS Collection 6. Surface snow depth measurements and an independent passive microwave snow extent classifier were also used for evaluation. Overall, the study is comprehensive and detailed. I commend the authors for the thorough nature of the study – multiple combinations of passive microwave measurements are considered, sensitivity to various configurations of the retrieval are compared, and multiple datasets are used for evaluation. Because of this comprehensive approach, description of the analysis is sometimes unclear in some places, and the logic is not always clear on the back and forth conversion between FSC information derived via the retrieval and comparison with MODIS, and binary snow extent information used for evaluation. This can get confusing in places. But overall, the technique shows good promise, and this initial overview makes for a new contribution worthy of publication The Cryosphere.

Please note that the paper requires a thorough edit for grammar, English usage, and word choice. Edits of this nature were too numerous to identify individually in my review.

Response: Thanks for your valuable comments and suggestions to improve our manuscript. We have replied to each comment below. The manuscript has been edited by a native English speaker. Additionally, to make the description of the conversion from fractional snow cover to binary snow cover clear, we changed "random forest FSC" to "random forest SCA" in binary snow cover area information evaluation in the revised manuscript.

**General comments**

Please double check all the data citations in Section 2.1 and Section 2.2. Some citations are missing from the reference list. While it's fine to provide the URL to the NSIDC webpage which hosts the data, the proper data citations (which are provided under the "Citing These Data" tab on the NSIDC webpages) must also be used.

Response: Thanks for your suggestion. We updated and added the corresponding data citations for the dataset used in Section 2.

Section 2.3.2: why is the IGBP land cover data product described here in addition to the MCD12Q1 product? This dataset does not seem to be used in the analysis: :

Response: MODIS land cover data have several classification scheme, including the IGBP classification schemes. The MODIS land cover data with IGBP classification scheme was used as the basis data of fractional snow cover retrieval model

Page 6 lines 14-23: Previous work has shown the potential for passive microwave SWE datasets, despite high uncertainty in the SWE retrievals, to provide useful snow extent information. This provides additional justification for the approach developed in this study. A brief mention of this could be added to this paragraph, including a citation to: Brown, R., C. Derksen, and L. Wang. 2010. A multi-dataset analysis of variability and change in Arctic spring snow cover extent, 1967-2008. Journal of Geophysical Research. 115: D16111, doi:10.1029/2010JD013975.

Response: Thanks for your suggestion. We cited the related literature and added the description about snow parameters (snow cover extent, snow depth and water equivalent) retrieval in page 7 lines 16-19 as follows:

*"A number of published work have demonstrated the potential to derive snow depth and SWE using passive microwave radiation data (Kim et al., 2019; Wang et al., 2019). Despite the high uncertainties associated with snow depth and SWE estimations, using passive microwave data can provide useful snow cover extent information (Brown et al., 2010; Foster et al., 2011)."*

Section 3.1: I was disappointed e that the analysis period was limited to January and February. This is a real limitation because the spring period is the most important with respect to the snow-albedo feedback and the contribution of snow melt to streamflow. Additionally, the snow melt period may pose significant challenges to the use of passive microwave data because of a loss of sensitivity to snow when it is wet. This limitation to the study is acknowledged in Section 5.1, but I suggest the conclusions and discussion clearly emphasize that these results are applicable to dry snow conditions, and that performance is likely to be weaker during snow melt.

Response: Thanks for your comment. We do agree that the estimation and analysis of fractional snow cover should cover the whole snow cover season (autumn, winter and spring). Noted that the fractional snow cover estimation work we're doing will cover all the year round. Additionally, we clarified the description information of applicable condition for this study in Section 6 (page 26 lines 1-3) based on your suggestion:
*"These models established using several data sources in January and February had better applicability in dry snow conditions, while estimation results could be less accurate in wet snow conditions."*

Section 3.2: the short-term cloud filter for single days of cloud cover is clearly described (page 8 line 21) but it's not clear how longer cloudy periods are dealt with. If cloud is present for two or more consecutive days, is that pixel masked as cloud as described on page 9 line 3? Please state this clearly.

Response: Thanks. If cloud is present for two or more consecutive days, the pixel would be masked as cloud according to short term cloud filter. Additionally, we revised and clarified the description about the short term cloud filter (page 9 lines 21-24)
*"2) Short-term temporal filter: if the status of a pixel in the input image (MCD10A1) in a given day (t) was cloud and both the preceding (t- 1) and succeeding (t + 1) days were snow-covered (or snow-free), the pixel in the output image (MCTD10A1) in the given day (t) was assigned as snow-covered (or snow-free) (summarized by Eq. 2)..."*
and revised the confused term "filter" in original sentence to
*"We adopted the most rigorous pixel filtering rule, by which one clouded pixel cannot be allowed within a 15\*15 pixel window "*
in page 10 lines 4-5.

Section 4.1.1: there is virtually no difference in performance between scenarios 1, 4, and 5, as summarized in Table 4, with the main difference in performance between scenarios due to the inclusion of ancillary fields (lat/lon; topography). While I agree that "location information and topographic factors play a crucial role in snowpack distribution" can a more physically-based explanation be provided for these results?

Response: Thanks for your comment. The results of Scenarios-1, 4, and 5 show that there indeed were no significant differences among these three scenarios. Generally speaking, inputting more information could make great contribution to improving the performance of snow cover parameters estimation. However, we found that inputting more information did not provide too much contribution for the performance improvement of fractional snow cover by analyzing the results of Scenarios-1, 4, and 5. Thus, we conclude that the input variables in Scenarios-1 have redundant information and it makes model establishment more time consuming. These statements have been similarly described in our manuscript *"The comparison among Scenarios -1, 4, 5 indirectly indicates that the variables used in Scenario -1 may have some information*

*redundancy and slightly weaken the efficiency of the random forest retrieval model*" in page 17 lines 17-19

Additionally, we added the explanation for the "location information and topographic factors" in page 17 lines 6-9
"*In this study, the retrieval method required these five basic input variables as auxiliary information in order to learn the characteristics of snow cover under different surface conditions to assist in accurately estimating snow cover properties. In contrast, in the absence of these basic input variables, the established model has no advantage in accurately predicting the characteristics of fractional snow cover under complex surface conditions*"

Section 4.3/Figures 6 and 7: the scatterplots seem to illustrate that the retrieval is capable of identifying low snow fraction and high snow fraction, but with less skill across the intermediate values. This may be in large part due to issues with the reference snow fraction from MODIS, which seems to be clustered around low and high snow fraction values as shown in Figure 7 (with the exception of forested areas as shown in Figure 7a). Please consider adding some text to the first paragraph of Section 4.3, or strengthening the text on page 20 lines 10-20 to make clear how the performance of the retrieval can be influenced by the behaviour of the reference dataset.

Response: Thanks for your comment. In order to clarify the influence of reference dataset to fractional snow cover retrieval, we added the following statement in page 20 lines 28-29.
"*This is mainly because a smaller number of samples with intermediate values from the reference dataset used in the training model may not properly capture the characteristics of the surface condition with intermediate fractional snow covers*"

Figure 8: the paper would be strengthened with more emphasis on the presentation of spatial results. Figure 8 is really important, but I found it unclear, especially panel D (the sub-panels within panel D are hard to read). Why is there so much white space in panel B? Zero snow fraction needs to have a separate colour than the range of 0 to 0.3, in order to clearly show where the retrieval estimates no snow versus very low fractions of snow (e.g. 0.1 to 0.3). I suggest a clear set of maps be presented, with emphasis on a comparison between MODIS and passive microwave estimates at the continental scale (as in panels B and C) for some key events which extended the snowline.

Response: Thanks you very much for your valuable suggestion.
1) The MODIS binary snow cover image (Fig. 8A) was translated to the reference MODIS fractional snow cover (Fig. 8B) by applying the pixel filtering rule at a 15*15 pixels window that do not allow an cloudy pixel when calculating the fractional snow cover. Then it resulted in that many pixels to be masked as "fill value" (white in Figure 8).
2) We modified and clarified why the separate color map was used in here.
 "*Fig. 8 shows the comparison between our estimated fractional snow cover and the reference MODIS fractional snow cover, and more importantly, provides another perspective for snow cover identification in Section 4.4. Thus, Fig. 8B and 8C used 0.3 as the threshold of fractional snow cover to define snow-covered and snow-free area, and this was adopted through the experiments in Section 4.4*" in page 20 lines 11-14.
3) Moreover, according to your suggestions, we strengthened the description of spatial results in order to improve the legibility of each image (Fig. 8), and revised the statements as follows:
"*Apart from the scatter plots and statistical analysis, Fig. 8 shows the distribution pattern of snow cover from a spatial perspective, including MODIS composite binary snow cover (Fig. 8A), MODIS fractional snow cover (Fig. 8B), and the estimated fractional snow cover by the proposed algorithm (Fig. 8C). When the most rigorous pixel filtering rule at the 15*15 pixel window was applied (see Section 3.2), the large number of cloud covered pixels (yellow) in Fig. 8A resulted in most areas of the MODIS fractional snow cover image (Fig. 8B) being represented by a "fill value". Additionally, the number of intermediate values for MODIS fractional snow cover in winter would be much lower than the number of values near the two extreme values (0 and 1). In contrast, the estimated fractional snow cover from passive microwave brightness temperature data can provide almost complete coverage and continuous spatial information on snow cover (Fig. 8C; Fig. S-7 in the Appendix). Fig. 8 shows the comparison between our estimated fractional snow cover and the reference MODIS fractional snow cover, and more importantly, provides another perspective for snow cover identification in Section 4.4. Thus, Fig. 8B*

*and 8C used 0.3 as the threshold of fractional snow cover to define snow-covered and snow-free area, and this was adopted through the experiments in Section 4.4. This means that the pixel was identified as snow cover when fractional snow cover value was less than 0.3. From Fig. 8A – C, the spatial pattern of estimated fractional snow cover from the proposed method seems to accurately capture the distribution of snow cover from MODIS under clear-sky conditions, such as the snow-free area in most areas of North America, and snow-covered areas in northern Canada. Fig. 8D presents a specific example comparing these two fractional snow cover datasets and MODIS composite binary snow cover products in central Canada on February 27th, 2017. Based on this example, we find that our estimated fractional snow cover was capable of obtaining snow cover distribution when most of the area was covered by cloud, which was not the case for MODIS. This example also show that the extent of snowline observed in the MODIS binary snow cover image (500 m), which was the boundary between snow-covered and snow-free, was well described and exhibited by the estimated fractional snow cover (6.25 km)"* in page 20 lines 3-23.

Moreover, the estimation results comparison of fractional snow cover for MODIS and our proposed algorithm in continuous value has been shown in the supplement file:

[Figure]

Figure S-7. Comparison of the reference MODIS fractional snow cover (A) with our estimated fractional snow cover (B) in continuous value (6.25-km) on February 27th, 2017 (2017058)

Page 18 lines 3-6/Page 19 lines 27-28/Page 22 lines 1-3: the explanation for the potential over-identification of snow in the microwave retrievals (compared to the Grody product) is not convincing. The misclassification of snow extent due to non-snow scatterers (like cold deserts/frozen ground) is not a prevalent issue in North America. To better understand the statement that "the non-snow scatterer is the major source of snow cover misclassification for random forest FSC results" it would be clearer to show a map of locations where the RF classifier identifies snow and the Grody algorithm does not. This aspect needs to be explored in more detail in the final manuscript.

Response: Thanks for your comment. Although the commission error of the proposed algorithm in snow cover identification only have 0.17, we provided additional information to explain this kinds of error. As you say, the non-snow scatters (like cold deserts, frozen ground) is not a prevalent issue in North America. According to our study, we can also conclude that the snow cover misclassification effected by cold deserts and frozen ground is not prevalent issue in North America. We then specified the different source error for commission error and revised the statement as follows:

  *"The records, which were misclassified as snow cover by random forest SCA, although they are non-snow scatter components (precipitation, cold desert, and frozen ground), account for 70.1% of total misclassification records (CE = 0.17), of which 63.0% comes from precipitation, 6.4% from cold desert, and 0.7% from frozen ground"* in page 23 lines 5-8.

Following your suggestion, we first analyzed the confidence of the comparison results between Random forest SCA and

Grody's algorithm SCA, when the in-situ station observation is absent. We provided the following statistical metrics (Table A) using the data in 2017. We can see that the percentage of "True observation" for Grody's algorithm only is 24.9% when RF classifier identifies snow-covered and the Grody's algorithm does not (Condition B); inversely, it should be classified as snow-covered. If we do not use the in-situ observation as the "true" observation, we do not have high confidence to say that the detection results by our proposed algorithm in Condition B are not right. Moreover, we show an example that provides a map for different condition combinations of Random forest SCA and Grody's algorithm SCA (Fig. S-9). The inconsistencies between Random forest SCA and Grody's algorithm SCA usually occurred in the mid-latitude region, in which it has the low fractional snow cover (Figure S-7). And also we revised the statement to

*"For different results for these two snow cover mapping algorithms, we have used an example to show the inconsistencies and consistencies in mapping between the random forest SCA and Grody's algorithm SCA (Fig. S-9)"* in page 23 lines 10-11.

Table A. The effect of precipitation, cold desert and frozen ground in snow cover misclassification. FP is false positive that means it is the number of pixels that are misclassified as snow cover by Random forest FSC. $SD_{obs} = 0$ denotes snow-free measured in station, otherwise, it is snow-covered; $SC_{Grody} = 0$ denotes snow-free (precipitation, cold desert and frozen ground) determined by Grody's algorithm, otherwise it is snow-covered; $FSC \leq 0.3$ denotes snow-free cover detected by our method, otherwise, it is snow-covered.

| No. | Conditions | Observation | | Percentage of "True observation" | |
|---|---|---|---|---|---|
| | | $SD_{obs} = 1$ | $SD_{obs} = 0$ | Random Forest | Grody's algorithm |
| A | $SC_{Grody} = 0$ & $FSC \leq 0.3$ | 17435 (13%) | 116069 (87%) | 87% | 87% |
| B | $SC_{Grody} = 0$ & $FSC > 0.3$ | 60601 (75.1%) | 20063 (24.9%) | 75.1% | 24.9% |
| C | $SC_{Grody} = 1$ & $FSC \leq 0.3$ | 4379 (51.5%) | 4120 (48.5%) | 48.5% | 51.5% |
| D | $SC_{Grody} = 1$ & $FSC > 0.3$ | 80167 (90.3%) | 8575 (9.7%) | 90.3% | 90.3% |

[Figure]

Figure S-7. Comparison of the reference MODIS fractional snow cover (A) with our estimated fractional snow cover (B) in continuous value (6.25-km) on February 27th, 2017 (2017058)

[Figure]

Figure S-9. The mixed snow cover detection map for different condition combinations of Random forest SCA and Grody's algorithm SCA on February 27th, 2017 (2017058) (the meaning of A-B can refer to Table A).

**Editorial comments:**

Abstract line 23: change '0.31 million' to '310 000'

Response: Thanks. "0.31 million" was changed to "*310 000*" in page 1 line 30.

Abstract line 26: I suggest not referring to the passive microwave dataset used for comparison as 'Grody's snow mapping algorithm' in the abstract.

Response: Thanks. We changed the statement to "*There was significant improvement in the accuracy of snow cover identification using our algorithm; the overall accuracy had increased by 18% (from 0.71 to 0.84), and the omission error had reduced by 71% (from 0.48 to 0.14), when the threshold of fractional snow cover was 0.3*" in abstract.

Page 2 line 2: change 'cycles' to 'cycle'

Response: we changed "cycles" to "*cycle*" in page 2, line 11.

Page 2 line 5: 'vast number of water resources' awkward wording

Response: we rephrased the sentence to "*Snowpack also stores a huge amount of water...*" in page 2, lines 13-14.

Page 3 lines 20-25: when possible, try to use product names instead of the author names. For example, the Kelly (2009) reference refers to the NASA standard AMSR-E snow water equivalent product. The citations should be

retained, just the product names changed.

Response: Thanks you for your comment. We inquired each algorithm and tried to find their products. If the corresponding products were not found, author's name was used as the name of the algorithm. We revised the statement to "*Specifically, they involved the application of common passive microwave snow cover mapping algorithms, such as Grody's algorithm (Grody and Basist, 1996), National Aeronautics and Space Administration (NASA) Advanced Microwave Scanning Radiometer – Earth Observing System (AMSR-E) SWE algorithm (Kelly, 2009), Singh's algorithm (Singh and Gan, 2000), Neal's algorithm (Neale et al., 1990), the FY3 algorithm (Li et al., 2007), and the South China algorithm (Pan et al., 2012) ...*" in page 4 lines 4-8.

Page 3 line 28 and page 20 line 17: change 'patch' to 'patchy'

Response: the word "patch" was revised to "*patchy*" in page 4, line14 and page 24 line 6.

Page 4 line 7: change   'predict' to 'retrieve'

Response: we changed "predict" to "*retrieve*" in page 4 line 25.

Page 5 line 7: change 'America' to 'United States'

Response: We changed "America" to "*United States*" in page 5 line 26.

Page 8 line 4: not clear what is meant by 'fill'

Response: We changed to "*fill value*" in page 9 line 4.

Page 18 lines 10-14: this text is unclear and seems very anecdotal. I think it can be removed.

Response: Thank you. We removed these unclear statements.

Figure 1: Add units to the legend. Why is there negative elevation?

Response: Thanks. We updated Figure 1. The negative value is located in the lake region which is under the land surface.

[Figure]

Fig. 1 Topographic map of North America.

Figure 4: caption is not clear

Response: The caption of Fig. 4 changed to "*The performance of random forest models with increasing the size of training sample for shrub type*"

Figure 9: add x-axis label to indicate snow depth

Response: Thanks. We added x-axis label to Fig. 9

[Figure]

Figure 10: add axis labels

Response: Thanks. We updated the Fig. 10.

[Figure]

Fractional snow cover
* * *
*********************Reply to comments from anonymous reviewer 2#*********************
* * *
**REVIEWER 2**

**Overview and General Comments**

This manuscript describes a new approach of estimating fractional snow fraction from satellite-based passive microwave (PM) sensors and higher resolution MODIS snow cover estimates. The authors present different regression and machine learning type algorithms, including multi-regression, artificial neural networks (ANN), and a random forest regression technique, for estimating the PM-based snow cover fraction using the MODIS snow cover as a reference input to the algorithms along with accounting for different PM retrieval and ancillary datasets, like vegetation types. The methods are demonstrated and validated against independent in situ measurements across the region of interest (Canada and the US).

Overall, the paper includes comprehensive descriptions of the data and methods used, and detailed background and justification for the work presented. It also is within the scope and appropriate for the journal, The Cryosphere. The supplementary material does help support the overall findings in the paper. However, some of the methods and conclusions may require some revision and may not be conclusive enough as there is a limitation on the years evaluated and the wintertime period focused on. A few major and minor comments are noted in this review that hopefully help to strengthen the paper and the organization of the methods and results presented. There are a few sections that were difficult to follow and some of the English grammar and syntax was unclear.

Response: Thanks for your suggestions and positive comments. According to you suggestion and comments, we have carefully revised the manuscript and provided point-by-point response following each comment.

One downside to this study is that the authors only focused on seven years of available passive microwave and optically based snow cover observations and then just the peak snow months of January and February. Though it seems to make sense to focus only on when the snowpack is at the peak months and more spatially continuous, however, it is also worthwhile to capture the temporal and spatial heterogeneity in the accumulation and ablation seasons and more fully test the algorithms described and applied in this study. Otherwise, the algorithms are only somewhat effective for peak wintertime in US and Canada and not applicable for studies, like prescribing observational snow cover conditions in climate projection or snow-land-atmosphere climate interaction studies, which are pointed out as one primary reason to perform this present study.

Response: Thank you very much. We do agree with your comment on extending the study period to the snow cover accumulation and ablation stages/seasons for the fractional snow cover retrieval models. For this issue, we have discussed in Section 5.1 and provided the detailed discussions
*"...In this study's datasets, a greater number of records were located near the extreme values of the fractional snow cover (0 and 1). Thus, it is reasonable to use stratified random sampling (Dobreva and Klein, 2011), however, not the proportional distribution of target values suggested by previous studies (Nguyen et al., 2018; Millard and Richardson, 2015). Even in this cases, the overestimation and underestimation often occur near 0.0 and 1.0 in the training datasets (Fig. 7 A – D) and evaluation datasets (Fig. 7 a – d), respectively. This is mainly because a smaller number of samples with intermediate values from the reference dataset used in the training model may not properly capture the characteristics of the surface condition with intermediate fractional snow covers. Therefore, it is necessary for future studies to increase the amount of samples by extending the study period to the snow accumulation and snow ablation stages (Xiao et al., 2018), where there is much more shallow snow and "patchy" snow cover. Another option is using data from multi-source sensors to generate reference snow cover data (e.g., Sentinel -1 Synthetic Aperture Radar data). By doing this, the proportion of fractional snow cover values in*

*the training sample may be distributed as evenly as possible (Colditz, 2015; Jin et al., 2014; Lyons et al., 2018)"* from page 23 line 28 to page 24 line 9.

In fact, the same idea on "It is also worthwhile to capture the temporal and spatial heterogeneity in the accumulation and ablation seasons and more fully test the algorithms described and applied in this study" has been one major task of our ongoing work. Specifically, it is to establish different fractional snow cover retrieval models on different snow cover stages (snow cover accumulation stage, snow cover stabilization stage and snow cover ablation stage), and to analyze the spatiotemporal variation characteristics of the estimated fractional snow cover.

Also, in relation to the timeframe of the training and validation data years, only having one year to perform the validation seems quite limiting, as a given year can be hard to note overall performance given snow cover can vary greatly from year to year (e.g., snow drought conditions). This is somewhat reflected in Figure 7 (right column panels), which show how highly variable and not as predictable in the validation year (2017). Please explain why a longer period of record is not used, e.g., 2002-2019 (Terra+Aqua MODIS combined) and the passive microwave combined product by Brodzik et al. (2018), to perform the training and validation period. Perhaps, use Water Years (WY) 2002-2013 for training and WY 2014-2018 for validation?

Using only one year for testing and a second year for validation is very limiting for this study, and it is highly recommended for additional years to be included. Also, for the four different approaches of estimating the fractional snow cover from passive microwave should have longer evaluations performed in this context as the summary of the results would be inconclusive for one year of validation.

Response:Thanks for your comments and suggestions. In the absence of available published materials on fractional snow cover estimation from passive microwave data, the first emphasis of this study should explore the possibility of estimating fractional snow cover estimation from passive microwave brightness temperature data. Therefore, we conducted a series of experiments with 8 years data (January and February only) to demonstrate the feasibility of estimating fractional snow cover from passive microwave data, as described in Section 6 (page 26 lines 4-11)
"*Numerous studies have investigated the relationship between common snowpack physical properties (e.g., snow depth and water equivalent) and passive microwave brightness temperature at different frequencies and polarizations (Chang et al., 1987; Dietz et al., 2011; Kim et al., 2019; Xiao et al., 2018). Unlike many previous studies, this study innovatively used passive microwave data to directly estimate fractional snow cover. The results showed that it is possible to directly obtain an estimated fractional snow cover with high accuracy from high-spatial-resolution passive microwave data (6.25 km) under all weather conditions. Further detailed study on the use of high spatial resolution passive microwave data for fractional snow cover estimation presents itself as an interesting research direction for the development of the studies on fractional snow cover estimation*". Overall, this study has basically achieved its preset goals.

Moreover, at the beginning of our experiment, we also tested and validated the performance of fractional snow cover retrieval model with the remaining data of 2011-2016 (besides the dataset used for training samples); its conclusion is consist with that of the current experiment (using a single year of data), and the accuracy indexes (MAE and RMSE) are not significantly different. To make sure that each experiment is completely independent, we then gave up the above experimental design and adopted that the data of different years were used in different phases. As a basis of estimating fractional snow cover from passive microwave data, there will be a lot of researches to carry out in future studies, such as to apply this algorithm to other study region and other study period, to improve the fractional snow cover retrieval algorithm, to generate a high accuracy product for change characteristics analysis of snow cover area.

Furthermore, one issue that has to be explained in detail is the use of the data in this study. As to Fig. 7, the major reason for the relatively even distribution of the data used in the left column panels with capital letters (A-D) is that these training data are obtained by applying a stratified random sampling strategy in the 6 years total available data (2011-2016; January and February). Distinct from the training datasets, the testing dataset and evaluation dataset cover 2010 (Fig 6; Fig S-4, 5, 6 in

the Appendix) and 2017 (Fig. 7), respectively. Through analyzing the distribution of the fractional snow cover datasets in 2010 and 2017, we found that more than 70% of the value are near 0 and 1. This feature also can be noted in all the fractional snow cover data available during 2011-2016.

However, although the stratified random sampling strategy is applied to the 6 years of data to select the training data, these training datasets over four types of land cover are not evenly distributed in each sub-interval between 0 and 1 (Fig 7A-7D). Especially, the number of intermediate fractional snow covers in winter would be much lower than the number of fractional snow covers near the two extreme values (0 and 1). This indicates that if the study period only increases the number of years without extending the study period to the other two seasons (autumn and spring), the study period cannot provide a satisfactory data set for training samples (and testing and evaluation samples), of which the distribution of each sub-interval should be very even. That is because there are many shallow snow and "patchy" snow cover in autumn and spring season and it can provide more diverse values of fractional snow cover. Thus, the best and the most effective solution is to extend the study period to the other two seasons and not only to increase several years of data in winter. Actually, we have realized the importance of expanding the study period to the other two seasons and discussed it in Section 5.1 as follows:

"*... In this study's datasets, a greater number of records were located near the extreme values of the fractional snow cover (0 and 1). Thus, it is reasonable to use stratified random sampling (Dobreva and Klein, 2011), however, not the proportional distribution of target values suggested by previous studies (Nguyen et al., 2018; Millard and Richardson, 2015). Even in this cases, the overestimation and underestimation often occur near 0.0 and 1.0 in the training datasets (Fig. 7 A – D) and evaluation datasets (Fig. 7 a – d), respectively. This is mainly because a smaller number of samples with intermediate values from the reference dataset used in the training model may not properly capture the characteristics of the surface condition with intermediate fractional snow covers. Therefore, it is necessary for future studies to increase the amount of samples by extending the study period to the snow accumulation and snow ablation stages (Xiao et al., 2018), where there is much more shallow snow and "patchy" snow cover. Another option is using data from multi-source sensors to generate reference snow cover data (e.g., Sentinel -1 Synthetic Aperture Radar data). By doing this, the proportion of fractional snow cover values in the training sample may be distributed as evenly as possible (Colditz, 2015; Jin et al., 2014; Lyons et al., 2018)*" from page 23 line 28 to page 24 line 9.

Some of the methods sections are hard to follow, though the authors provide many details there and in the Supplemental material. For example, Section 3.3.1 of "Selecting input variables" was at times hard to follow and why each scenario was selected. Improving the organization of the sections to flow better in terms of their logic and why different experiments were performed would be helpful for the overall background and discussions of this study.

The English grammar and syntax used require additional review and editing by editorial services to help correct these issues before resubmitting. A few suggested corrections are offered below in the technical corrections section.

Response: Thanks for your positive comment to improve our manuscript. The revised manuscript has been proof read by a native English speaker. Additionally, we clarified the background of the variables selection and setting for each scenarios, and revised the statement about why different experiments were performed in Section 3.1 (from page 10 lines 19 to page 11 line 14).

"*A decision tree was established using all variables shown in Scenario -1 (Table 1), and was utilized to compare with five scenarios in terms of prediction performance and efficiency. Note that these 19 input variables were determined by using the Correlation Attribute Evaluation method in the Waikato Environment for Knowledge Analysis 3.8.3 (WEKA) data mining software. This method evaluates the worth of the attribute by measuring the correlation between the attribute and the target (Frank et al., 2004; Eibe Frank, 2016). The brightness temperature and its linear combination can also directly be used to detect snow cover based on Xu et al. (2016) study; thereby, Scenario -2 only contained brightness temperature and its linear combination without consideration to the effects of location and topographic factors. Wiesmann and Mätzler (1999) reported that V and H polarizations were dominated by scattering and snow stratigraphy, respectively. Thus, Kim et al. (2019) only assimilated V polarization with an ensemble snowpack model to estimate snow depth. Therefore, in Scenario -3, we attempted to evaluate the performance of the established retrieval model by only using the brightness temperature in 19, 37 and 91 GHz*

*(V polarization) based on Wiesmann and Mätzler (1999) and Kim et al. (2019). In Scenario -4, we used similar input variables to those used for snow depth estimation in Xiao et al. (2018), and examined whether these same parameters can or cannot estimate the fractional snow cover. In Scenario -5, unlike the variables used in Scenario -4, we attempted to use the basic input variables coupled with the brightness temperature linear combination for fractional snow cover retrieval.*

*There are other variable selection strategies based on the importance rank when using random forest method. For example, Mutanga et al. (2012) implemented a backward feature elimination method to progressively eliminate less important variables, whilst Nguyen et al. (2018) summarized the grade of the variable and selected the top eight important variables as the input variables in the training model. Similarly, this study assessed the importance of input variables on four land cover types using the same size of the training sample (15 000) (Xiao et al., 2018). We then counted the number of times of each variable that was ranked in the top nine important variables (summarized in Table S2, Appendix), which were then used as the input variables for Scenario -6 (listed in Table 1). By assessing the performance of models established by these six scenarios, an optimal combination of input variables for the fractional snow cover retrieval model may be selected (see Section 4.1.1). All input variables were normalized to [0, 1].”*

**Specific Comments**

Abstract: The authors introduce "Grody's snow cover mapping algorithm" towards the end of the abstract without any other background. Perhaps they could provide one introductory phrase on this algorithm within the abstract to give more context.

Response: Thanks for your comment. We revised the original sentence to *"There was significant improvement in the accuracy of snow cover identification using our algorithm; the overall accuracy had increased by 18% (from 0.71 to 0.84), and the omission error had reduced by 71% (from 0.48 to 0.14), when the threshold of fractional snow cover was 0.3"* in Abstract

Page 2, Lines 9-10: The authors mention that snow cover data from station measurements are "time-consuming, [and] cumbersome,". What do the authors mean by these adjectives? Please clarify here. Any dataset, including satellite, requires time and careful derivation of the final product. However, in situ snow cover data are spatially discontinuous and require more time to maintain.

Response: Thank you. According to your suggestion, we clarified the sentence to
*"Snow cover data is typically obtained from meteorological stations or in-situ manual measurements, which is spatially discontinuous and labor intensive"* in page 2 lines 17-18.

Page 6, lines 11-12: Would like to point out here that North America includes Mexico as well. The authors should specify that their study domain spans the continental U.S. and Canada only.

Response: Thank you. We specified the study domain definition and revised the original sentence to
*"Fig. 1 shows the elevation pattern for North America, limited to Canada and United States in this study."* in page 7 lines 3-4.

Page 7, lines 10-11: Authors state here that "to the best of our knowledge, there are no researchers have developed fractional snow cover ⋮ ⋮ ⋮ using passive microwave data." Please take a look at the following references and cite appropriately:

Foster, J.L., D. K. Hall, J. B. Eylander, G. A. Riggs, S. V. Nghiem, M. Tedesco, E. Kim, P.M. Montesano, R. E. J. Kelly, K. A. Casey and B. Choudhury (2011): A blended global snow product using visible, passive microwave and scatterometer satellite data, International Journal of Remote Sensing, 32:5, 1371-1395, DOI: 10.1080/01431160903548013

Response: Thanks for your comment. The study carried out in Foster et al. (2011) was to yield a blended snow cover product with a 25-km resolution by combining MODIS snow cover product, AMSR-E snow water equivalent product, and QSCAT data, which have several parameters including snow cover extent, snow water equivalent, fractional snow cover, onset of snowmelt and areas of snow cover that are actively melting. We find that there is essential difference between Foster's study and our work in fractional snow cover estimation. In contrast, current study devoted to retrieving fractional snow cover from passive microwave brightness temperature at 6.25-km resolution, which means that the estimated results are based on passive microwave data. We changed it to "*Second, to the best of our knowledge, there are few attempts to directly develop fractional snow cover from passive microwave data*" in page 8 lines5-6

Page 8, lines 24-27: It would be helpful here to provide a lead in sentence to introduce your first two equations.

Response: Thanks your valuable suggestion. We revised and clarified the description about these two equations as follows (in page 9 line 16-26):
"*1) Combining snow cover images from two sensors on a given day: the first simple filter was applied under the assumption that snowmelt and snowfall did not occur within the two sensor observations. Whether a pixel in Terra ($S_t^{Aqua}$) or Aqua ($S_t^{Terra}$) snow cover image in a given day (t) was observed as snow cover or snow-free, the pixel in the output image (MCD10A1) was assigned the same ground status (shown in Eq. 1). The results showed about 3% of cloud cover was removed compared to MOD10A1 (Gafurov and Bárdossy, 2009).*

*2) Short-term temporal filter: if the status of a pixel in the input image (MCD10A1) in a given day (t) was cloud and both the preceding (t - 1) and succeeding (t + 1) days were snow-covered (or snow-free), the pixel in the output image (MCTD10A1) in the given day (t) was assigned as snow-covered (or snow-free) (summarized by Eq. 2). Compared to the first filter, this short-term temporal filter may markedly reduce the number of days (10% ~ 40%) for cloud coverage and increase the overall accuracy of snow cover detection (Gafurov and Bárdossy, 2009; Tran et al., 2019)...*" in page 8 lines 22-30.

Page 8, last line: "Calculation areas should be in a larger feet :::" What is meant here by "feet"? It does not seem to make sense to use this word here, but perhaps "footprint area" makes more sense? Please correct.

Response: Thanks for your suggestion. We corrected the sentence to
"*Calculated areas should be a larger footprint area than the pixel resolution to avoid MODIS geolocation uncertainties...*" in page 10 lines 1-2.

Page 11, lines 2-3: MODIS Collection 5 products are considered older and not "current", as they have been replaced by Collection 6. Recommend removing "current" here.

Response: Thank you. We removed "current" and revised the sentence to "*This type of regression method has been applied in generating the standard MODIS fractional snow cover product Collection 5...*" in page 12 lines 12-13.

Subsection 3.4.1: The authors discuss both the linear and multi-linear regression methods here, which makes the discussion confusing to follow. They then have the reader refer to the Supplementary material for more information. It is recommended that the authors better describe in this subsection how the "linear regression" is applied. Was it based on the equations in Salomonson and Appel (2004) or new linear equations and parameters derived for the four different vegetation categories? Please try to better organize and explain this linear method in this subsection.

Response: Thanks for your comment and suggestion. We revised the statement about linear regression method as follows:
"*For optical remote sensing studies, there is a classical and general linear regression method used to estimate the sub-pixel*

*snow cover area in medium- to high-spatial-resolution image. This only involve the relationship between NDSI and fractional snow cover derived from high-resolution snow cover maps (Salomonson and Appel, 2004; Salomonson and Appel, 2006). This type of regression method has been applied in generating the standard MODIS fractional snow cover product Collection 5. Similarly, the multiple linear regression method was used as a reference method in this study to estimate fractional snow cover based on passive microwave data. The inputs were the same as the other three methods in this study...*" in page 12 lines 8-15.

Page 14, lines 10-11: Please provide citations and references where possible for the metrics, especially Cohen's kappa coefficient and the F1 score.

Response: Thanks. We add the citation for the metrics and correspondingly the sentence was revised to
"*Six accuracy assessment indices were used for the analysis of snow cover detection capability (Liu et al., 2018; Gascoin et al., 2019); overall accuracy (OA), precision (that is, a positive prediction value), recall, specificity (that is, the true negative rate), F1 score (Zhong et al., 2019), and Cohen's kappa coefficient (Foody, 2020).*" in page 16 lines 4-7.

Page 15, Lines 11-12: Authors indicate here that their "Scenario-6" variable sensitivity case "generated the worse performance, with the low R, the great MAE and RMSE". When looking at Table 4 results, Scenario-6 appears to perform rather well overall. Perhaps it would help if the authors specify here that of the Scenarios of 1, 4-5 and 6, Scenario-6 performs the "worst". It is also recommended to change the last part of that sentence to: " this scenario's setting had the third worst performance with lower R values and higher MAE and RSME values."

Response: Thank you very much for your suggestion. We revised the statement to
"*Moreover, when compared to Scenarios-1, 4, 5, the setting in Scenario-6, where input variables were selected by importance, had the third poorest performance, with a low R, and a high MAE and RMSE*" in page 17 lines 13-16.

Page 15, line 31 to top of Page 16: Make "Figure" plural and change the last part of this sentence to something like: "show that this finding was not coincidental." This sentence is a bit hard to understand in what is meant by "not coincidental". Please elaborate or better explain the meaning here.

Response: Thanks. We clarified the statement and revised to
"*Interestingly, the 0.3% training sample size had the shortest modeling time of the three sample size (Fig. 4); Figs. S-1, 2, 3 also exhibit similar findings on modeling time.*" in page 18 lines 5-7.

Page 16, line 29: Please clarify here what is meant by "neglected to assess the rationality of estimated value : : :". Are you referring to the out-of-bounds events that occur in the other methods, other than the random forest approach and that that "rational" was not well checked?

Response: Thanks for your suggestion and comment. We revised the sentence to
"*Previous studies have generally neglected the analysis and evaluation of whether the estimated value is out-of-range*" in page 19 lines 8-9.

Page 21, line 1: Authors state that only a few studies validate the accuracy of MODIS snow cover products in forested areas. Actually, there are several in addition, including:

Arsenault, K.R., P.R. Houser and G. J.M. De Lannoy, 2014: Evaluation of the MODIS snow cover fraction product, Hydro. Proc., 30, 3, pps. 980-998. https://onlinelibrary.wiley.com/doi/full/10.1002/hyp.9636

Kostadinov, T. S., and T. R. Lookingbill, 2015: Snow cover variability in a forest ecotone of the Oregon Cascades via MODIS

Terra products, Rem. Sens. Env., 164, pps. 155-169. https://www.sciencedirect.com/science/article/pii/S0034425715001303

Response: Thank you very much for your suggestion. We added the suggested literatures and revised the statement to *"Several studies have validated and evaluated the accuracy of MODIS snow cover products, particularly in forested areas (Parajka et al., 2012; Zhang et al., 2019; Arsenault et al., 2014; Kostadinov and Lookingbill, 2015)"* in page 24 lines 20-22.

Page 22, lines 14-16: The first statement here about the "strong limitations in the understanding of physical mechanism" is a bit hard to understand. Are the authors referring to the underlying physics and characteristics that relate the fractional snow to the signature of the passive microwave bright temperature responses? Perhaps, it might be better to frame these concluding statements more in that way vs. "mechanisms".

Response: Thanks for your constructive suggestion. We clarified and revised the sentence to *"However, it also contains significant limitations in understanding the physics that relates fractional snow cover to the signature of passive microwave brightness temperature (Cohen et al., 2015; Che et al., 2016). Future studies need to use physical snowpack models and radiation transfer theory to explore the physical mechanistic relationships between microwave brightness temperature and fractional snow cover (Pan et al., 2014)"* in page 26 lines 16-20.

Table 1: In the row of references, does the Xiao et al. (2018) paper cover both Scenario-4 and -5 columns in the table? If so, it might be helpful to specify this in the body of the paper.

Response: Thanks. The Xiao et al. (2018) study only cover the variables used in Scenario-4, not in Scenario-5. Thus, we did not provide the related reference for Scenario-5.

Figure 8: In panel A, more binary MODIS snow cover present (e.g., large green pixeled areas in Canada), but that does not seem to get translated over to panel B for the fractional MODIS snow cover (mostly filled in with no fractional values). Please explain why most of the derived MODIS snow cover fraction is removed here, especially over Canada? Also, for the MODIS snow fractional product, there is no fractional snow representation between 0.3 and 0.8, the other two categories shown in panel B. What is happening here in that regard – no fractional snow within 0.3 and 0.8 at any noticeable gridcells? Please provide an explanation in the text as well.

Response: Thanks for your suggestions to improve our manuscript.
1) The MODIS binary snow cover image was translated to the reference MODIS fractional snow cover by applying the pixel filtering rule at a 15*15 pixels window that do not allow an cloudy pixel when calculating the fractional snow cover. Therefore, many pixels (6.25-km) were masked as "fill value" (white in Figure 8).
2) Each category (0-0.3; 0.3-0.5; 0.5-0.8; 0.8-1) was exhibited in MODIS fractional snow cover image (Fig. 8B), just the difference in the amount of pixels. The intermediate values of fractional snow cover usually can be found at the edge of the two extreme values (0 and 1).
Based on your suggestion, we revised the description about Fig. 8 as follows:
*"Apart from the scatter plots and statistical analysis, Fig. 8 shows the distribution pattern of snow cover from a spatial perspective, including MODIS composite binary snow cover (Fig. 8A), MODIS fractional snow cover (Fig. 8B), and the estimated fractional snow cover by the proposed algorithm (Fig. 8C). When the most rigorous pixel filtering rule at the 15*15 pixel window was applied (see Section 3.2), the large number of cloud covered pixels (yellow) in Fig. 8A resulted in most areas of the MODIS fractional snow cover image (Fig. 8B) being represented by a "fill value". Additionally, the number of intermediate values for MODIS fractional snow cover in winter would be much lower than the number of values near the two extreme values (0 and 1). In contrast, the estimated fractional snow cover from passive microwave brightness temperature data can provide almost complete coverage and continuous spatial information on snow cover (Fig. 8C; Fig. S-7 in the Appendix). Fig. 8 shows the comparison between our estimated fractional snow cover and the reference MODIS fractional snow cover, and more importantly, provides another perspective for snow cover identification in Section 4.4. Thus, Fig. 8B*

*and 8C used 0.3 as the threshold of fractional snow cover to define snow-covered and snow-free area, and this was adopted through the experiments in Section 4.4. This means that the pixel was identified as snow cover when fractional snow cover value was less than 0.3. From Fig. 8A – C, the spatial pattern of estimated fractional snow cover from the proposed method seems to accurately capture the distribution of snow cover from MODIS under clear-sky conditions, such as the snow-free area in most areas of North America, and snow-covered areas in northern Canada. Fig. 8D presents a specific example comparing these two fractional snow cover datasets and MODIS composite binary snow cover products in central Canada on February 27th, 2017. Based on this example, we find that our estimated fractional snow cover was capable of obtaining snow cover distribution when most of the area was covered by cloud, which was not the case for MODIS. This example also show that the extent of snowline observed in the MODIS binary snow cover image (500 m), which was the boundary between snow-covered and snow-free, was well described and exhibited by the estimated fractional snow cover (6.25 km).* " in page 20 lines 3-23..

Finally for Figure 8, it would be helpful to assign a different color and category for the non-snow pixels (at fractional value of 0.) in panels B and C to better discriminate the non-snow areas from the snow-based areas. Currently, snow-free pixels are lumped in with the low snow fraction category of 0 to 0.3.

Response: Thanks for your comment. In this study, we clarified why 0.3 is adopted as the threshold of fractional snow cover. "*Fig. 8 shows the comparison between our estimated fractional snow cover and the reference MODIS fractional snow cover, and more importantly, provides another perspective for snow cover identification in Section 4.4. Thus, Fig. 8B and 8C used 0.3 as the threshold of fractional snow cover to define snow-covered and snow-free area, and this was adopted through the experiments in Section 4.4*" in page 20 lines 11-14.

In addition, a comparison example of the reference MODIS fractional snow cover with our estimated fractional snow cover in continuous value (Figures S-7 vs Fig 8.) in the supplement have been provided to show the continuous change characteristics of fractional snow cover in the Norther America on February 27th, 2017 (2017058).

[Figure]

Figure S-7. Comparison of the reference MODIS fractional snow cover (A) with our estimated fractional snow cover (B) in continuous value (6.25-km) on February 27th, 2017 (2017058)

Fig. 11: This is a nice figure that summarize and present these results well.

Response: Thanks for your positive comments.

**Technical corrections**

Page 2, line 25: Please specify what "FY" stands for in "FY series sensors".

Response: We revised the sentence to "...*Fengyun (FY) series sensors....*" in page 3 line 7.

Page 3, line 25: Awkward phrasing here: "To unite resolution, : : :" Perhaps try: " To be at a common resolution, : : :"

Response: Thanks for your suggestion. We revised the sentence to "*To achieve a common resolution, bilinear interpolation was used to aggregate the 3.125 km spatial resolution data to 6.25 km*" in page 5 lines 15-16.

Page 5, line 7: Recommend here to separate the two phrases here with either a semi-colon (between "collected" and "all available") or place the conjunction "and" after the comma.

Response: Thanks. We revised the sentence to "*... Canada and United States were collected, and all available records from these sites were included in this study.*" in page 5 lines 26.

Page 6, line 6: Please specify what "ETOPO1" stands for.

Response: Thanks. The elevation dataset's name is called ETOPO1 refer to the website ([https://data.nodc.noaa.gov/cgi-bin/iso?id=gov.noaa.ngdc.mgg.dem:316](https://data.nodc.noaa.gov/cgi-bin/iso?id=gov.noaa.ngdc.mgg.dem:316)), and do not have more full name for these characters.

Page 6, line 11: Add citation and reference for "ArcGIS 10.5" software.

Response: We cited the related reference and revised the sentence to "*The slope and aspect data were obtained from ETOPO1 data by ArcGIS 10.5 (Buckley, 2019)*" in page 7 lines 2-3.

Page 6, line 17: Replace "heterogeneous" with the noun, "heterogeneity".

Response: Thanks. We replace "heterogeneous" with "*heterogeneity*" in page 7 line 9.

Page 7, line 7: MODIS misspelled here as "MODSI".

Response: Thanks. We changed "MODSI" to "*MODIS*" in page 8 line2.

Page 7, line 31: Remove "with" before "accurate".

Response: Thank you. We removed "with" before "*less accurate*" in page 8 line 30.

Page 9, line 21: Either replace the semicolon with a period, or make the word, "Thereby", lower-case.

Response: Thanks. We changed "Thereby" to "*thereby*" in page 10 line 25.

Page 9, line 27: Change the "not" in this line to "cannot". Also on that same line, the word use of "Correspondingly" here

does not seem to make sense.

Response: Thanks. We revised the sentence to "*... can or cannot estimate the fractional snow cover. In Secenatio-5 ....*" in page 11 line 1.

Page 10, line 5: Make "variable" plural here in "an optimal combination of input variables".

Response: Thanks. We changed "variable" to "*variables*" in page 11 line 13.

Page 13, line 6: "researches" should be changed to "researchers".

Response: Thanks you. We changed "researches" to "*researchers*" in page 14 line 23.

Page 18, line 5: Remove "be" before "misclassified" and change "into" to "as". Also, please remove the phrase, "As we all know", and change the start of the second sentence there to: "Permafrost is known to be widely distributed in the northern part of…"

Response: Thank you. We removed "be" before misclassified and changed "into" to "as", accordingly, the sentence changed to "*... these scatters were easily misclassified as snow cover in less snow cover conditions...*" in page 21 lines 7-9. And we have removed the description "Permafrost is known to be widely distributed in the northern part of…" based on the revised needs

Page 20, line 4: Change "researches" to "studies".

Response: We changed "researches" to "*studies*" in page 23 line 20.

Page 21, lines 23-24: Change "were" to "was" in relation to "The accuracy of the proposed algorithm was further : : :".

Response: Thank you. The sentence was changed to "*The results of the evaluation using the reference fractional snow cover data in 2017 showed that ....*" in page 25 lines 19-20.

Table 2 caption: "unite" should be "unit", and "clod desert" should be "cold desert".

Response: Thank you very much. We changed "unite" to "*unit*" and modified "clod desert" to "*cold desert*" in Table 2

Figure 7: The use of the capitalized and lower-case plot labels is fine but not conventional. Would it make more sense to simply use, "A, B" then "C, D", etc., for the paired columns?

Response: Thanks for your comment. Horizontally, the capital letters indicate the results in the training stage, while the lowercase letters represent the results in evaluation stage; from the vertical perspective, the results in two stages in each row are the same type of land cover which was represented by the same level of letters that are easily distinguished. The caption of Fig. 7 was modified to "*... Left column with capital letters is the results in the training stage (A-D); right column with lowercase letters is the results in the evaluation stage (a-d).*"

---

## Author Response (AR2)

Dear editor,

We would like to thank you and the reviewer for the insightful comments and suggestions to improve our manuscript. We have carefully revised the manuscript according to the reviewer's suggestions and comments, and provide point-by-point response following each comment and suggestion.

In the following, reviewer comments are given in black and responses are given in blue (the revised sentence of the manuscript was set in *italics*). The corresponding changes have been made in the revised paper with track changes. We mainly did the following modification in the updated manuscript:

1) We added the Fig. 12 that described the snow cover mapping results comparison between random forest SCA and Grody's algorithm SCA;

2) We added another example of the comparison between MODIS and the estimated fractional snow cover (Figure S-8 in the Appendix) on January 10th, 2017 (2017010).

3) We added another two years data (2008-2009) to validate the snow cover mapping capability in Section 4.4.

4) The figures (Fig. 2 and Fig. 9) were updated.

We think the revised manuscript has addressed all the reviewers' comments and hopefully it is now suitable for publication in The Cryosphere

Sincerely,

Xiongxin Xiao

Response to Reviewer #2

Overview and General Comments:

The revised manuscript is much improved in terms of its organization, explanations of the results and English writing, grammar and syntax. The authors have addressed sufficiently many of the reviewers' original comments and suggestions, however, there are a few areas that may require additional attention and a couple of concerns that were not fully addressed.

The authors have conducted a comprehensive study in considering the inputs and characteristics associated with passive microwave retrievals and algorithms to estimate snow cover fraction. However, it focuses still on only two winter months and is limited by the number of years used in the training, testing and validation, using the period from 2008-2017 (in relation to SSMI/S sensor F-16). Selecting just the mid-winter months of January and February to demonstrate the application of the random forest regression approach as to determine snow cover fraction does limit the study to mostly dry snow events, and ignores the ablation and accumulation seasons, which are understandably harder to train and estimate and may require more time to address. If it is not too costly (as based on their computational timing estimates shown in the paper), could the authors include a couple of other months, at least for the random forest regression and Grody SCA algorithm methods, to see how they perform? For example, include December (reflecting more accumulation phase) and March (more ablation phase processes) to see how well the algorithms perform for different conditions and intermediate snow levels, especially the March case.

Response: Thanks for your comment and constructive suggestion to enhance our manuscript.

Although we have finished searching the theory and method in fractional snow cover retrieval from passive microwave brightness temperature, the presentation of retrieval results in other months following this study's method are hard to finished in such limited time. As you said, snow properties between snow accumulation phase and snow ablation phase exhibit great differences. If you want to generate fractional snow cover results for other months, you need to respectively rebuild a new model suitable for the snow ablation phase or other snow cover phases. For example, to establish fractional snow cover model for snow ablation phase, the daily MODIS and passive microwave data downloading, data preprocessing, information extracting, model establishing, and results analyzing would take at least 4 months.

As an explorative experimental study, this study needs to explore and demonstrate the feasibility of using passive microwave data to estimate fractional snow cover, because we did not know whether the passive microwave data can be used to retrieve the sub-pixel snow cover area of passive microwave data. In addition, this study has emphasized why we only used the data in January and February "*To explore the feasibility of estimating fractional snow cover using passive microwave data, this study only selected January and February of 2008 – 2017 as the study period in here*" in page 4 lines 21-23.

In addition, we have discussed the disadvantage of the study period only limited in January and February in Section 5.1 "*Therefore, it is necessary for future studies to increase the number of samples by extending the study period to the snow accumulation and snow ablation stages (Xiao et al., 2018), where there is much more shallow snow and "patchy" snow cover. Another option is using data from multi-source sensors to generate reference snow cover data (e.g., Sentinel -1 Synthetic Aperture Radar data). By doing this, the proportion of fractional snow cover values in the training sample may be distributed as evenly as possible (Colditz, 2015; Jin et al., 2014; Lyons et al., 2018).*"

During the long review of this manuscript, we have further explored other issues in fractional snow cover retrieval from passive microwave data (Xiao et al., 2020). It should be pointed out that the data and methods used in our recently completed study are quite different from those in this study. Our latest research also explored the different performance of fractional snow cover retrieval from passive microwave data in different months (September to May). The results, which could answer your question, found that the best prediction performance of retrieving fractional snow cover is in January, the worst is in May, and the accuracy of the estimated fractional snow cover is varying with snowpack properties (Xiao et al., 2020).

Reference:

Xiao, X., He, T., Liang, S.: Retrieval of fractional snow cover from passive microwave data coupled with microwave radiation transfer model and machining learning method, in preparation, 2020.

Though the authors point out (page 18-L11, page 21-L12, and page 23-L7) that the amount of intermediate fractional snow cover values is much lower, since they focus on peak winter months, the question arises then -- does their random forest regression snow cover retrieval approach, that was trained and tested for this study, adequately predict intermediate snow cover fraction values?

Response: Thanks for your comment. At the beginning of this study, we only using the data during 2014 - 2016 to provide training sample selection dataset. Due to the study time focus in winter with more cloud contamination in optical remote

sensing image, we found the amount of intermediate fractional snow cover values in the training sample in is not enough and also doubted the robust of the established retrieval models. Then, we increased the number of intermediate values in the training sample data set by extending the study period to 2011. In order to make the training samples more representative, we associated with the stratified sampling method to randomly select enough training sample from the expanding datasets. As a result, we can find that the fractional snow cover values have a relatively good distribution in Fig. 7A - 7D and have a reliable predictive capability in fractional snow cover (Fig. 6 and 7). In addition, the verification experiments with in-situ measurement show that our proposed fractional snow cover retrieval method has high ability in snow cover mapping with high overall accuracy (OA = 0.85; Fig 11)

When looking at Fig. 6, which compares the reference (MODIS) with the random forest (RF) passive-microwave snow cover estimates for forest land type, the reader sees some of the intermediate snow cover fraction values captured across the RF-based estimates. However, when one looks at the Supplemental Figs. 4-6, for the other three land types (prairie, shrub and bare soil), most of all the values congregate either between 0.8 and 1.0 or near 0.2, for the RF estimated values, with only a few points near 0. With not knowing the distribution of land cover types, it is hard to know how many "intermediate" values are captured mainly due to vegetation canopy presence. For the other land cover types, intermediate values are not well captured, possibly reflecting mostly dry snow winter conditions, thus not making the approach tested a truly "fractional" snow cover product.

Response: Thanks for your comment. As you seen, the distribution of fractional snow cover values shows great varieties over different land cover types. Due to cloud contamination and rigorous filter rules, not so much number of intermediate fractional snow cover values were obtained for our further analysis. One important fact should be pointed that the presentation of evaluation results of predicted values in this paper heavily is dependent on the number of reference MODIS fractional snow cover values. When compared to the other three methods, the random forest-based algorithm has the best performance in fractional snow cover estimation. From this explorative experiment, we have demonstrated that it is feasible using passive microwave remote sensing data to estimate fractional snow cover.

To generate more robust and high accuracy fractional snow cover products, there are many issues that need to pay more attention to solve and investigate in future work, e.g., 1) the application of the proposed method in other study periods (seasons or months); 2) the different sensitivity to snow cover extent for optical and passive microwave satellite sensor in coarse resolution; 3) the different responses of optical and passive microwave satellite sensors to snow cover interception by forest or other vegetation (Lv and Pomeroy, 2019); 4) the effect of different land cover types in sub-pixel (Salminen et al., 2018); 5) filling data gape of passive microwave data for generating full coverage fractional snow cover products, etc.

Reference:

Lv, Z., & Pomeroy, J. W.: Detecting intercepted snow on mountain needleleaf forest canopies using satellite remote sensing. Remote Sensing of Environment, 231, 111222, 2019.

Salminen, M., Pulliainen, J., Metsmki, S., Ikonen, J., Heinil, K., & Luojus, K.: Determination of uncertainty characteristics for the satellite data-based estimation of fractional snow cover. Remote Sensing of Environment, 212, 103-113, 2018.

The authors stated in their introduction section that "there is an urgent need to acquire snow cover area within a sub-pixel to provide accurate snow cover information", as a main motivation for performing this study. I feel the authors do try to capture

the fractional nature of snow cover with their algorithm and passive microwave estimates. However, not many spatial map examples are provided to show how well the overall the passive microwave fractional snow cover estimates perform for the mid-winter months (Jan-Feb) against the MODIS-based fractional estimates to demonstrate this. It would be helpful if there was at least one other date provided and shown (e.g., besides Feb 27, 2017, in Fig 8) and perhaps a date that the MODIS product had less cloud coverage and more intermediate snow present.

Response: Thanks for your suggestion. The reference MODIS fractional snow cover data could not provide snow cover area information in most cases due to cloud contamination while the passive microwave-based fractional snow cover data can offer the users and researchers continue snow cover area information in continental scale (Fig. 8 and Figure S-7 in the Supplements). Following your suggestion, we added a spatial map example of reference MODIS fractional snow cover with relative much mediate values and the estimated fractional snow cover on January 10th, 2017 (2017010) in Supplement (Figure S-8).

[Figure]

Figure S-8. Comparison of the reference MODIS fractional snow cover (A) with our estimated fractional snow cover (B) at 6.25-km spatial resolution on January 10th, 2017 (2017010)

Thank you for addressing and explaining further the reason for the "binary" nature of the aggregated (15x15) MODIS snow cover gridcells compared (Figure 8B panel), due to the "rigorous" screening of only including snow or non-snow pixels within the 15x15 aggregate. Also, the side analysis (not included in the paper) is helpful to see if you were to relax that constraint by allowing a 5% amount of pixels to be factored in (that are cloud or inland water), and how that translated into more "intermediate" valued gridcells, ranging from 0.3 and 0.8. With the original rigorous screening process still applied in the paper, at least shown in Fig. 8, the MODIS snow cover reference acts more as a "binary" snow cover, which does not seem to reflect fractional snow cover here. By selecting such thresholds (e.g., <= 0.3 cutoff as no-snow) and rigorous constraints, this may eliminate many pixels that could be used in the training process, even though the design is to utilize the most accurate input data values to establish the regression parameters.

Response: Thanks for your constructive suggestions. In the following study, we will design several experiments to analyze this idea.

In addition, the Supplemental Figure, S-7B, shows the authors' passive-microwave based fractional snow cover estimates on a "continuous" scale, from 0 to 100% coverage (per pixel), related to Fig. 8. Based on this image, areas showing low snow cover values, e.g., < 0.2, seem to be appearing in areas like southern Florida (e.g., color-graduation near values of 0.2) and desert regions in the Southwestern U.S., which typically do not see much or no snow at all. In this case, the MODIS-based

"non-snow" areas (in dark blue) look correct, and the lower estimated snow cover fraction values in the passive microwave product seem to overestimate the "low or no snow" fraction values, which is definitely shown in their Figs. 6 and 7.

Response:Thanks. We not only found a significant overestimation in the low reference fractional snow cove value (or snow-free) in this study, but also found this similar results in our latest study (Xiao et al., 2020). And also, the minimum estimate of fractional snow cover in this study approximately 0.01. We defined the problem you proposed as snow cover misclassification from passive microwave satellite data. There are four possible reasons that can explain this snow cover misclassification from passive microwave satellite data:

1) The main source of snow cover misclassification in this study may be the heterogeneity of the land surface. Specifically, the attenuation and scattering of passive microwave radiation are not only caused by the snowpack, but also by other non-snow scatterer (e.g., desert, arid land, cold land) (Grody and Basist 1996; Dai et al., 2017). At the 6.25-km spatial resolution, there may be great various kinds of land cover except snow cover in winter. In low fractional snow cover pixel, however, most area of the mixed pixel is covered by non-snow type (more than 80%). Then, in this pixel, the scattering contribution from non-snow scatterer is more likely to mistakenly considered to be due to more snow cover area or more snowpack amount (Dai et al., 2017). For this reason, Grody's algorithm designed to develop a series rules to filter out these snow-free surfaces for decreasing snow cover overestimation (Grody and Basist, 1996). From another aspect to understand the radiation scattering contribution of non-snow scatterer, shallow snow depth (~ 0 cm) is seriously overestimated in some cases up to 20 or 50 cm (Che et al., 2016; Xiao et al., 2018) and the shallow snow always occur with low fractional snow cover. Based on this explanation, we knew that the snow cover area is likely to be overestimated in low snow cover condition.

2) The second error source may be the inversion method used in this study. According to the introduction of random forest method in Section 3.4.4, we knew that random forest predictor (classification or regression) is an ensemble predictor that produces multiple decision trees. The final predicted output by the random forest predictor is determined by averaging all the results of the established sub decision trees (Breiman, 2001; Belgiu and Drăguţ, 2016). This may result in the final predicted result would be greater than 0 when the true value is close to 0. These overestimation results near 0 also can be found in other studies applying random forest method (Wei et al., 2019). Under the condition of low snow cover, some sub decision trees may overestimate the snow area in sub-pixel, and then average with other sub decision trees, the final output will be greater than 0 or greater.

3) The third error source of snow cover misclassification may be explained by the principle difference between optical and passive microwave satellite sensors in observing land surface snow cover. Optical sensor is based on snow have high reflectance in the visible band while passive microwave sensor is based on that microwave radiation signal can be scattered by ice crystal particles in snowpack (Grody and Basist 1996). The sensitivity of the optical sensor to detect snow cover is always higher than that of the passive microwave sensor in low snow cover area. This area always is covered by shallow snow, which means that the optical sensor is relatively easy to identify snow cover from other land cover types.

4) In addition to the above three explanations, the occurrence of cold precipitation also can be an error source for snow cover area overestimation. Our recent study (Xiao et al., 2020), in which we analyzed the relationship between the estimated results and climatic factors at several meteorological stations, found the occurrence of cold precipitation can contribute to the overestimation of snow cover area when ground is snow-free (or low fractional snow cover) condition, as described in Section 4.4 this study "The records, which were misclassified as snow cover by random forest SCA, although they are non-snow scatter

components (precipitation, cold desert, and frozen ground), account for 72.6% of total misclassification records (CE = 0.17), of which 64.3% comes from precipitation, 7.0% from cold desert, and 0.9% from frozen ground"

We revised the explanation statements in section 4.3: "*For the snow cover area overestimation, one possible error source is the used inversion method, of which the final predicted outputs are obtained by averaging all results of the established sub decision trees (see Section 3.4.4) (Breiman, 2001; Belgiu and Drăguț, 2016) and this overestimation can be found in other study applying random forest study (Wei et al., 2019). Additionally, the attenuation and scattering of passive microwave radiation are not only caused by the snowpack. The non-snow scatterer (e.g., precipitation, cold desert, frozen ground) may be the majority error source potentially contributing to the overestimation of snow cover area as these scatters were easily misclassified as snow cover in less snow cover conditions (Grody and Basist, 1996; Dai et al., 2017). A detailed analysis on the misclassification due to non-snow scatterer is provided in Section 4.4.*" in page 19 lines 8-16.

What would be helpful in relation to Figs. 8 and S-7, would be to see an example of the same panels, e.g., like with respect to Fig. 8, but a case where MODIS has less cloud presence to show spatially the similarities and differences between the passive microwave product to the optical based sensor.

Response: Thanks for your suggestion. The purpose of Figure S-7 is to present a continues value comparison of MODIS and the PM-based estimated fractional snow cover as a reference of Fig. 8. As we described in this study, the rigorous filter rules make it is really hard to access more intermediate fractional values. Specially, according to our analysis, the numbers of values of MODIS ranging 0.2 ~ 0.8 are no more than 1500 pixels within a day in most cases. From several MODIS fractional snow cover images at 6.25 km, we found the intermediate value is always around the 100% and 0% snow cover area. But we think

*this suggestion can be applied in future study when comparing different fractional snow cover products.*

In addition to these issues, the motivation and methods described are disconnected at times in the paper, and the authors should want to present aspects of their study in a more coherent way. For example, the Section 3.1 Overview section is very helpful to provide readers a "roadmap" as to what is to come and expected in the study, but even the purpose of exploring the four different methods in relation to introducing the fourth method as an hypothesized better approach is simply stated, but those connections seem to be lacking in the Introduction and Section 3.1. I would recommend stating more directly in the Introduction section as to why the random forest regression was selected and provide a short description as what are its advantages, relative to other methods previously applied. I feel the first paragraph in your Section 3.4.4 subsection (on page 12) should be given earlier on in the paper, like in the Introduction section, as to why it is being pursued as an application and how it contributes to your overall hypothesis.

Response: Thanks for your suggestions and comments. We removed the description of random forest from Section 3.4.4 to Section 3.1 as following:

"*Random forest is an ensemble learning method, gaining the attention of many researchers because it is more efficient and robust than the single method (Breiman, 2001). As a classifier, random forest has been successfully employed to detect snow cover (Tsai et al., 2019), land cover (Rodriguez-Galiano et al., 2012), and woody invasive species (Kattenborn et al., 2019). The random forest regression method can also successfully estimate land surface temperature (Zhao et al., 2019), biomass (Mutanga et al., 2012), and soil moisture (Qu et al., 2019). In this study, random forest regression (described in Section 3.4.4) was selected as the retrieval method to mine the relationship between passive microwave brightness temperature and fractional snow cover. We also compared random forest with other three methods (linear regression, MARS and ANN) widely used in fractional snow cover retrieval from optical remote sensing data in model performance. Fig. 2 provides an overview of the workflow that consists of four parts*" in page 7 lines 7-15.

Specific Comments

How were the specific years available for the SSMI/S PM dataset selected originally for: 1) training, 2) testing and 3) evaluation? For example, was 2010 simply selected randomly for the testing year of the four different snow cover retrieval algorithms? If you performed all your algorithm testing and training for the datasets used for different sets of years, do you think you would obtain the same level of results and validation, even though samples are drawn randomly? For example, if you had chosen 2008-2014 for training, 2015 for testing and 2016-2017 for validation, would you expect your table values (e.g., Table 5) and plots to look similar in nature (e.g., Fig. 7) as currently shown? Why or why not?

Response: Thanks for your comments. One thing needs to be emphasized and clarified again is that the selection of experimental data is random. Here, we provide details of the process of experimental design changes. At the beginning, the designed experiment was only using 2014 - 2016 data for training and 2017 for testing and evaluation, but we found the intermediate value in the training data is not enough, and then we updated the experiment design by extending the study period to 2011 for training sample. Moreover, in order to ensure that the evaluation and analysis in different parts independent, we added the data of 2010 used in Section 4.3 for testing the performance of different methods. Up to now, we added another two years data in 2008 and 2009 in evaluation and validation stage following your suggestions.

From the data selection aspect, the year of data used in the training, testing, or evaluation phases does not affect the final experimental conclusion. The most important thing is to ensure enough representative training data sets for establishing the retrieval model. The purpose of the data used in the evaluation stage and testing stage is to evaluate the performance of established retrieval models. Specifically, in our latest study (Xiao et al., 2020), we only used two years data set to provide the training samples because there is less cloud and provide more MODIS reference data in other two seasons (autumn and spring). And the results showed a same level of accuracy in January and February. Moreover, the widely used random forest method is robust because it is based on ensemble learning strategy and has high generalization ability.

In my opinion, if you can give more representative and enough number of training samples, the experimental design as you suggested can also obtain the same level results.

Reference:

Xiao, X., He, T., Liang, S.: Retrieval of fractional snow cover from passive microwave data coupled with microwave radiation transfer model and machining learning method, In preparation, 2020

If you used more years (e.g., 20+ years) of overlapping satellite passive microwave sensor datasets, e.g., different SSM/I versions, could you apply the random forest regression approach separately for each month (and more months) and by region (e.g., U.S. Rocky Mountains, Upper Great Plains), since each of these categories would have distinct snow characteristics, like differences in snow densities (e.g., coastal mountains vs. continental plains regions, or accumulation vs. ablation periods), which can affect brightness temperature signals?

Response: It's a valuable suggestion and comment. Thank you very much. Because of high heterogeneity in coarse resolution of passive microwave data, it is necessary to consider more influential factors to improve the fractional snow cover retrieval from passive microwave data.

As factors you mentioned above, all of these would affect the brightness temperature by different ways and means. For example, in mountainous areas, the complex mountainous terrain will affect the distribution of snow cover, as well as the climatic factors, and then affect the snow property in the local area, thus affecting the scattering of microwave radiation signal observed by satellite sensor. When the study area is relatively small, e.g., U.S. Rocky Mountains, or Upper Great Plains, the method could be applied into different seasons associate with the effect of different forest cover fraction and climate conditions (wind speed, air temperature, soil temperature, etc.). From our recent experiments, we found the time variation and weather conditions play an important role in fractional snow cover estimation from passive microwave data.

Abstract: The authors state around line 14: "under all-weather conditions", which I feel is not representative to summarize in the abstract. The authors can say that they applied the random forest regression technique, along with three other retrieval algorithms, for "dry snow conditions in peak Northern Hemisphere winter months (January and February)."

Response: Thank you. We revised this statement to "… *to estimate fractional snow cover over North America in winter months (January and February)*" in lines 14-15

Abstract (around lines 19-20): The authors mention that they obtain "with higher accuracy and no out-of-range estimated values" associated with the random forest regression technique. They may also want to note that this algorithm tends to

underestimate upper fractional snow cover values and especially overestimates when no to little snow fraction is present, in comparison to the reference snow cover dataset.

Response: Thanks for your comment. The description was changed to "*Although over- and under-estimation around two extreme values of fractional snow cover, the proposed retrieval algorithm out-performed the other three approaches (linear regression, artificial neural networks, and multivariate adaptive regression splines), using independent test data for all land cover classes, with higher accuracy and no out-of-range estimated values*" in lines 18-21.

Page 4, Lines 26-27: The authors state here that they used "bilinear interpolation" to "aggregate the 3.125 km spatial resolution data to 6.25 km". Typically, bilinear interpolation is used for downscaling coarser grid datasets to finer grids. How was such an interpolation used here for "aggregation"? Please provide additional background on why this was selected, versus an averaging or mode-based scheme.

Response: Thank you. We revised "… bilinear interpolation was used to aggregate the 3.125 km spatial resolution data to 6.25 km" to "… *we aggregated the 3.125 km spatial resolution data to 6.25 km by averaging the surrounding 4 pixels*" in page 4 lines 27-28.

Page 5, lines 7-8: Authors state that GHCN-D data from 50,000 sites across Canada and US were collected and included in this study. Actually, this number is higher than what is available for the sites with actual snow depth measurements taken (e.g., may be closer to 10,000 stations for the year of interest, e.g., 2017). The authors may want to mention here how many stations are exactly included in their ground station analysis.

Response: Thanks. We revised the statements to "*More than 50 000 measurement sites across Canada and United States were collected, and all available records applied in validation stage are from 18000 sites*" in page 5 lines 9-11.

Pages 6-7, Section 3.1: I would recommend, early on in this section, mentioning that you compare your application of random forest regression as a new retrieval method against the other three known methods (linear regression, MARS and ANN), and specifically mention them in the first paragraph of this section. The reason is that these other methods have been applied before and you are demonstrating the application of this fourth method. If it has never been applied in such an application related to passive microwave snow cover estimates, you will want to highlight that as both novel and part of your hypothesis that you are testing in deriving snow cover estimates from passive microwave brightness temperatures.

Response: Thanks for your suggestion. We add the description "*We also compared random forest with other three methods (linear regression, MARS and ANN) widely used in fractional snow cover retrieval from optical remote sensing data in model performance*" in the first paragraph of Section 3.1 (page 7 lines 13-15)

Page 13, lines 6-7: The authors state here that "As several attempts to optimize the parameters of random forest structure had failed, all parameters used were the default values." Can you further describe here then what default values you began with, and one of the approaches you applied to optimize the parameters?

Response: Thanks. When you go into the software WEKA and select random forest method, all parameters are list in the panel: batch size (100), max. depth (unlimited), number of Features ($\log_2(\text{predictors}) + 1$), number of iterations (100) and other 12 parameters. In the parameter optimization experiments, we mainly changed above 4 parameters: batch size, max. depth, number of Features, number of iterations.

Pages 17-18: Please expound here on why the shrub and bare land types may be experiencing more "binary" type extremes ("more distributed at two polar ends") for snow cover presence (shown in Figs. 7B,b, and D,d) than the other two land types.

Response: Thanks for your comments. We just randomly selected the dataset following the description "*The independent data, which was randomly selected from the datasets in 2008 – 2009 and 2017 and the selecting rule is same as the training sample, was used to further evaluate the predictive capability of random forest models in all range values*" in Section 4.3. The selected datasets are presented in Fig. 7. We do not control MODIS data selection except that using the rigorous cloud filter rules. As to why more cloud cover during this period, this is beyond the scope of this study.

Page 18, near lines 10-15: "This means that the pixel was identified as snow cover when fractional snow cover value was less than 0.3." In this last sentence here, did you mean to that "the pixel was identified as snow cover when fractional snow cover value was greater than 0.3"? The authors may want to check what they reported here.

Response: Thank you very much. We revised the sentence to "*This means that the pixel was identified as snow cover when fractional snow cover value was greater than 0.3*" in page 18 lines 27-28.

Page 19, lines 1-2: What is meant by "overestimated (~0) and underestimated (~0)"? Did you mean to make the "0" with "overestimated" a "1"? Please clarify what is meant here for the two cases.

Response: Thank you very much. We revised to "*... were overestimated (~0) and underestimated (~1)*" in page 19 line 18.

Section 4.4: For the snow depth evaluation of the random forest approach (in Section 4.4) vs. the Grody SCA algorithm (also using passive microwave retrievals), I believe only two months total of data were used for the validation (Jan-Feb, 2017). Since the GHCN station snow-depth measurements are not used in the training stage, would it be more representative of this part of the validation to use more years, e.g., 2008-2010 and 2017? Two months only of evaluation does not seem sufficient for this validation.

Response: Thanks for your comments. Following your suggestion, we added another two years data in 2008 and 2009 which is same as used in Section 4.3 to validate the snow cover mapping results for the proposed algorithm and Grody SCA algorithm. In this section, we collect more than 18000 meteorological stations. And more than 900 000 records are gathered to validate the capability of snow cover map for these two methods. We updated the presentation of Figure 11. Correspondingly, we modified the description of the statistics in Section 4.4.

[Figure]

Fig. 11. The accuracy indicators (OA, precision, recall, specificity, F1-score, kappa) of snow cover detection from two algorithm (Grody' algorithm; Random forest).

Page 22, line 20: The authors mention that passive microwave based satellites work "around the clock", which is partly true. Most microwave measurements (unless they are from geosynchronous satellites) may have a repeat time over the same location every 2 to 3 days, producing gaps in available data and not continuous (e.g., per day) in time (hence why you see the non-overlapping swaths of "missing" or "filled" areas in panel C of Fig. 8). It might be better to simply remove that phrase here.

Response: Thank you for your suggestion. We removed "around the clock" in page 23 line 6.

Figure 8: Why are the legend categories for Figure 8B and C distributed unevenly and alternating in range values (e.g., 0-0.3, 0.3-0.5, 0.5-0.8, 0.8-1.0)? It seems more appropriate to apply "quarters" and have four equal category ranges (e.g., 0-0.25, 0.25-0.50, 0.5-0.75, 0.75-1.0). Please provide explanation as to how these ranges were selected within the main text.

Response: Thanks for your comment. Each threshold used here has its specific meaning and purpose. This threshold 0.3 was selected through a series of experiments described in Section 4.4. Then this study used 0.3 as a threshold to define snow-covered or snow-free. Typically, 0.5 was chosen as a threshold to define snow-covered and snow-free in fractional snow cover study using optical remote sensing data. Therefore, the using of 0.3 and 0.5 can make other readers or researchers easily compare the capability of snow cover mapping of our proposed method with other methods.

To reduce misunderstandings, I modified the threshold selection description: "*Thus, Fig. 8B and 8C used the threshold 0.3 to define snow-covered and snow-free area, and this threshold was adopted through a series of experiments in Section 4.4. The threshold 0.5 was selected according to previous optical remote sensing study on fractional snow cover (Arsenault et al., 2014; Gascoin et al., 2019)*" in page 18 lines 25-28.

Technical Corrections

Page 8, line 3: Recommend changing "no-forest" to "non-forest".

Response: We changed "no-forest" to "*non-forest*" in page 8 line 13.

Page 8, lines 20-21: The superscripts used to define snow cover for Terra and Aqua MODIS sensors, e.g., "S^Aqua", seem to be reversed for the given satellite. I believe you meant to use "Terra" for the Terra satellite snow cover ground status, "S". "Whether a pixel in Terra (S^Aqua)"… should be: "Whether a pixel in Terra (S^Terra)", and then the same for the Aqua snow cover term following that.

Response: Thanks. We revised it in page 8 line 31.

Page 17, lines 27-28: You may want to clarify within the sentence that starts as "Fig. 7A and 7a show …", that this corresponds to the forest land type.

Response: Thanks. We added "*For forest type, Fig. 7A and 7a show that …*" in page 18 line 5.

Page 17, line 30: Within the phrase, "best performance on the evaluation data", change "on" to "for".

Response: Thanks. We changed "on" to "*for*" in page 18 line 8.

Page 18, line 3: Remove "can" from "we can found that …"

Response: we removed "can" in page 18 line 12.

Page 18, lines 27-28: You may want to update this sentence to include the additional years you drew randomly from for snow cover validation – 2008 and 2009.

Response: Thanks. We modified this sentence to "… *on independent datasets from 2008 – 2010 and 2017 on each land cover type*" in page 19 line 7.

Description for Fig. 8 on Page 18 should mention the date of the images sooner in the paragraph - February 27th, 2017.

Response: Thanks. We add the date information "*on February 27th 2017*" in page 18 line 17.

Page 19, line 10: Add "be" to "may easily be neglected".

Response: Thanks. We revised the sentence to "*… may easily be neglected …*" in page 19 line 28.

Page 19, line 30: Change "In additional" to "In addition".

Response: we changed "In additional" to "*In addition*" in page 20 line 17.

Page 20, line 6: Change: "goo agreement" to "good agreement".

Response: we changed "goo agreement" to "*good agreement*" in page 20 line 24.

Page 20, lines 12-14: Authors accidentally copied or repeated similar information here from Section 4.3, describing the results from Fig. 8, and not related to what is described in this paragraph related to Fig. 11. I recommend them removing these last two sentences in this paragraph.

Response: Thanks. We removed these two sentences in page 20.

Page 21, line 10: The authors may simply want to list the datasets used, such as: "The estimation results of the random forest model [for the training, testing and evaluation] datasets …"

Response: Thank you. we modified the sentence to "*The estimation results of the random forest model (for the training, testing and evaluation datasets; Sections 4.2 and 4.3) showed that* …" in page 22 lines 5 - 6.

Page 21, line 14: Change "this" to "these" in the phrase, "Even in [this] cases …"

Response: Thanks. We changed "this" to "*these*" in page 22 line 10.

Page 22, line 10: Typically, the terms used here, "under-forested and over-forested" in terms of snow cover are more often referred to as: "under or above forest canopy". I would recommend updating this paragraph to reflect the use of these terms.

Response: Thanks. We changed the description "under-forested and over-forested" to "*under or above forest canopy*" in page 23 lines 4-9.

Table 5 caption: Change "brackets" to "parentheses". Brackets are different from what is used in the table.

Response: we changed "brackets" to "*parentheses*" in Table 5 caption.

Table 7: This table and its caption are hard to follow, especially what the percentages are being reported on the right. Please further note what is meant in the caption and what are indicated by the percentages.

Response: We added the data in 2008-2009 in validation stage in section 4.4 and Table 7 was removed.

Figure 2: The diagram still shows for the fractional snow cover "evaluation" dataset – just 2017. I believe the authors added 2008-2009 to the validation dataset. Is this correct? If so, authors may want to update this diagram to reflect the total years used in the validation period.

Response: Thanks. We updated Figure 2.

[Figure]

Figure 9 B: X-axis label is misspelled – change: "Snow detph" to "Snow depth".

Response: Thanks. We updated Figure 9

[revised manuscript text omitted]

---

## Author Response (AR3)

Dear editor,

We would like to thank you and the reviewer for the insightful comments and professional suggestions to improve our manuscript. We have carefully revised the manuscript according to your and reviewer's suggestions, and provide point-by-point response following each comment and suggestion.

In the following texts, reviewer comments are shown in black and responses are given in blue (the revised sentence of the manuscript was set in *italics*). The corresponding changes have been made in the revised paper with track changes. We mainly did the following modification in the revised manuscript:

1) We added additional experiments in fractional snow cover evaluation section (Section 4.3). The data in December (2007, 2008, 2016) was used to analyze the predictive performance of the established retrieval models. The results were shown in Table 6 and Fig. 8.

2) We provide more detailed explanation to the evaluation results of the estimated fractional snow cover.

3) We added the description "Noted that: the testing data used is the entire extracted data in 2010" to Fig. 6 caption and the text "Noted that: the sample selection rule of evaluation data is same as the training data" to the caption of Fig. 7 and Fig. 8.

We think the revised manuscript has addressed all the reviewers' comments and believe it is now suitable for publication in The Cryosphere.

Sincerely,

Xiongxin Xiao

**Response to Reviewer #2**

Overview and General Comments:

The authors have further improved their manuscript with this latest round of edits and updates, in response to the editor and reviewers' inputs. Overall, the paper seems suitable for publication in The Cryosphere, but a couple of final issues remain that make it difficult to suggest for one of the "accepted" categories.

First, this paper remains mainly a feasible study for applying the random forest approach in determining fractional snow cover fraction from passive microwave and optical based sensors, since only two winter months are considered and no other snow season-based months. The authors stated in their latest response to the reviewer and editor that they have written a second paper that explores these issues in the fractional snow cover retrievals from passive microwave data, addressing additional months and how accuracy varies by month and snowpack properties. One key aspect mentioned is that accuracy decreases outside of the two winter months featured in this paper.

Second, the authors noted in their response to the reviewer that it would take another four months in which to generate other months outside of just January and February, including just one or two additional months (i.e., December or March). If the model applied in this study is not easily adaptable to other winter based months, then can this approach be considered effective and 'efficient' enough to apply for, say, climate studies (one of the original identified targets)?

Response: Thanks for your comments and constructive suggestions to enhance our manuscript. Following your suggestions, we implemented additional experiments to examine the adaptability of the established fractional snow cover retrieval model for other months. Based on previous studies on snow cover observation properties observation and analysis (Dai et al., 2012; Zhong et al., 2014), snow physical properties (e.g., grain size, density) show small fluctuation and hold relatively stable during December to February (called snow stabilization stage). After entering the Spring (March to May), with the rise of temperature, snowpack will begin to melt, and the snow grain size and snow density will increase. As a result, snowpack microwave scatter features have significant differences between these two seasons. To avoid introducing the influence of snow physical properties evolution into the evaluation tests, here, the datasets in December (2007, 2008, and 2016) were selected as independent dataset to evaluate the retrieval models over different land cover. The independent evaluation results are shown in Table 6 (Evaluation-2). Correspondingly, we modified the description of evaluation results in Section 4.3. We add the following statements in the revised manuscript (Section 4.3): "*Another independent data in December (2007, 2008, 2016) was selected to examine the predictive capability of the established retrieval models in fractional snow cover to other month. To avoid the influence of snow physical properties evolution in the evaluation tests, we only considered December (Xiao et al., 2018)*" in page 17 lines 28 – 30.

"*In the second evaluation experiments (Evaluation-2 in Table 6; Fig. 8), the best performance in predicted fractional snow cover is over prairie and the relatively large underestimation can be seen over forest (MAE and RMSE). Meanwhile, we do not see striking differences between these two evaluation experiments (Evaulation-1 vs. Evaluation-2) with respect to their RMSE values. The difference of the used evaluation sample can explain the slight diversity in statistics metrics (R).*" in page 18 lines 6 – 10.

Table 6. The performance of random forest models on training, and evaluation datasets over four land cover types. Evaluaiton-1 is used to evaluated the estimation performance of the established retrieval models on fractional snow cover in January and February. Evaluation-2 is used to analyze the prediction performance of the retrieval models in December.

| Land cover type | Training | | | Evaluation-1 | | | Evaluation-2 | | |
|---|---|---|---|---|---|---|---|---|---|
| | R | MAE | RMSE | R | MAE | RMSE | R | MAE | RMSE |
| Forest | 0.702 | 0.166 | 0.207 | 0.636 | 0.180 | 0.221 | 0.658 | 0.180 | 0.222 |
| Shrub | 0.772 | 0.146 | 0.191 | 0.712 | 0.160 | 0.212 | 0.643 | 0.167 | 0.223 |
| Prairie | 0.807 | 0.142 | 0.182 | 0.752 | 0.148 | 0.189 | 0.762 | 0.166 | 0.213 |
| Bare land | 0.807 | 0.144 | 0.190 | 0.719 | 0.165 | 0.216 | 0.744 | 0.162 | 0.217 |

[Figure]

Fig. 8 The color-density scatter plots between the estimated fractional snow cover and MODIS-derived fractional snow cover in December for four land cover types: forest (A), shrub (B), prairie (C), bare land (D). Statistics metrics refer to Table 6: Evaluation-2. Noted that: the sample selection rule of evaluation data is same as the training data.

A couple of other specific issues that should be addressed include:

10 -The new Figure S-7 included still does not show much spatial coverage comparison for the MODIS vs. passive-microwave retrieval fractional snow cover products. We know MODIS snow cover products can be become quite obscured by clouds or when rigorous rules are applied during the MODIS pixel aggregation process. However, with the two example figures provided in this paper, it is difficult to see if the passive microwave-based version of snow cover fraction is truly representative as the optical based one, when clouds are not an issue. The authors stated that this is still

15 for future areas of study to be explored.

Response: Thank you very much for your comments. With respect to the conditions without clouds, we have extracted data pair in several years to evaluate the representatives of the predicted fractional snow cover against the MODIS

reference data (Section 4.3). From the evaluation results (Fig7 and Fig. 8), we can conclude that proposed retrieval algorithm can predict the snow cover area in sub-pixel scale with less error.

In addition, limited by the fact that the evaluation results in Section 4.3 are dependent on MODIS reference fractional snow cover, the ground snow depth measurements, which are not affected by cloud, were used to evaluate the estimation in Section 4.4. The results show the proposed retrieval algorithm can well detect the snow cover area with high overall accuracy (> 0.85) and has improved snow cover identification capability by 20%. This similar statement has been provided in the manuscript:

"*In winter with clouds and snow cover, the MCTD10A1 data still contained a large number of clouds (Fig. 9A; yellow) despite the implementation of the cloud removal and filling process for MODIS snow cover data. When applying the rigorous pixel filters (see Section 3.2), there was very little snow cover data for further model training and results analysis in one imagery (Fig. 9B). To evaluate and validate the estimated fractional snow cover in the absence of reference MODIS fractional snow cover we conducted further analysis on snow cover detection capability. The ground snow depth measurements were utilized to investigate the accuracy of snow cover identification from two snow cover data*" at the beginning of Section 4.4.

Currently, these two kinds of evaluation approaches have been widely applied in evaluating and analyzing the fractional snow cover estimation derived from optical remote sensing data (Gascoin et al, 2019). However, this work serves as an exploratory study with passive microwave remote sensing data, and the underestimation and overestimation still exist in the primary estimation results. Therefore, the fractional snow cover obtained from passive microwave data is a challenging research field, which urges many researchers to solve this problem together.

-It is recommended that the authors "polish" some of their recent updated and added text, since some of the grammar or syntax may be a bit off or incorrect.

Response: Thanks for your constructive suggestions. The added text in last updated version has been polished by a native English speaker.

In summary, this most recently submitted version of the manuscript and response to reviewers seem sufficient overall. However, if the authors have already worked on a separate paper that addresses the shortcomings outlined in this

feasibility study, then the reviewer would suggest not to accept this paper and encourage the submission of the paper instead. If the editor would like to accept this paper, the reviewer would have no other serious objection.

Response: Thank you very much for your comments and suggestions to enhance our manuscript. The above mentioned three issues all have been discussed through adding other experiments or providing more detailed explanations. Moreover, it should be noted that all above related experiments and materials does no refer to any articles that have not yet been published

[revised manuscript text omitted]

---

## Author Response (AR4)

Dear editor,

We would like to thank you for the professional and constructive comments to improve out manuscript. We carefully revised the manuscript according to your suggestions, and provide point-by-point response following each comment. In the following texts, your comments are shown in black and responses are given in blue (the revised sentence of the manuscript was set in *italics*). The corresponding changes have been made in the revised paper with track changes.

We think the revised manuscript has addressed all the reviewers' comments and believe it is now suitable for publication in The Cryosphere.

Sincerely,

Xiongxin Xiao

**Editor's comments**

Second, I have the good news that I have decided to accept your manuscript subject to minor revisions. Although your implementation of reviewer suggestions was overall limited (e.g. including melting seasons, expanding on spatial comparison MODIS), I think the manuscript could now be accepted if you implement the following minor adjustments:

- text edits as suggested in attach

Response:

Page 1 line 10: we revised "… regional variations of snow cover However, …" to "… regional variations of snow cover. However, …".

Page 3 line 14: we changed "swathe" to "swath", and modified "… generate a massive amount of daily observations that …" to "generate a large amount of daily observation that …".

Page 9 line 10: we changed "…should be a larger footprint area…" to "should have a larger footprint area…".

Page 9 line 29: we changed "worth" to "importance"

Page 14 line 10: we modified "optic" to "optical".

Page 16 line 8: we modified "increase" to "increases".

- (See my earlier comment which was not implemented): The differences in sampling strategy between Fig. 6 (S4-S6) and 7 should be clarified. Fig. 6 clearly shows a binary distribution and Fig. 7 not. I assume this is the result of the stratified sampling, but this is never clarified clearly in the text although it is essential.

Response:

[revised manuscript text omitted]

---

## Author Response (AR5)

Dear editor,

We would like to thank you for the professional and constructive comments to improve out manuscript. We carefully revised the manuscript according to your suggestions, and provide point-by-point response following each comment. In the following texts, your comments are shown in black and responses are given in blue (the revised sentence of the manuscript was set in *italics*). The corresponding changes have been made in the revised paper with track changes.

We mainly added the sample selection description of the testing and the evaluation data of Fig. 6 – 8.

We think the revised manuscript has addressed all the reviewers' comments and believe it is now suitable for publication in The Cryosphere.

Sincerely,

Xiongxin Xiao

**Editor's comments**

My main concern is that the data distribution of points looks very different for fig. 6D (almost bimodal) and fig. 7A&a (more intermediate fractions) and it is not clear how this is possible and why? You would expect rather similar distributions (as they are sampled from the same dataset with only different years). So please clarify in each caption where the data come from highlighting the a) years, b) months and c) sampling strategy. Now, when you say entire records of 2010, but it is still not clear which months, for example. I then guess it is also Jan and Feb 2010 (and perhaps Dec) as your study only focuses on these months and always neglects the other months? If so, what is the difference with fig.7 where also only Jan/Feb 2008/2009/2017 are used. So why does the distributions look that different? And why do you have so much more points in 2010 than in 2008/2009/2017.

Response: Thank you very much for your suggestions. Fig. 6 used the entire extracted data in 2010 (January and February). However, the evaluation data used in Fig. 7 and 8 only accounts of 0.3% of total extracted data in three years. Therefore, Fig. 6 and Fig.7 have significant differences in point distribution. Details about the sample selection indeed need to be added in Fig. 6, 7 and 8. We adopted your suggestions and modified the data selection description.

Page 40 line 5: we revised the description in the caption of Fig. 6 to "Noted that: all extracted records in January and February 2010 were used as the testing sample". This revision also is done for Fig. S4-S6.

Page 42 lines 4-5: we revised to "0.3% of the independent evaluation dataset in January and February of 2008, 2009 and 2017 were randomly selected as the evaluation sample with stratified random sampling" in the caption of Fig. 7

Page 42 lines 9-10: we revised to "0.3% of the independent evaluation dataset in December of 2007, 2008 and 2016 were randomly selected as the evaluation sample with stratified random sampling" in the caption of Fig. 8

[revised manuscript text omitted]

Fig. 8 The color-density scatter plots between the estimated fractional snow cover and MODIS-derived fractional snow cover in December for four land cover types: forest (A), shrub (B), prairie (C), bare land (D). Statistics metrics refer to Table 6: Evaluation-2. Noted that:  0.3% of the independent evaluation dataset in December of 2007, 2008 and 2016 were randomly selected as the evaluation sample with stratified random sampling.

[Figure]

Fig. 9. Comparison of our estimated fractional snow cover (C, 6.25-km) with the reference MODIS fractional snow cover (B, 6.25-km) with respect to the MODIS composite binary snow cover products (A, 500-m); and a comparison example in the Central Canada area (D) on February 27th, 2017 (2017058). [Cf. the results in continuous value (Fig. S-7 in the Appendix)]

Fig. 10. The changes of accuracy indicators (OA, precision, recall, specificity, F1-score, kappa) for snow cover detection results of two algorithm (A: Grody' algorithm; B: Random forest) with increasing in situ snow depth value.

[Figure]

Fig. 11. The changes of accuracy indicators (OA, precision, recall, specificity, F1-score, kappa) for snow cover detection results with increasing fractional snow cover value (FSC).

[Figure]

Fig. 12. The accuracy indicators (OA, precision, recall, specificity, F1-score, kappa) of snow cover detection from two algorithm (Grody' algorithm; Random forest).

[Figure]

Fig. 13. The mixed snow cover detection map for different condition combinations of Random forest SCA and Grody's algorithm SCA on February 27th, 2017 (2017058). A: $SC_{Grody} = 0 \, \& \, FSC \leq 0.3$; B: $SC_{Grody} = 0 \, \& \, FSC > 0.3$; C: $SC_{Grody} = 1 \, \& \, FSC \leq 0.3$; D: $SC_{Grody} = 1 \, \& \, FSC > 0.3$; $SC_{Grody} = 0$ denotes snow-free (precipitation, cold desert and frozen ground) determined by Grody's algorithm SCA, otherwise it is snow-covered; $FSC \leq 0.3$ denotes snow-free cover detected by random forest SCA, otherwise, it is snow-covered.